# SuperAnimal pretrained pose estimation models for behavioral analysis

Shaokai Ye[1], Anastasiia Filippova[1], Jessy Lauer[1], Steffen Schneider[1], Maxime Vidal[1], Tian Qiu[1], Alexander Mathis [1] & Mackenzie Weygandt Mathis [1] ✉

Quantification of behavior is critical in diverse applications from neuroscience, veterinary medicine to animal conservation. A common key step for behavioral analysis is first extracting relevant keypoints on animals, known as pose estimation. However, reliable inference of poses currently requires domain knowledge and manual labeling effort to build supervised models. We present SuperAnimal, a method to develop unified foundation models that can be used on over 45 species, without additional manual labels. These models show excellent performance across six pose estimation benchmarks. We demonstrate how to fine-tune the models (if needed) on differently labeled data and provide tooling for unsupervised video adaptation to boost performance and decrease jitter across frames. If fine-tuned, SuperAnimal models are 10–100× more data efficient than prior transfer-learning-based approaches. We illustrate the utility of our models in behavioral classification and kinematic analysis. Collectively, we present a data-efficient solution for animal pose estimation.

Measuring and modeling behavior is an important step in many clinical, biotechnological, and scientific quests[1–6]. A key part of many behavioral analysis pipelines is animal pose estimation, yet this requires domain knowledge and labeling efforts to obtain reliable pose models[2,3,7,8]. Open-source pose estimation software, such as DeepLabCut[9,10] and other tools[11–14] also reviewed in[7], have gained popularity in the research community interested in understanding animal behavior. Compared to commercial solutions constrained to fixed cage and camera settings[15], DeepLabCut offers flexibility to train customized pose models of various animals in diverse settings. Notably, it requires few human-labeled images (around 100–800) to train a typical lab animal pose estimator that matches human-level accuracy[9,10] due to its transfer learning abilities[9,16].

However, regardless of the data efficiency of current solutions, their flexibility still comes with the cost of requiring users to label if they want to define keypoints themselves (note, some unsupervised approaches are available, but lack the ability of users to customize to keypoints of scientific interest[17,18]). Then, they train deep neural networks, an effort that is often duplicated across labs given that often users study similar model organisms.

A solution is to build generalized, foundation models[19], for common model organisms across labs and in-the-wild settings (proposed and discussed in[7]). Such models could be used across labs and settings without further training and/or requiring little fine-tuning. Yet, there are several key challenges to building these models. Firstly, data on the same species is rarely labeled the same way or even with the same names (for example consider simply naming the nose on a mouse: "nose", "snout", "mouse1_nose", etc—all found in the literature[9,15]), which brings semantic and annotation bias challenges: how do we merge such data? Secondly, even if we unify the naming, how do we train across datasets that don't have the full super-set of keypoints? Missing data would confuse a network without any interventions.

To provide the research community with an easy method to build such high-performance models we present a new panoptic paradigm—which we call the SuperAnimal method—for building unified pretrained pose-aware models, and the ability to perform fine-tuning and

[1]École Polytechnique Fédérale de Lausanne (EPFL), Brain Mind Institute & Neuro-X Institute, Geneva, Switzerland. ✉e-mail: mackenzie.mathis@epfl.ch

video adaptation across many species, environments and animal or video sizes (Fig. 1a).

In brief, our new approach allows for merging and training diverse, differently labeled datasets. We developed an optimal keypoint matching algorithm to automatically align out-of-distribution datasets with our models. Then, at inference time, to minimize domain shifts, we developed a spatial pyramid search method to account for changes in animal size or leverage a top-down detector. We also provide a rapid, unsupervised video-adaptation method that uses pseudo-labeling to boost performance and minimize temporal jitter in videos and allows users to fine-tune videos without access to source data or requiring any target labeling on that video.

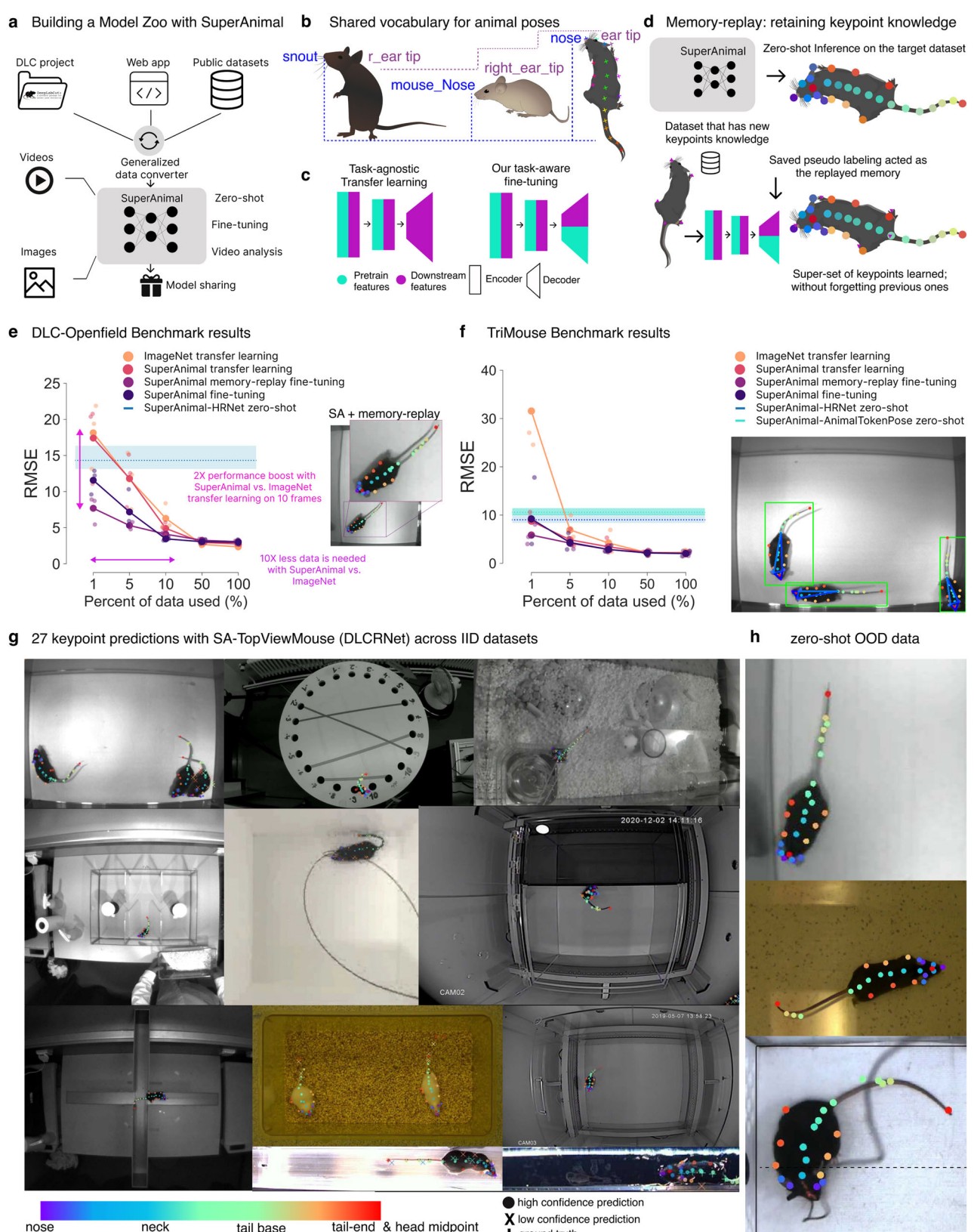

**Fig. 1 | The DeepLabCut Model Zoo, the SuperAnimal method, and the SuperAnimal-TopViewMouse model performance. a** The website can collect data shared by the research community; SuperAnimal models are trained and can be used for inference on novel images and videos with or without further training (fine-tuning). **b** The panoptic animal pose estimation approach unifies the vocabulary of pose data across labs, such that each individual dataset is a subset of a super-set keypoint space, independently of its naming. Mouse cartoons from scidraw.io: https://beta.scidraw.io/drawing/87, https://beta.scidraw.io/drawing/49, https://beta.scidraw.io/drawing/183. **c** For canonical task-agnostic transfer learning, the encoder learns universal visual features from ImageNet, and a randomly initialized decoder is used to learn the pose from the downstream dataset. For task-aware fine-tuning, both encoder and decoder learn task-related visual-pose features in the pre-training datasets, and the decoder is fine-tuned to update pose priors in downstream datasets. Crucially, the network has pose-estimation-specific weights. **d** Memory replay combines the strengths of SuperAnimal models' zero-shot inference, data combination strategy, and leveraging labeled data for fine-tuning (if needed). Mouse cartoon from scidraw.io: https://beta.scidraw.io/drawing/183. **e** Data efficiency of baseline (ImageNet) and various SuperAnimal fine-

tuning methods using bottom-up DLCRNet on the DLC-Openfield OOD dataset. 1–100% of the train data is 10, 50, 101, 506, and 1012 frames respectively. Blue shadow represents minimum, maximum and blue dash is the mean for zero-shot performance across three shuffles. Large, connected dots represent mean results across three shuffles and smaller dots represent results for individual shuffles. Inset: Using memory replay avoids catastrophic forgetting. **f** SuperAnimal vs. baseline results on the TriMouse benchmark, showing zero-shot performance with top-down HRNet and AnimalTokenPose, and fine-tuning results with HRNet. 1–100% of the train data is 1, 7, 15, 76, and 152 frames respectively Inset: example image of results. **g** SuperAnimal-TopViewMouse (DLCRNet) qualitative results on the within-distribution test images (IID). They were randomly selected based on the visibility of the keypoints within the figure (but not on performance). Full keypoint color and mapping are available in Supplementary Fig. S1). **h** Visualization of model performance on OOD images using DLCRNet. (**e, f, g**) Images in (**e–h**) are adapted from https://edspace.american.edu/openbehavior/video-repository/video-repository-2/ and released under a CC-BY-NC license: https://creativecommons.org/licenses/by-nc/4.0/.

We developed models based on state-of-the-art (SOTA) convolutional neural networks (CNNs) such as HRNet[20] or DLCRNet[10], and transformers[21–23]. We show that the resulting models have excellent "zero-shot" performance, which we define to mean testing performance on unseen datasets that include animals not used in training, and critically because we set up our pose estimation task as a panoptic segmentation task where the ground truth data does not have all class labels (key points), it tests the ability of the model to perform inference on a super-set of keypoints not always used in training. Additionally, our approach outperforms ImageNet-pretraining on six benchmarks, which has been the standard in the field of animal pose estimation. If our models are used for fine-tuning, we show they are 10 to nearly 100× more data efficient in the low data regime (and can still improve performance in the high-data setting), and our video adaptation method allows for smooth, refined videos that can be used in behavioral analysis pipelines.

## Results

The SuperAnimal method comprises generalized data conversion, training with keypoint gradient masking and memory replay, a keypoint matching algorithm, and the ability to fine-tune models plus video adaptation that pseudo-labels using unlabeled video data (Fig. 1a), which will be explained below. Firstly, we describe the data we used to train our two exemplary models with our SuperAnimal approach.

### Animal pose data

In order to demonstrate the strength of our SuperAnimal method, we present two datasets that cover over 45 species: TopViewMouse-5K and Quadruped-80K, which are built from over 85,000 images sourced from diverse laboratory settings and in-the-wild data (Supplementary Fig. S1a, b), yet critically they are not labeled in the same manner. First, we used a new generalized data converter (see Methods) to unify the annotation space of those datasets and named the first dataset TopViewMouse-5K (as it contains approximately 5k images). Specifically, we merged 13 overhead-camera view-point lab mice datasets from across the research community[9,10,15,24,25] (see Methods) and from our own experiments (Fig. 1e, h). Similarly, we collected side-view quadruped datasets[16,26–32], including a new annotated "iRodent" dataset with images from iNaturalist (see Methods), to form Quadruped-80K (Supplementary Fig. S1b, and see Supplemental Datasheets). We define six benchmark datasets of varying difficulty and always leave one out of the training in order to show performance of the model on unseen data. There are: DLC-Openfield[9], TriMouse[10], iRodent (new), Horse-10[16], AP-10K[31] and AnimalPose[33]. Note, that our released SuperAnimal-TopViewMouse and SuperAnimal-Quadruped

weights (see Supplementary Information, Model Cards) are trained on all available data described above (Supplementary Fig. S1b).

### The SuperAnimal method

Collectively, SuperAnimal is a formulation that treats diverse pose datasets as if they collectively formed one single super-set pose template, trains unified models for image and video analysis, and ultimately allows sharing of these models through standardized repositories (Fig. 1a). This panoptic super-set approach effectively allowed us to overcome a major challenge with combining datasets that are not identically labeled across labs or datasets, as it is often the case even for the same species (Fig. 1b and Supplementary Fig. S1a). Multi-dataset training allows the model to receive richer learning signals (Fig. 1c, d), resulting in the model having "pose priors" (whereas ImageNet pre-training, is common in animal pose[10,16,33] has no pose-specific features). For multi-dataset training, we developed keypoint gradient masking (Supplementary Fig. S1c, d) to train neural networks across disjoint datasets without penalizing "missing" ground truth data from the super-set of keypoints (Fig. 1b). We also developed a keypoint matching algorithm (Methods and Supplementary Fig. S2a, b) to help minimize the mismatch caused by annotator bias in the human-annotated datasets (see Supplementary Note).

The neural networks we trained always consist of an encoder and a decoder. While transfer learning has been important for animal pose[9,16], we hypothesized now that we have base encoders that have pose priors, that the trained decoder could be leveraged (Fig. 1c, d). Therefore, we tested two ways to train the architectures: one, via transfer learning, defined as fine-tuning only the pre-trained encoder but using a randomly initialized decoder in the downstream dataset; two, via fine-tuning, defined as fine-tuning both the encoder and decoder (see Methods). We also note that we used both bottom-up and top-down methods[7], meaning without or with an object detector, respectfully, as noted in figure captions.

### Benchmarking

We aim to show the performance of our approach in three important settings: (1) zero-shot inference: how performant is the base model on unseen, out-of-distribution data? (2) fine-tuning on a new dataset: how does the base model compare to using a base model trained on ImageNet (i.e., ImageNet transfer learning)? (3) If zero-shot and/or fine-tuning is efficient, how good is the base model performance on videos and for downstream tasks like behavior quantification?

We report results on two model classes: SuperAnimal-TopViewMouse (SA-TVM) then SuperAnimal-Quadruped (SA-Q). For each class, we consider several architectures for zero-shot and fine-tuning (Figs. 1, 2, Supplementary Fig. S2c, d), and then consider

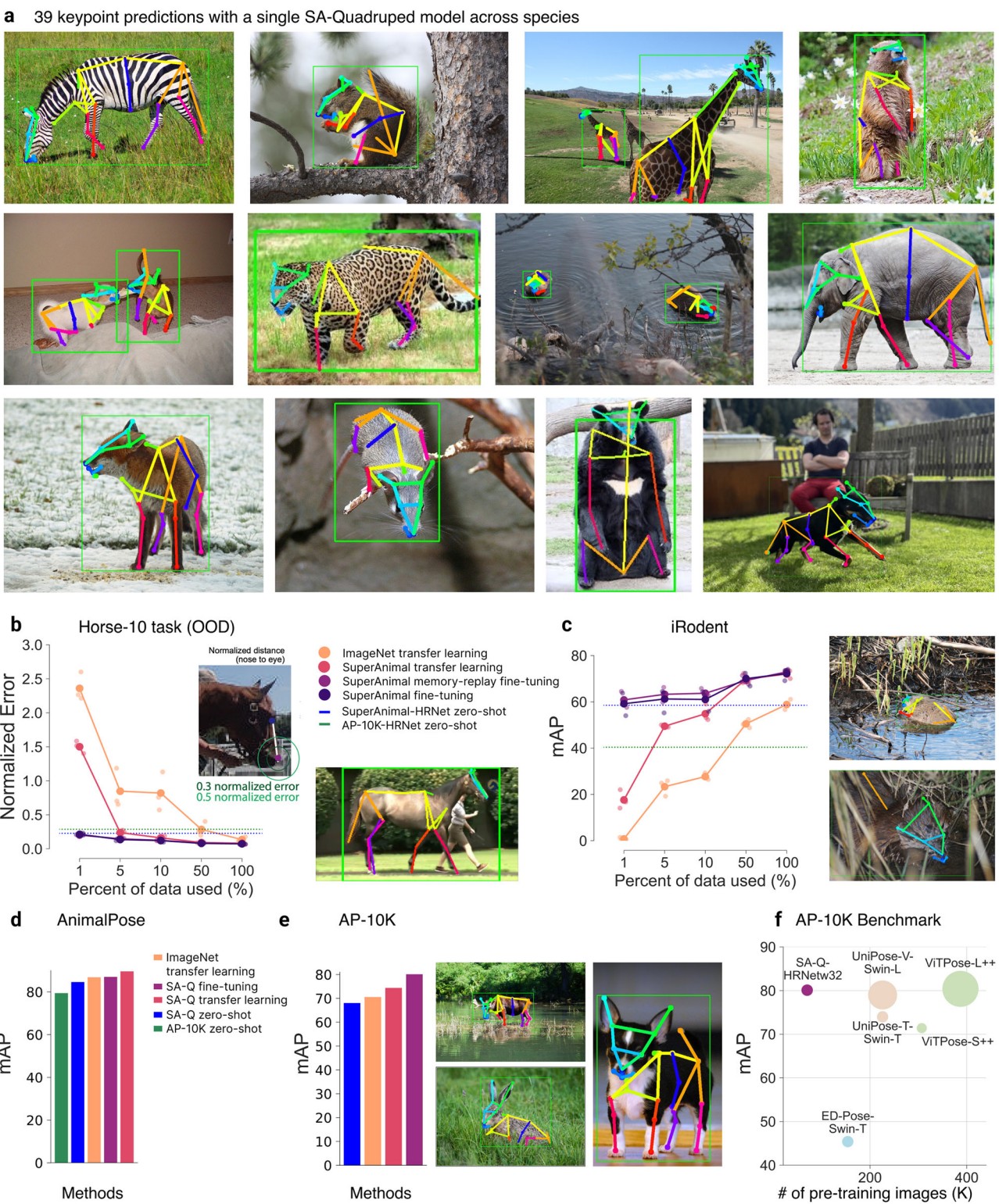

**a**  39 keypoint predictions with a single SA-Quadruped model across species

**b** Horse-10 task (OOD)

ImageNet transfer learning
SuperAnimal transfer learning
SuperAnimal memory-replay fine-tuning
SuperAnimal fine-tuning
SuperAnimal-HRNet zero-shot
AP-10K-HRNet zero-shot

Normalized distance (nose to eye)

0.3 normalized error
0.5 normalized error

**c** iRodent

**d** AnimalPose

ImageNet transfer learning
SA-Q fine-tuning
SA-Q transfer learning
SA-Q zero-shot
AP-10K zero-shot

**e** AP-10K

**f** AP-10K Benchmark

SA-Q-HRNetw32
UniPose-V-Swin-L
ViTPose-L++
UniPose-T-Swin-T
ViTPose-S++
ED-Pose-Swin-T

performance in video and behavioral analysis (Figs. 3 and 4). To evaluate each model's performance we tested "within distribution" also known as "independent and identically distributed" (IID), and on images considered "out-of-distribution" (OOD). IID images are similar in appearance and from the same dataset, but not identical to those used in training. OOD data stems from images in datasets that were never included in training[34], and they constitute the key benchmark results for showing the utility of the models in applied settings (Tables 1 and 2).

**Zero-shot performance (SA-TVM)**

Using the panoptic SuperAnimal approach we first consider the performance in the OOD zero-shot setting and find that it has excellent performance across both top-view mouse benchmarks (Table 1). We tested both a bottom-up DLCRNet (Fig. 1e–h) and a top-down HRNet-w32[10,13,35] (Supplementary Fig. S3), which was recently shown to be excellent in crowded animal scenes[36], and our transformer for testing. Specifically, to test performance, we built SA-TVM models that did not contain data from the DLC-Openfield[9] or TriMouse dataset[9,10].

**Fig. 2 | SuperAnimal-Quadruped. a** Qualitative performance with SuperAnimal-Quadruped (HRNet-w32). Image randomly selected based on visibility of the key-points within the figure (but not on performance). A likelihood cutoff of 0.6 was applied for keypoint visualization. Full keypoint color and mapping are available in Supplementary Fig. S1). Images in panels a and e are adapted from https://github.com/AlexTheBad/AP-10K/blob/main/LICENSE and are under a CC-BY license: https://creativecommons.org/licenses/by/4.0/ Bottom right image is courtesy of the authors. HRNet-w32 and same cutoff of 0.6 are used in other panels. **b** Performance on the official OOD Horse-10 test set, training with the official IID splits, reported as a normalized error from eye to nose, see inset adopted from ref. 16 and qualitative zero-shot performance. HRNet-w32 is trained on AP-10K and Quadruped-80K, respectively, for zero-shot performance comparison. 1–100% of the data is 14, 73, 146, 734, and 1469 frames, respectively. The (**b**) images are adapted from https://www.mackenziemathislab.org/horse10 and released under a CC-BY-NC license: https://creativecommons.org/licenses/by-nc/4.0/. **c** Performance on the OOD iRodent dataset, reported mAP. Colors and zero-shot baseline are as in (**b**). 1–100% of the data is 3, 17, 35, 177, and 354 frames, respectively. See inset for qualitative zero-shot performance. Images in (**c**) are adapted from iNaturalist https://www.inaturalist.org/ and are under a CC-BY license: https://creativecommons.org/licenses/by/4.0/. **d** Performance on the OOD AnimalPose dataset, reported as mAP. HRNet-w32 trained on AP-10K is used as an additional zero-shot baseline. Benchmark images cannot be shown due to copyright concerns, but please see ref. 51. **e** Performance on the OOD AP-10K dataset, reported as mAP. Qualitative zero-shot performance is also shown. AP-10K raw images are licensed under CC-BY: https://creativecommons.org/licenses/by/4.0/. **f** AP-10K benchmark with SA-Q and other pose data pre-trained models. The size of dots represents the parameter size of each model. The number of pre-training images represents the number of pose data models trained before being fine-tuned on AP-10K.

We found that the SuperAnimal methods were critical: using gradient masking SA-TVM DLCRNet zero-shot performance was $14.31 \pm 1.00$ RMSE vs. $27.90 \pm 1.20$ without gradient masking (Supplementary Fig. S4a). Memory replay was critical to avoid catastrophic forgetting qualitatively (see Supplementary Video 1) and quantitatively, measured with keypoint dropping (See Supplementary Fig. S4c and Supplementary Tables S1 and S2).

Collectively, they show excellent zero-shot performance on both benchmarks (Fig. 1e, f, Supplementary Fig. S4c, and Supplementary Tables S3 and S4). SA-TVM performed well within distribution (IID) and critically OOD data across diverse camera and cage settings (Fig. 1g, h). Note that the performance of zero-shot inference is even likely underestimated by annotator bias (see Supplementary Note). Concretely, zero-shot SA-TVM DLCRNet bottom-up showed a RMSE error of 14.31 pixels, 4.88 pixels with the HRNet-w32-based top-down approach, and 4.57 pixels with AnimalTokenPose on the DLC-Openfield dataset, where the average mouse's nose width is ~10 pixels[9] (Fig. 1e, f). Thus, we found that without any labeling we could still outperform ImageNet-based transfer learning (Fig. 1e; mixed-effect model; in the low-data regime (1% training data ratio) for TriMouse: $d = 8.03$ [5.27, 10.79] $p < 0.0001$; Supplementary Tables S5 and S6; for DLC-Openfield: $d = 3.86$ [1.88, 5.84] $p = 0.0002$, Supplementary Tables S7 and S8).

### Fine-tuning performance (SA-TVM)

For fine-tuning SuperAnimal models we consider two ways: one, naive fine-tuning (see Methods), and inspired by the excellent zero-shot inference of pre-trained models[37] and continual learning[38], we developed a tailored fine-tuning approach that combines zero-shot inference and few-shot learning, which we call "memory replay" fine-tuning (Fig. 1d). We find that in the user-relevant low-data regime, fine-tuning significantly outperforms ImageNet transfer learning (Fig. 1e, f; mixed effects model, DLC-Openfield: $d = 7.19$ [4.61, 9.78]; $p < 0.0001$, Tri-Mouse: $d = 8.06$ [5.29, 10.82]; $p < 0.0001$; Supplementary Tables S5, S6, S9, and S10). This is approximately a 10× data efficiency factor and large margin of performance gain (Fig. 1e). Note that effect sizes remain moderate to large even when training with 5% of the data ($d > 0.59$).

For example, if the model is memory replay fine-tuned with only ten randomly selected images on DLC-Openfield, the SA-TVM pre-trained model obtained an RMSE of 7.68 pixels, whereas ImageNet pre-training was 18.14 pixels. The baseline ImageNet pre-trained model required 101 (randomly selected) images to reach a performance similar to SA-TVM (6.28 pixels; Fig. 1e). Therefore, we outperformed DeepLabCut-DLCRNet (i.e., the ImageNet baseline) by over 2X in the low data regime (i.e., with 10 frames of labeling; $p < 0.0001$, Cohen's $d = 4.88$), and we can achieve the same performance as DeepLabCut-ImageNet weights with 10× less data (i.e, using 10 frames with our SA-

TVM memory replay gives the same RMSE as ImageNet transfer learning with 101 images).

One important point is that the SA-TVM model is now imbued with a "pose prior". Historically, the transfer learning using ImageNet weights strategies assumed no "animal pose task priors" in the pre-trained model, a paradigm adopted from previous task-agnostic transfer learning[39]. Yet, here we show that naively fine-tuning on datasets that do not have the full super-set of points might cause catastrophic forgetting (Supplementary Video 2). Namely, if we fine-tuned with the four keypoints dataset from DLC-Openfield, the model would forget the full 27 keypoints.

### Zero-shot performance (SA-Q)

Developing pre-trained animal pose models to work in the wild is a challenging task. There are two main reasons for its difficulty: (1) the lack of labeled data, and (2) the diversity of the data. Firstly, compared to the widely used COCO human keypoint benchmark[40] that has 200K images, the biggest wild animal pose keypoint benchmarks have 10–36K images from AP-10K[31] and APT-36K[32], respectively. Yet even with Quadruped-80K, we generate, the number of images is still much less than that in COCO. Secondly, the appearance size of the animals is a long tail distribution (discussed below), which can pose a challenge for models to learn.

To tackle such challenges, we developed top-down HRNet-w32 based SA-Q models (Fig. 2a), tested zero-shot our transformer (Supplementary Fig. S4d). We tested SA-Q performance on four OOD benchmarks that had various official metrics: Horse-10[16] reports the normalized error (NE, normalized by the animal's size, see inset in Fig. 2b), iRodent, AP-10K[31], and AnimalPose[28] report the mAP. As a reminder, for every benchmark we took a leave-one-out strategy such that the benchmark data was not used for training. Following the common practice in top-down animal pose works[31,41], we report results using ground-truth bounding boxes and flip test in the test time (see Methods).

In addition to ImageNet pre-trained weights as a baseline, in benchmarks other than on AP-10K we also used a HRNet-w32 pre-trained on AP-10K (green dash line or column, Fig. 2b–d) as an additional baseline comparison. For comparisons to the official benchmarks we report the official metrics in Fig. 2. We also report mAP and RMSE for all benchmarks, which can be found in Supplementary Tables S11, S12, S13, and S14.

Horse-10 is a benchmark challenge that tests OOD robustness of generalizing to held-out individual horses. We evaluated on the official splits and show zero-shot SA-Q gives 0.228 NE (Fig. 2b), which outperforms an AP-10K-trained model that achieves a 0.287 NE. Note the current SOTA performance in Horse-10 benchmark is around 0.3 NE with a bottom-up EfficientNet[16]. We thus used top-down HRNet-w32 as a stronger baseline that gives 0.135 OOD NE. Importantly, we find that SA-Q has zero-shot that is as good as supervised training with 50%

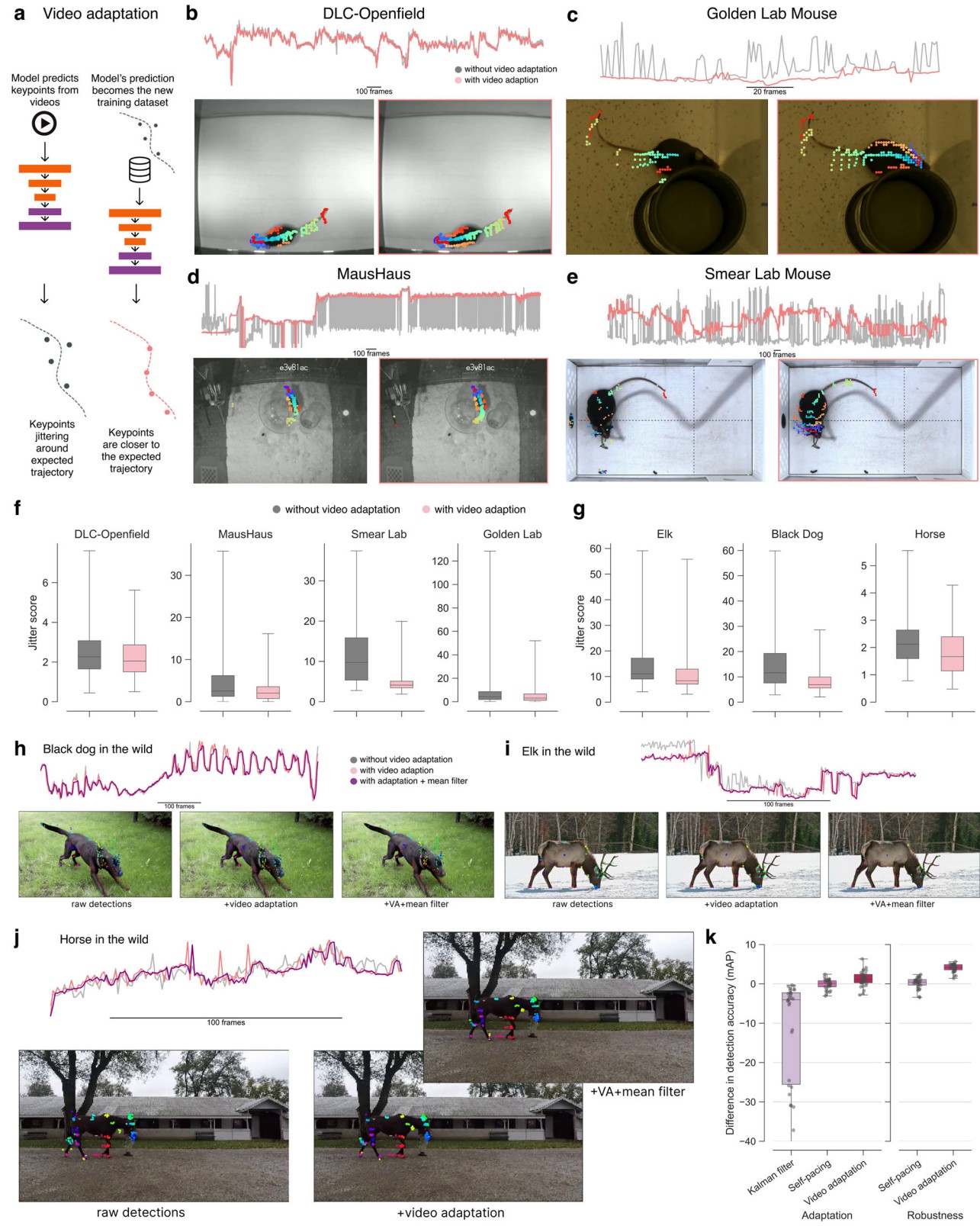

training data (734 images) using the HRNet ImageNet baseline (Fig. 2b), and with SA-Q + fine-tuning we can achieve 0.146 normalized error with 5% (73 frames) of data, and with the full data a 0.07 OOD NE, setting a new SOTA.

iRodent is a challenging new dataset comprising a diverse set of images of rodents that are often under heavy occlusion, have a complex background, and have very various appearance sizes (Fig. 2c), yet

with SA-Q we can achieve excellent zero-shot performance 58.6 mAP (Fig. 2c), which is on par with 58.9 mAP obtained by fully trained (100% training data) HRNet-w32 using ImageNet weights. In contrast, AP-10K's weights gives 40.4 mAP, 18.2 points lower than ours.

On the AnimalPose, which is a benchmark dataset consisting of dogs, cats, cows, horses, and sheep with 20 keypoints[28], our SA-Q zero-shot performance is 84.6 mAP, which is almost on par with the 86.9

**Fig. 3 | Unsupervised video adaptation methods. a** Illustration of the unsupervised video adaptation algorithm. **b**–**e** Animal size described by convex hull of keypoints using the SA-TVM model. Frequent changes of the convex hull indicate non-smooth keypoint predictions and below are example images with and without video adaptation showing the trailing keypoints for 10 past frames of data (to demonstrate the motion smoothness). Images in (**b**) are adapted from https://github.com/DeepLabCut/DeepLabCut/blob/main/examples/openfield-Pranav-2018-10-30/videos/m3v1mp4.mp4 and are under a CC-BY license: https://creativecommons.org/licenses/by/4.0/. Images in (**c**) are adapted from https://edspace.american.edu/openbehavior/project/open-field-social-investigation-videos-donated-sam-golden/ and released under a CC BY-NC-SA license: https://creativecommons.org/licenses/by-nc-sa/4.0/. Images in (**d**) are adapted from Mathis Laboratory of Adaptive Intelligence (2024) "MausHaus Mathis Lab". Zenodo. https://doi.org/10.5281/zenodo.10593101 and is under a CC-BY license: https://creativecommons.org/licenses/by/4.0/. Images in (**e**) are adapted from https://edspace.american.edu/openbehavior/project/olfactory-search-video-donated-matt-smear/ and released under a CC BY-NC-SA license: https://creativecommons.org/licenses/by-nc-sa/4.0/. **f**, **g** Change in jitter score before and after video adaptation. The box plots show jitter scores across test videos DLC-

Openfield (n = 2329 samples), MausHaus (n = 3270 samples), Smear Lab (n = 144 samples), Golden Lab (n = 4859 samples), Elk (n = 265 samples), Black Dog (n = 637 samples), Horse (n = 239 samples). In box plots, the middle line indicates the median. The bounds of the box indicate the first and third quartiles and the whiskers extend to the 0th and 100th percentile. Overall, our method had a significant effect on reducing jitter (linear mixed effect model: $F(1, 23286) = 190.03$, $p < 0.0001$; Supplementary Table S20, in all but the dog ($p = 0.36$, $d = -0.03$) and Golden lab ($p = 0.62$, $d = -0.06$) videos; two-sided post-hoc contrasts with Tukey multiplicity adjustment: Supplementary Table S21. **h**, **i** Same analysis as in (**b**−**e**) using the SA-Q model. Note that additional median filtering post-video adaptation examples (dark purple line) can be used if needed. **j**, **k** Video adaptation, self-pacing, and Kalman filter's performance on the Horse-30 video dataset where (**j**) is an example of one of 30 videos from the dataset Images in (**j**) are adapted from "Pretraining boosts out-of-domain robustness for pose estimation" WACV (Jan 2021) https://www.mackenziemathislab.org/horse10 and released under a CC-BY-NC license: https://creativecommons.org/licenses/by-nc/4.0/. In box plots, the middle line indicates the median. The bounds of the box indicate the first and third quartiles and the whiskers extend to the 0th and 100th percentile.

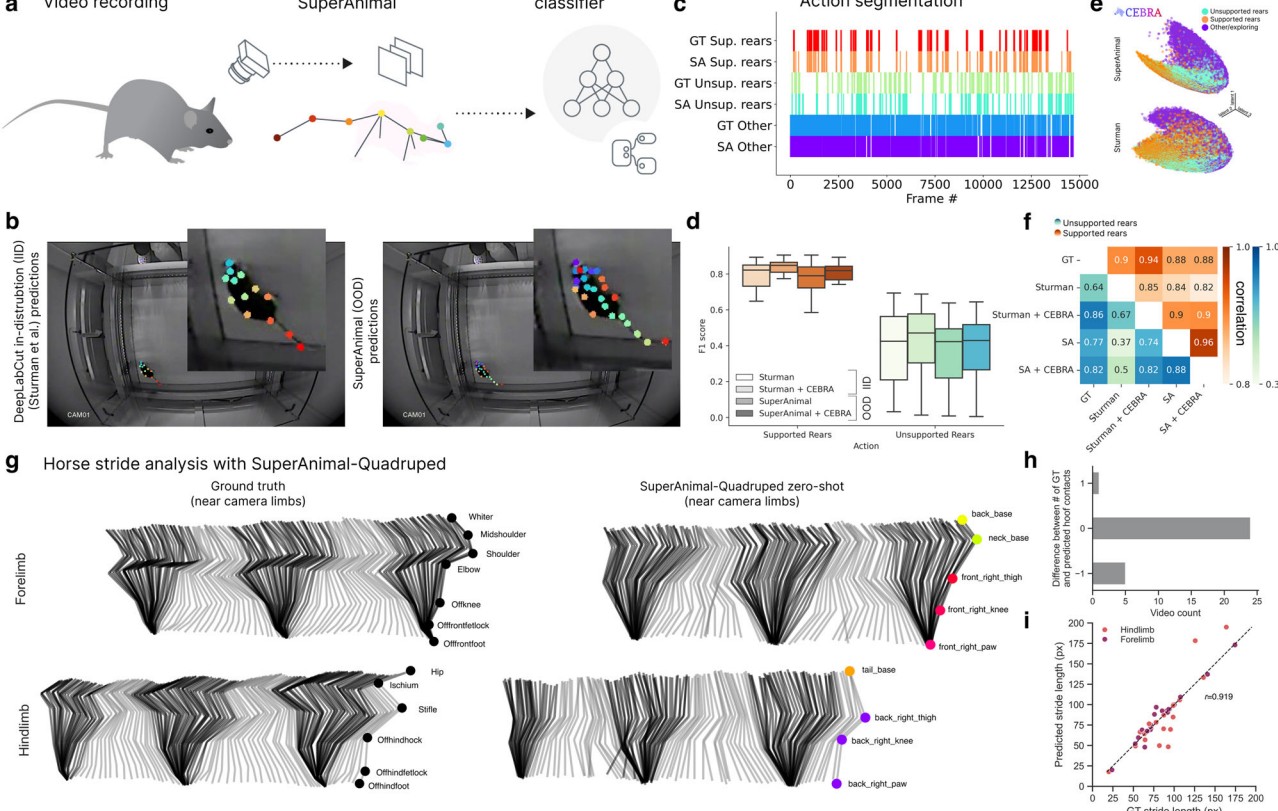

**Fig. 4 | Zero-shot behavioral quantification with SuperAnimal. a** Workflow overview for behavioral analysis with SuperAnimal. Artwork by Gil Costa. **b** Images in (**b**) are adapted from Lukas von Ziegler, Oliver Sturman, and Johannes Bohacek (2020) "Videos for deeplabcut, noldus ethovision X14, and TSE multi conditioning systems comparisons". Zenodo. https://doi.org/10.5281/zenodo.3608658 under a CC-BY license: https://creativecommons.org/licenses/by/4.0 with their DeepLabCut "in distribution" model (left) and our SuperAnimal zero-shot, out-of-distribution, results (right). **c** Ethogram comparing ground truth annotations vs. zero-shot predictions from SuperAnimal-TopViewMouse. **d** F1 score computed across IID (Sturman) and SuperAnimal with, or without CEBRA on the two behavioral classes (n = 20 videos). Box plots' middle lines indicate the median. The bounds of the box indicate the first and third quartiles and the whiskers are drawn to the farthest datapoint within 1.5*IQR. **e** CEBRA[48] embedding on Sturman keypoints and

SuperAnimal-based keypoints in 3D, transformed with FastICA for visualization. **f** Correlation matrix that demonstrates the correlation between SuperAnimal-TopViewMouse and ground-truth annotations averaged across three annotators and across the model and keypoint configurations. **g** We analyzed 30 horse videos where every frame had a ground truth (GT) annotation of keypoints[16] (left) vs. our SuperAnimal-Quadruped model (right). The right limbs (closest to the camera) from one representative gait trial are shown. Swing and stance phases are colored in light gray and black zones, respectively. **h** Histogram delineating the number of videos where the ground contact by the hoof was identical to the GT vs. over or under-counted by one stride (no error larger than one was found). **i:** We computed the error between the GT stride length vs. model prediction for the hoofs (i.e., right_back_paw vs. Offhindfoot, etc). Each dot represents a stride, color denotes hindlimb vs. forelimb, near legs only.

**Table 1 | Main results on mouse benchmarks**

| Method | Pre-trained weights | Data ratio | mAP | RMSE | Dataset | Architecture |
|---|---|---|---|---|---|---|
| Zero-shot | SuperAnimal | – | 50.397 | 14.32 | DLC_Openfield | DLCRNet |
| Zero-shot | SuperAnimal | – | 95.219 | 4.881 | DLC_Openfield | HRNetw32 |
| Zero-shot | SuperAnimal | – | 96.348 | 4.572 | DLC_Openfield | AnimalTokenPose |
| Transfer learning | ImageNet | 0.01 | 62.226 | 18.136 | DLC_Openfield | DLCRNet |
| Transfer learning | ImageNet | 0.01 | 91.458 | 7.001 | DLC_Openfield | HRNetw32 |
| Transfer learning | ImageNet | 1.00 | 99.23 | 2.340 | DLC_Openfield | DLCRNet |
| Transfer learning | ImageNet | 1.00 | 100 | 1.131 | DLC_Openfield | HRNetw32 |
| Memory replay | SuperAnimal | 0.01 | 74.225 | 7.688 | DLC_Openfield | DLCRNet |
| Memory replay | SuperAnimal | 0.01 | 99.599 | 2.381 | DLC_Openfield | HRNetw32 |
| Memory replay | SuperAnimal | 1.00 | 97.946 | 3.071 | DLC_Openfield | DLCRNet |
| Memory replay | SuperAnimal | 1.00 | 99.868 | 1.210 | DLC_Openfield | HRNetw32 |
| Zero-shot | SuperAnimal | – | 76.139 | 9.013 | TriMouse | HRNetw32 |
| Zero-shot | SuperAnimal | – | 70.372 | 10.580 | TriMouse | AnimalTokenPose |
| Transfer learning | ImageNet | 0.01 | 26.116 | 31.562 | TriMouse | HRNetw32 |
| Transfer learning | ImageNet | 1.00 | 97.730 | 2.276 | TriMouse | HRNetw32 |
| Memory replay | SuperAnimal | 0.01 | 90.320 | 5.850 | TriMouse | HRNetw32 |
| Memory replay | SuperAnimal | 1.00 | 98.547 | 2.103 | TriMouse | HRNetw32 |

The mAP on multiple architectures, CNN (HRNet, DLCRNet), and Transformer based models (AnimalTokenPose model) on SuperAnimal-TopViewMouse. As a reminder, transfer learning means using a randomly initialized decoder that is also trained. Memory replay involves fine-tuning the encoder and decoder.

mAP from fully supervised models that also use HRNet-w32 (Fig. 2d). Moreover, we beat the zero-shot 79.4 mAP with HRNet-w32 that was pre-trained on the AP-10K dataset.

Lastly, on the AP-10K benchmark, we used the official train-val splits. The benchmark officially tests fine-tuning performance on the validation set (see below), but first, we tested zero-shot and find with SA-Q a 68.0 mAP (Fig. 2e), which is already close to the reported 71.4 mAP that is obtained by a fully fine-tuned ViTPose-S++[41].

**Fine-tuning performance (SA-Q)**
Thus, while the SA-Q shows generally strong zero-shot performance (matching or beating strong supervised baselines), we tested its fine-tuning capacity.

On Horse-10 we show in (Fig. 2b) that using memory replay to fine-tune SA-Q, with only 5% of the training data, we match the ImageNet-transfer-learning baseline with 100% training data (which is a 20X data efficiency gain). In both the low and high data regimes, we significantly outperform ImageNet-transfer-learning baseline (Supplementary Tables S15 and S16).

On iRodent, SA-Q significantly outperforms ImageNet-transfer-learning baseline in both the low and high data regime (Fig. 2c, Supplementary Tables S17 and S18). In particular, using memory replay to fine-tune SA-Q gives 73 mAP with full training data, outperforming ImageNet-transfer-learning by 14.1 mAP, and can be nearly 100X more data efficient in the low data regime (meaning, one needs nearly 354 images to reach the same zero-shot performance and/or fine-tuning with three images).

Next, on AnimalPose we show that our fine-tuning SA-Q gives 87 mAP, slightly better than 86.9 mAP from ImageNet-transfer-learning (Fig. 2d), and using transfer learning gives 89.6 mAP, outperforming ImageNet weights by 2.7 points.

AP-10K is one of the most challenging animal pose estimation benchmarks with many strong baselines (e.g., ViTPose++[41], UniPose[42]) (Fig. 2e, f). SA-Q gives 80.1 mAP after fine-tuning on AP-10K, which is very close to 80.4 obtained by a top-down baseline ViTPose-L++, which uses a vision transformer that is 10X (307 M parameters) bigger than our architecture (HRNet-w32, that has 29M parameters) and it was pre-trained on 307K pose images, which is then 4.38× more than our 70K pose image dataset (Quadruped-80K excluding 10K AP-10K) (Fig. 2f). We also outperform the 79.0 mAP obtained by UniPose-V-Swin-L,

which is a bottom-up method that is pretrained on 226K pose images (plus previously pre-trained on 400M images for using CLIP weights)[37]. Note, Swin-L[43] has 197M parameters, making it 6X larger than the HRNet-w32 we used. Lastly, we are 34.7 points higher than 45.4 mAP reported by another strong bottom-up method, ED-Pose[44], which was pretrained on 154K pose images. Thus, our performance in AP-10K benchmark shows that our approach is not only data-efficient but also parameter-efficient. Taken together, this means that our method's strong performance is not simply due to more data or bigger networks, it is the algorithmic advancements and the animal pose prior from Quadruped-80K.

Collectively, the SuperAnimal method presents an efficient way to achieve strong zero-shot and few-shot performance and also provides better starting weights for fine-tuning (vs. ImageNet-based transfer learning). Of course, despite strong generalization, there can still be failures. Note that both SuperAnimal models—TopViewMouse and Quadruped—learned to predict the union of all keypoints defined in multiple datasets even if no single dataset might have defined all of these keypoints (i.e., as in TopViewMouse-5k), and even if fine-tuned on data without the super-set they still retain the super-set.

**Unsupervised video adaptation**
Independent of the use case (i.e., zero-shot or few-shot fine-tuning), to optimize performance on unseen user data we also developed two unsupervised methods for video inference that help overcome differences in the data SuperAnimal models were trained on compared to what data users might have (Fig. 3a, and Supplementary Fig. S5a). These so-called distribution shifts can come in various forms (e.g., spatial or temporal; see Methods). For example, a bottom-up model can not perform well if the video resolution or animal appearance size is dramatically different from those data which we trained on, and the animal datasets are particularly diverse in size, which can pose challenges (Supplementary Fig. S5b, c). Therefore, inspired by[45], we developed an unsupervised test-time augmentation called spatial-pyramid search that significantly boosted performance in three OOD videos (Supplementary Fig. S5c–e, Supplementary Video 3, Supplementary Table S19; and see Methods). This is unsupervised, as the user does not need to label any data, they simply give a range of video sizes. Note that in practice this does slow down inference time depending on the search parameter space, and this method is not needed with top-

**Table 2 | Main results on quadruped benchmarks**

| Method | Pre-trained weights | Data ratio | mAP | RMSE | Dataset | NE_IID | NE_OOD | Architecture |
|---|---|---|---|---|---|---|---|---|
| Zero-shot | SuperAnimal | – | 68.038 | 12.971 | AP-10K | – | – | HRNetw32 |
| Zero-shot | SuperAnimal | – | 66.110 | 12.849 | AP-10K | – | – | AnimalTokenPose |
| Transfer learning | ImageNet | 1.00 | 70.548 | 11.228 | AP-10K | – | – | HRNetw32 |
| Memory replay | SuperAnimal | 1.00 | 80.113 | 11.296 | AP-10K | – | – | HRNetw32 |
| Zero-shot | AP-10K | – | 79.447 | 5.774 | AnimalPose | – | – | HRNetw32 |
| Zero-shot | SuperAnimal | – | 84.639 | 4.884 | AnimalPose | – | – | HRNetw32 |
| Zero-shot | SuperAnimal | – | 83.043 | 5.154 | AnimalPose | – | – | AnimalTokenPose |
| Transfer learning | ImageNet | 1.00 | 86.864 | 5.757 | AnimalPose | – | – | HRNetw32 |
| Fine-tuning | AP-10K | 1.00 | 86.794 | 4.860 | AnimalPose | – | – | HRNetw32 |
| Memory replay | SuperAnimal | 1.00 | 87.034 | 4.636 | AnimalPose | – | – | HRNetw32 |
| Zero-shot | AP-10K | – | 65.729 | 4.929 | Horse-10 | 0.296 | 0.287 | HRNetw32 |
| Zero-shot | SuperAnimal | – | 71.205 | 3.958 | Horse-10 | 0.227 | 0.228 | HRNetw32 |
| Zero-shot | SuperAnimal | – | 68.977 | 4.081 | Horse-10 | 0.239 | 0.233 | AnimalTokenPose |
| Transfer learning | ImageNet | 0.01 | 0.934 | 46.255 | Horse-10 | 2.369 | 2.36 | HRNetw32 |
| Transfer learning | ImageNet | 1.00 | 90.516 | 1.837 | Horse-10 | 0.036 | 0.135 | HRNetw32 |
| Fine-tuning | AP-10K | 0.01 | 66.284 | 5.029 | Horse-10 | 0.286 | 0.285 | HRNetw32 |
| Fine-tuning | AP-10K | 1.00 | 93.973 | 1.220 | Horse-10 | 0.036 | 0.083 | HRNetw32 |
| Memory replay | SuperAnimal | 0.01 | 73.366 | 3.719 | Horse-10 | 0.209 | 0.202 | HRNetw32 |
| Memory replay | SuperAnimal | 1.00 | 95.165 | 1.153 | Horse-10 | 0.040 | 0.073 | HRNetw32 |
| Zero-shot | AP-10K | – | 40.389 | 37.417 | iRodent | – | – | HRNetw32 |
| Zero-shot | SuperAnimal | – | 58.557 | 33.496 | iRodent | – | – | HRNetw32 |
| Zero-shot | SuperAnimal | – | 55.415 | 34.666 | iRodent | – | – | AnimalTokenPose |
| Transfer learning | AP-10K | 0.01 | 12.910 | 92.649 | iRodent | – | – | HRNetw32 |
| Transfer learning | ImageNet | 0.01 | 0.785 | 152.225 | iRodent | – | – | HRNetw32 |
| Transfer learning | ImageNet | 1.00 | 58.857 | 35.651 | iRodent | – | – | HRNetw32 |
| Fine-tuning | AP-10K | 0.01 | 43.144 | 37.704 | iRodent | – | – | HRNetw32 |
| Fine-tuning | AP-10K | 1.00 | 61.635 | 26.758 | iRodent | – | – | HRNetw32 |
| Memory replay | SuperAnimal | 0.01 | 60.853 | 31.801 | iRodent | – | – | HRNetw32 |
| Memory replay | SuperAnimal | 1.00 | 72.971 | 24.884 | iRodent | – | – | HRNetw32 |

Here, the base SuperAnimal-Quadruped model had none of the held-out datasets. Full results can be found in Fig. 2 for fine-tuning with different amounts of data, but the best fine-tuning performance is shown, which matches the top performance of the SuperAnimal (SA) variant as shown in Fig. 2. Cao et al.[33] do not report a unified single mAP, rather per animal, therefore we trained a model using their dataset to estimate top-line performance if only trained on AP. Number as reported in ref. 41 using the data from ref. 31.

down pose models as top-down detection standardizes the size of the animal in both train and test time before the cropped image is seen by the pose models.

Secondly, to improve temporal video performance we propose a new unsupervised domain adaptation method (Fig. 3a). Others have considered pseudo-labeling for images but they always required access to the full underlying dataset, which is not practical for users[33,46,47]. Our approach is tailored for pose video adaptation without the need for the ground-truth data. The method runs pose inference on the videos and treats the output predictions as the pseudo ground-truth labels and then fine-tunes the model.

First, we used the animal's size (estimated by convex hull formed by animal keypoints, see more details in Methods) as an indicator to measure the improvement in smoothness of video pose predictions. Qualitative performance gain for SA-TVM is shown in Fig. 3b–e.

We also use a jitter score (see Methods) as the indicator to measure whether video adaptation mitigates the jittering that can be seen in pose estimation outputs. Overall, our method had a significant effect on reducing jitter ($F(1, 23286) = 190.03$, $p < 0.0001$; Supplementary Table S20, in all but the dog ($p = 0.36$, $d = −0.03$) and Golden lab ($p = 0.62$, $d = −0.06$) videos; Supplementary Table S21, Fig. 3f–j and Supplementary Video 4).

To quantitatively measure the improvement of video adaptation, we define adaptation gain and robustness gain (see Methods) to evaluate the method's improvement to the adapted video (a subset of

the video dataset) and to the target dataset (all videos in the video dataset). We used Horse-30[16] where 30 videos of horses are densely annotated to evaluate video adaptation (Fig. 3k).

We compare our method to Kalman filtering and so-named self-pacing[33] (see Methods), and find that it significantly improves mAP in terms of video adaptation gain ($p < 0.003$, Cohen's $d > 0.785$) and robustness gain ($p = 0.0001$, Cohen's $d = 3.124$; Fig. 3k; Supplementary Tables S22, S23, S24).

Notably, video adaptation outperforms self-pacing by 4 mAP in terms of robustness gain, demonstrating that it not only adapts to one single video, but to all 30 videos in the dataset. This is important because our method demonstrates successful domain adaptation to the whole video dataset rather than to a single video.

Our method does not take extensive additional time, and practically speaking, can be run during video analysis. For example, if a video (of a given size) can be run at 40 FPS, our video adaptation would slow down processing to approx. 12 FPS, while self-pacing would be closer to 4 FPS (thus slower and less accurate).

**SuperAnimal models can be used with unsupervised behavioral analysis**

To illustrate the value of our zero-shot predictions for behavioral quantification (Fig. 4a), we first turned to an open-source dataset that was used to benchmark the performance of open-source machine learning tools vs. some commercially available solutions[15]. Specifically,

we used the open-field test (OFT) dataset presented in Sturman et al.[15]. We evaluated the performance of SuperAnimal weights in an action segmentation task. To make OFT out-of-distribution, we made a variant of the SA-TVM model by excluding the OFT dataset during training from full SA-TVM.

As a strong baseline, we used the DeepLabCut keypoints trained by Sturman et al., who trained in a supervised way on each video specifically, thus making it in-domain (Fig. 4a, b). We asked if the SuperAnimal model variant, which has never been trained on the 20 videos they present, is sufficient to classify two critical kinematic-based postures: unsupported rearing in the open field, and supported rearing against the box wall (Fig. 4a, b, see also Supplementary Video 5). If the keypoints were too noisy, this task would be very challenging.

In order to transform keypoints into behavioral actions via segmentation, we used skeleton-based features to convert keypoints to feature vectors (see Methods). We then either used only a MLP-based classifier as in Sturman et al., or we used a newly described non-linear clustering algorithm called CEBRA[48] to further improve the feature space, followed by the same classifier (see Methods and Fig. 4c–e).

We found that SA-TVM zero-shot could be as good as the super-vised keypoint model in predicting both behaviors (Fig. 4d–g; linear mixed effect model, fixed effect of "method": $F = 0.999$, $p = 0.393$; Supplementary Table S25). Moreover, using CEBRA slightly improved upon the behavior classification, independent of which keypoints were used (Fig. 4e, f). We also compared the correlation of our result based on SuperAnimal or Sturman keypoint data against the three annotators per video and found that our model is well correlated to the ground truth annotations, particularly when using CEBRA (Fig. 4f).

As a further test, we compared the performance of using our keypoints vs. the officially provided keypoints on the MABe benchmark[49]. In brief, we used the top-down SA-TVM model and ran inference over three million frames over 1830 videos (without video adaptation to test baseline performance). We then used the outputs and ran PointNet[50], which was provided as a baseline method in MABe. Here, we find nearly identical performance on the behavioral classification tasks as the fully-supervised pose estimation data they provide (Supplementary Table S26, Supplementary Fig. S6a). Further suggesting that SA-TVM can be combined with other approaches for mouse behavioral classification.

Moreover, since the time of pre-printing this work[51], SA-TVM has been used zero-shot with post-hoc unsupervised analysis of mouse behavior with Keypoint-MoSeq[52] and (both SA-TVM and SA-Q) within AmadeusGPT[53]. Therefore, collectively this demonstrates that without any training, the SA-TVM model can be used for downstream behavioral analysis on out-of-distribution data.

Lastly, to show the utility of the SA-Q model in video analysis we performed gait analysis in horses. Here, we turn to a ground truth video dataset where every frame of the video was annotated by an equine expert[16]. We computed the stride and swing phase of the gait and showed that the SA-Q model with video adaptation can match ground truth (Fig. 4g–h, and with filtering see Supplementary Fig. S6b) in 24 out of 30 videos, where we only miss one stride detection (either over or under, (Fig. 4h). We also computed the hoof-ground contacts and find generally good agreement between ground truth and predictions ($r = 0.919$; Fig. 4i). The fraction of contacts within 1–5 frames of ground truth was 69.9–81.7%, respectively, averaged across front and hind limbs across all videos. Collectively, this suggests our SuperAnimal models can be used in real-world tasks both in and outside the laboratory.

## Discussion

We propose an approach to create robust, cross-lab neural network models that are applicable for rodents and many other quadrupeds (>45 species). Our approach is general, and it will be an important future goal to expand the DeepLabCut Model Zoo to additional animals (e.g., insects, birds, or fish) and behavioral contexts. moreover, what keypoints are of relevance also depends on the experiment. For instance, in reaching experiments[9,54], different keypoints are of interest than in open field studies, but many groups could still benefit from such collective model-building efforts.

Building a pretrained pose model via supervised learning benefits from the availability of the annotated pose datasets, and we show that our formulation removes the obstacles of leveraging inhomogenous pose datasets, which enables SuperAnimal models to benefit from learning pose prior from larger datasets. Alternatively, unsupervised keypoint discovery can be used[55,56]. While the unsupervised approach requires no pose annotations, the learned keypoints might lack interpretability and it is not clear yet whether it allows zero-shot inference on OOD data. Therefore, both approaches that create predictions based on the super-set of annotated keypoints from different studies and unsupervised keypoint discovery are promising, complementary directions.

Moreover, labs may be more incentivized to share their data knowing their work can be leveraged by a global community effort to build more powerful models. The DeepLabCut Model Zoo web platform allows access to SuperAnimal pre-trained models, aids in collecting and labeling more data (Supplementary Fig. S6d), and hosts other user-shared models at http://modelzoo.deeplabcut.org.

Taken together, we aimed to reduce the (human and computing) resources needed to create or adapt animal pose models in both lab and in-the-wild animal studies, thereby increasing access to critical tools in animal behavior quantification. We developed a new framework called panoptic pose estimation, where models can be used across various environments in a zero-shot manner and if fine-tuned, they require 10–100× less labeled data than previous models (Figs. 1 and 2). They also show increased performance compared to ImageNet transfer learning, plus we demonstrate their ability to be used in downstream tasks such as behavioral classification (Fig. 4), suggesting they could become foundation models for animal pose estimation.

## Methods
### Datasets
We collected publicly available datasets from the community, as well as provided two new datasets for showing how to build models with the SuperAnimal method, iRodent, and MausHaus, as described below. Thereby, we sought to cover diverse individuals, backgrounds, scenarios, and postures. We did not modify the source data otherwise. In the following, we detail the references for those datasets.

**TopViewMouse-5k**. 3CSI, BM, EPM, LDB, OFT See full details at ref. [15] and in ref. [57]. BlackMice See full details at ref. [24]. WhiteMice See details in SIMBA ref. [25]. Courtesy of Prof. Sam Golden and Nastacia Goodwin. TriMouse See full details at ref. [10]. DLC-Openfield See full details at ref. [9]. Kiehn-Lab-Openfield, Swimming, and treadmill See details at ref. [58]. Courtesy of Prof. Ole Kiehn, Dr. Jared Cregg, and Prof. Carmelo Bellardita. MausHaus We collected video data from five single-housed C57BL/6J male and female mice in an extended home cage, carried out in the laboratory of Mackenzie Mathis at Harvard University and also EPFL (temperature of housing was 20–25 °C, humidity 20-50%). Data were recorded at 30Hz with 640 × 480 pixels resolution acquired with White Matter, LLC eV cameras. Annotators localized 26 keypoints across 322 frames sampled from within DeepLabCut using the k-means clustering approach[59]. All experimental procedures for mice were in accordance with the National Institutes of Health Guide for the Care and Use of Laboratory Animals and approved by the Harvard Institutional Animal Care and Use Committee (IACUC) ($n = 1$ mouse) and by the Veterinary Office of the Canton of Geneva (Switzerland; license GE01) ($n = 4$ mice). MausHaus data is banked on zenodo[60].

For ease of use, we packaged these datasets into one directory that can be accessed at https://zenodo.org/records/10618947 [61].

**Quadruped-80K**. AwA-Pose Quadruped dataset, see full details at ref. 62. AnimalPose see full details at ref. 28. AcinoSet see full details at ref. 26. Horse-30 Horse-30 dataset, benchmark task is called Horse-10; see full details at ref. 16. StanfordDogs see full details at refs. 63,64. AP-10K see full details at ref. 31. APT-36K see full details at ref. 32 iRodent We utilized the iNaturalist API functions for scraping observations with the taxon ID of Suborder Myomorpha[65]. The functions allowed us to filter the large amount of observations down to the ones with photos under the CC BY-NC creative license. The most common types of rodents from the collected observations are Muskrat (Ondatra zibethicus), Brown Rat (Rattus norvegicus), House Mouse (Mus musculus), Black Rat (Rattus rattus), Hispid Cotton Rat (Sigmodon hispidus), Meadow Vole (Microtus pennsylvanicus), Bank Vole (Clethrionomys glareolus), Deer Mouse (Peromyscus maniculatus), White-footed Mouse (Peromyscus leucopus), Striped Field Mouse (Apodemus agrarius). We then generated segmentation masks over target animals in the data by processing the media through an algorithm we designed that uses a Mask Region Based Convolutional Neural Networks(Mask R-CNN)[66] model with a ResNet-50-FPN backbone[45], pretrained on the COCO datasets[40]. The processed 443 images were then manually labeled with pose annotations, and bounding boxes were generated by running Mega Detector[67] on the images, which were then manually verified. iRodent data is banked at https://zenodo.org/record/8250392.

For ease of use, we packaged these datasets into one directory, which is banked at: https://zenodo.org/records/10619173 [68].

**Additional OOD Videos.** In Fig. 3, for video testing we additionally used the following data: Golden Lab mouse: see details at ref. 69. Smear Lab Mouse: see details at ref. 70. Mathis Lab MausHaus: New video conditions, but the same as MausHaus ethics approval as above. BlackDog: video from https://www.pexels.com/video/unleashing-the-pet-dog-outdoors-4763071/, Elk video from https://www.pexels.com/video/a-deer-looking-for-food-in-the-ground-covered-with-snow-3195531/. Horse-30 videos: we used the ground truth annotations for 30 horse videos as described in ref. 16.

**Benchmarking: data splits and training ratios.** Pre-training datasets: For every test of an OOD dataset we create a dataset that has all datasets that exclude the OOD dataset. Within the pretraining datasets, we used 100% of the images and annotations, and we used the OOD datasets for performance evaluation.

OOD datasets: For AP-10K, we used the official training and validation set. For AnimalPose, iRodent, and DLC-Openfield, we create our own splits and shuffles. We use the 80:20 train test ratio for AnimalPose and iRodent and we use the 95:5 train test ratio for DLC-Openfield.

Note that in our data release, each leave-one-out dataset is noted in the metadata such that others can easily benchmark their models in the future.

**Panoptic pose estimation**
We cast animal pose estimation as panoptic segmentation[71] on the animal body; i.e., every pixel on the body is potentially a semantically meaningful keypoint that has an individual identity. Ideally, an infinite collection of diverse pose datasets covers this and the union of keypoints that are defined across datasets makes the label space of panoptic pose estimation.

**Data conversion and panoptic vocabulary mapping (generalized data converter).** Data came from multiple sources and in multiple formats. To homogenize different annotation formats (COCO-style, DeepLabCut format, etc.), we implemented a generalized data converter. We parsed more than 20 public datasets and re-formatted them into DeepLabCut projects. Besides data conversion, the generalized data converter also implements key steps for the panoptic animal pose estimation task formulation. These steps include:

1. Hand-crafted conversion mapping. The same anatomical keypoint might be named differently in different datasets, or different anatomical locations might correspond to different labels in different datasets. Thus, the generalized data converter used a hand-crafted conversion mapping (see Supplementary Figs. S1a and S5) to enforce a shared vocabulary among datasets. We checked the visual appearance of keypoints to determine whether two keypoints (in different datasets) should be regarded as identical. In such cases, the model had to learn (possible) dataset-bias in a data-driven way. We can also think of it as a form of data augmentation that randomly shifts the coordinate of keypoints by a small magnitude, which is the case for keypoints which most dataset creators agree on (e.g., keypoints on the face). For keypoints on the body, the quality of the conversion table can be critical for the model to learn a stable representation of animal bodyparts.

2. Vocabulary projection. After the conversion mapping was made, keypoints from various datasets were projected to a super-set keypoint space. Every keypoint became a one-hot vector in the union of keypoint spaces of all datasets. Thereby the animal pose vocabularies were unified.

3. Dataset merging. After annotations were unified into the super-set annotation space, we merged annotations from datasets by concatenating them into a collection of annotation vectors. Note that if the images only displayed a single species, we essentially built a specialized dataset for that species in different cage and camera settings. If there were multiple species present, we essentially grouped them in a species-invariant way to encourage the model to learn species-agnostic keypoint representations, as is the case for our SuperAnimal-Quadruped model.

**The SuperAnimal algorithmic enhancements for training and inference**
**Keypoint gradient masking.** First, we manually verified a semantic mapping of the datasets with diverse naming (i.e., nose in dataset 1 and snout in dataset 2). Then, we defined a master keypoint space naming, where no one dataset needed to have all the names identified. This yielded sparse keypoint annotations into the super-set keypoint space (Supplementary Fig. S1b, c). Training naively on these projected annotations would harm the training stability, as the loss function penalizes undefined keypoints, as if they were not visible (i.e., occluded).

For stable training of our panoptic pose estimation model, we mask components of the loss function across keypoints. The keypoint mask $n_k$ is set to 1 if the keypoint $k$ is present in the annotation of the image and set to 0 if the keypoint is absent. We denote the predicted probability for keypoint $k$ at pixel $(i, j)$ as $p_k(i, j) \in [0, 1)$ and the respective label as $t_k(i, j) \in \{0, 1\}$, and formulate the masked $L_k$ error loss function as

$$\mathcal{L}_{L_k} = \sum_{k=1}^{m} \sum_{i,j} n_k \cdot \| p_k(i,j) - t_k(i,j) \|_z, \tag{1}$$

with $z = 2$ for mean square error and $z = 1$ for L1 loss (e.g., used for locref maps in DLCRNet[10]) and the masked cross-entropy loss function as

$$\mathcal{L}_{CE} = - \sum_{k=1}^{m} \sum_{i,j} n_k t_k(i,j) \log p_k(i,j). \tag{2}$$

Note that we make distinct the difference between not annotated and not defined in the original dataset and we only mask undefined keypoints. This is important as, in the case of sideview animals, "not annotated" could also mean occluded/invisible. Adding masking to not annotated keypoints will encourage the model to assign high likelihood to occluded keypoints.

Also note that the network predictions $p_k(i, j)$ are generated by applying a softmax to the logits $l_k(i, j)$ across all possible keypoints,

including masked ones:

$$p_k(i,j) = \frac{\exp l_k(i,j)}{\sum_{k'=1}^{M} \exp l_{k'}(i,j)}. \qquad (3)$$

M is the total number of keypoints. The masking in the loss function then ensures that probability assigned to non-defined keypoints is neither penalized nor encouraged during training.

**Automatic keypoint matching.** In cases where users want to apply our models to an existing, annotated pose dataset, we recommend to use our keypoint matching algorithm. This step is important because our models define their own vocabulary of keypoints that might differ from the novel pose dataset. To minimize the gap between the model and the dataset, we propose a matching algorithm to minimize the gap between the models' vocabulary and the dataset vocabulary. Thus, we use our model to perform zero-shot inference on the whole dataset. This gives pairs of predictions and ground truth for every image. Then, we cast the matching between models' predictions (2D coordinates) and ground truth as bipartite matching using the Euclidean distance as the cost between pairs of keypoints. We then solve the matching using the Hungarian algorithm. Thus for every image, we end up getting a matching matrix where 1 counts for match and 0 counts for non-matching. Because the models' predictions can be noisy from image to image, we average the aforementioned matching matrix across all the images and perform another bipartite matching, resulting in the final keypoint conversion table between the model and the dataset (example affinity matrices are shown in Supplementary Fig. S2a, b).

Note that the quality of the matching will impact the performance of the model, especially for zero-shot. In the case where, e.g., the annotation *nose* is mistakenly converted to keypoint *tail* and vice versa, the model will have to unlearn the channel that corresponds to nose and tail (see also case study in Mathis et al.[7]). For evaluation metrics such as mAP where a per keypoint sigma is used, we sample the sigmas from the SuperAnimal sigmas (See Supplementary Table S1).

**Memory replay fine tuning.** Catastrophic forgetting[72] describes a classic problem in continual learning[38]. Indeed, a model gradually loses its ability to solve previous tasks after it learns to solve new ones.

Fine-tuning a SuperAnimal models falls into the category of continual learning: the downstream dataset defines potentially different keypoints than those learned by the models. Thus, the models might forget the keypoints they learned and only pick up those defined in the target dataset. Here, retraining with the original dataset and the new one, is not a feasible option as datasets cannot be easily shared and more computational resources would be required.

To counter that, we treat zero-shot inference of the model as a memory buffer that stores knowledge from the original model. When we fine-tune a SuperAnimal model, we replace the model predicted keypoints with the ground-truth annotations, resulting in hybrid learning of old and new knowledge. The quality of the zero-shot predictions can vary and we use the confidence of prediction (0.7) as a threshold to filter out low-confidence predictions. With the threshold set to 1, memory replay fine-tuning becomes naive-fine-tuning.

Memory replay pseudo-code:

```
def is_defined(keypoints):
    # Check whether the original dataset defines each
      keypoint. We use a flag '-1' to denote that a given
      keypoint is not defined in the original dataset.
      Note this is different from not annotated in the
      COCO convention, which use flag '0'
    return True if keypoints[2] >= 0 else False
```

```
def load_pseudo_keypoints(image_ids):
    # get the pseudo keypoints by image IDs.
    # note, pseudo keypoints are loaded from disk and
      fixed throughout the process, so no label drifting
      is expected as in typical online pseudo labeling
    return pseudo_keypoints

def get_confidence(keypoints):
    # get the model confidence of the predicted key-
      points. Unlike ground truth data that have 3
      discrete flags, predicted keypoints have con-
      fidence that can be used as likelihood readout for
      post-inference analysis
    return keypoints[2]

def memory_replay(model, superset_gt_data_loader,
optimizer, threshold):

    # gt data is preprocessed such that annotations
      are now in superset keypoint space.
    # we extended the visibility flag of COCO annota
      tion to following (-1: not defined in the target
      dataset, 0: not labeled, 1: labeled but not
      visible, 2: labeled and visible)

    for batch_data in superset_gt_data_loader:

        gt_keypoints = batch_data['keypoints']
        image_ids = batch_data['image_ids']
        images = batch_data['images']
        # model() is a pytorch style forward function
        preds = model(images)
        pseudo_keypoints = load_pseudo_keypoints(
        image_ids)
        # 3 here is (x, y, flag)
        batch_size, num_kpts, _ = gt_keypoints.shape
        # iterate through batch
        for b_id in batch_size:
            # iterate through keypoints
            for kpt_id in range (num_kpts):
                # since this specific bodypart is not
                  defined in the new dataset, we use saved
                  pseudo labels (zero-shot prediction)
                  as gt. This prevents catastrophic for
                  getting and drifting. We can also use
                  confidence to filter the pseudo
                  keypoints]
                if not is_defined(gt_keypoints[b_id]
                    [kpt_id]) and get_confi-
                        dence(pseudo_key-
                        points[b_id][kpt_id])     >
                        threshold:
                    # we assume a single animal scenario
                      for simplicity. For multiple ani
                      mals, matching between gt and
                      pseudo keypoints need to be done.
                      gt_keypoints[b_id][kpt_id]   =
                      pseudo_keypoints[b_id][kpt_id]

    loss = criterion(preds, gt_keypoints)
    optimizer.zero_grad()
    loss.backward()
    optimizer.step()
```

## Model architectures

For SuperAnimal-TopViewMouse we used both a bottom-up model (DLCRNet) and top-down model (HRNet-w32), or transformer (AnimalTokenPose) (see below). Whereas for SuperAnimal-Quadruped we only use top-down based HRNet-w32. Please refer to the Supplementary Fig. S6 and Supplementary discussion for why we use only top-down models for quadruped.

## Bottom-Up model

**DLCRNet.** The SuperAnimal-TopViewMouse used the bottom-up approach as described in DeepLabCut[9,10]. We use DLCRNet_ms5[10] as the baseline network architecture for its excellent performance on animal pose estimation. A batch size of 8 was used and the SuperAnimal-TopViewMouse was trained for a total of 750k iterations. In the fine-tuning stage, a batch size of 8 was used for 70k iterations. The Adam optimizer[73] was used for all training instances, and we otherwise used default parameters. We follow DeepLabCut's multi-step learning rate scheduler to drop learning rates three times from $1e-4$ to $1e-5$. Cross-entropy is used for learning heatmaps. For fine-tuning experiments, we keep the same optimizer, batch size, and learning rate scheduler. The total number of training steps is adjusted to 70k iterations. During video adaptation, we keep the same optimizer and learning rate scheduler, but with batch size 1 and total training steps as 1000. We observe that the low computational budget as described is sufficient for the model to adapt.

## Top-Down models

**Object detectors.** For the object detectors, we trained Faster R-CNN using ResNet-50 as the backbone[74] and incorporated Feature Pyramid Networks[45] for enhanced feature extraction. The training was conducted over 100 epochs using the AdamW optimizer and LRListScheduler. We initiated the training with a learning rate of 0.0001, which was decreased to $1e-05$ at the 90th epoch. The batch size was set to 4 for both the SuperAnimal-TopviewMice and SuperAnimal-Quadruped.

We processed the TopViewMouse-5K and Quadruped-80K datasets to ensure that there is only one animal category, namely top-view mice or quadrupeds, in each dataset. This approach was adopted to train the model to detect generic animal types effectively. During training, image resizing to $1333 \times 800$ pixels, random flipping, normalization, and padding were applied as part of the data augmentation process.

**HRNet-w32.** HRNet-w32[20] is used for the top-down based SuperAnimal-Quadruped models. The training protocol follows that described in the AP-10K paper[31]. Specifically, we employed the Adam optimizer[73] with an initial learning rate of $5e-4$. The total training duration was set to 210 epochs, with a step decay applied to the learning rate at epochs 170 and 200. A batch size of 64 was used. Consistent with the AP-10K protocol, random flip, half-body transformation, and random scale rotation were applied during training, along with flip testing during evaluation.

For fine-tuning models with a very small number of unique images (e.g., fewer than 64 images in the training set), we fixed the running stats of batch normalization layers and used a smaller initial learning rate of $5e-5$. This setting improves training stability.

HRNet-w32 was also employed for the top-down based SuperAnimal-TopviewMouse models, adhering to the exact same training protocol as the SuperAnimal-Quadruped.

**AnimalTokenPose.** Inspired by recent results of Vision Transformers[21] on human pose estimation tasks[23] we assessed ViT's zero-shot performance. We conducted experiments with the original ViT architecture in three setups: with masked auto-encoder (MAE)[75] initialization, DeiT[76] initialization, and truncated normal initialization with standard deviation 0.02 and 0 mean. Following the original setup[21], we did not use a convolutional backbone. The input image of size $224 \times 224$ was split into patches of $16 \times 16$ pixels, the depth of the transformer encoder was equal to 12 and each attention layer had 12 heads with a feature dimension of 768. It was crucial to use a pre-trained vision transformer; without pre-training, the model did not converge for either dataset (data not shown).

We also adapted the TokenPose model by Yang et al.[22], which adds information about each keypoint in learnable queries called keypoint embeddings. The model was originally used for human pose estimation with a fixed number of keypoints. Combining TokenPose and panoptic animal pose estimation, we obtain AnimalTokenPose models that are able to achieve high zero-shot performance in OOD datasets we prepared (Figs. 1 and 2).

For keypoint estimation, 12 transformer encoder blocks with feature vector of size 192 were stacked. While the ViT encoder received raw pixels as an input, in TokenPose[22] the images of size $256 \times 256$ are first processed by a convolutional backbone, and captured abstract features are then split into patches of size $4 \times 4$. As in TokenPose[22], we used the first three stages of HRNet[77] and 2 stacked residual blocks from a ResNet[78].

The training procedure for AnimalTokenPose is identical to HRNet-w32 detailed above.

## Video inference methods and considerations

**Domain shifts and unsupervised adaptation.** These domain shifts[79] describe a classical vulnerability of neural networks, where a model takes inputs from a data domain that is dissimilar from the training data domain, which usually leads to large performance degradation. We empirically observe three types of domain shifts when applying our models in a zero-shot manner. These domain shifts range from pixel statistics shift[80], to spatial shift[81], to semantic shift[79,80]. To mitigate those, we applied two methods, test time spatial-pyramid search and video adaptation.

**Handling the train and test time resolution discrepancy for bottom-up models.** One notable challenge for our bottom-up models face at inference time is the discrepancy in the animal appearance sizes and image resolutions between train and test stages. Even though scale jitter augmentation is part of most pose estimation frameworks' data augmentation pipeline, including DeepLabCut's[10,59,82], the model can still have trouble handling dramatic changes in the image resolution or the animal appearance sizes. To further deal with scale changes, we employ spatial-pyramid search at test-time (see below). The same challenge happens in fine-tuning stage. The downstream dataset (and the animals present in it) could have very different animal sizes from the pre-training datasets, causing a distribution shift to the pre-trained models. We thus apply resizing (height 400 pixels and same aspect ratio) to downstream datasets if their sizes are drastically different from our training images.

**Test time spatial-pyramid search for bottom-up models.** As bottom-up models do not enforce the standardization of the animal size seen by the pose estimator, the relative animal size (ratio between the animal's bounding box area and the area of the image) seen in the pre-training stage and testing stage can be very different. In other words, the bottom-up model performs best with the animal sizes seen in the training stage. The relative animal size in the test time is unknown and as a result, it can cause performance degradation due to spatial distribution shift. We propose to apply multiple rescaling factors to the test image and aggregate the models' predictions.

Therefore, during the inference, we build a spatial-pyramid composed of model's predictions for multiple copies of the original image

at different resolutions. We use model's confidence as the criterion to filter out the resolutions that give sub-optimal performance and aggregate (taking median) predictions from resolutions that have above-threshold confidence as our final prediction.

The train-test resolution discrepancy[83] has been studied actively and most approach it through multi-resolution fusion[10,45,77]. Previous work mostly focuses on IID settings where the resolution of testing images did not vary considerably from the training images. Moreover, prior work approaches multi-resolution fusion via deep features, requiring modifications of the architecture and adding more parameters. In contrast, the proposed spatial-pyramid search is designed to aid SuperAnimal models in zero-shot scenarios where the resolutions of testing images are most likely out of distribution to our training images. We did not apply multi-resolution fusion via deep features for that requires fixing choice of architectures. On the other hand, commonly used multi-scale testing in IID setting does not need to carefully filter out very noisy predictions. This method can also be used for calibration to find the optimal scale.

Spatial-pyramid pseudo-code:

```
def spatial_pyramid_search(images, model, scale_
    list, confidence_threshold, cosine_threshold):

    # generate rescaled version of original images
    with multiplescaling factor
    rescaled_images   =   rescale_images(images,
    scale_list)
    preds_per_scale = []
    # gather predictions of the model, assuming the
    final pred_keypoints are projected to the original
    image space by the forward function

    for rescaled_image in rescaled_images:

        pred_keypoints = model(rescaled_image)
        # using median to get a good estimate of expec
        ted keypoint positions
        median_keypoint    =    get_median_key
        point(pred_keypoints)
        # If the rescaled image is not suitable for the
        model, we expect the model have a confidence
        less than a given threshold
        pred_keypoints = filter_by_confidence(pred_
        keypoints, confidence_threshold)
        # A median filter alone does not remove out
        liers. After confidence filtering, we compare
        the remained predictions to the median key
        point and drop the low quality predictions
        pred_keypoints = filter_by_cosine_similar
        ity(pred_keypoints,       median_keypoint,
        cosine_threshold)
        preds_per_scale.append(pred_keypoints)

    return get_median_keypoints(preds_per_scale)
```

**Video adaptation.** To aid SuperAnimal models to adapt to novel videos, we inference the model on the videos, and treat these predictions as the pseudo ground-truth[84] labels to train on. We remove the predictions that have low confidence to filter out unreliable predictions. We found that it is critical to fix the running stats of batch normalization layers during the adaptation training (See supp for more details). Empirically, 1000 iterations with batch size 1 is sufficient to greatly reduce the jitter. The optimal number of iterations and the confidence threshold are hyperparameters for different videos.

Video adaptation pseudo-code:

```
def get_pseudo_predictions(frame_id):
    # return pseudo prediction by frame id

def video_adaptation(model,      video_data_loader,
    optimizer, threshold):
    for data in video_data_loader:
        # fix the running stats of BN layers
        model.eval()
        frame_id = data['frame_id']
        Image = data['image']
        pseudo_keypoints    =    get_pseudo_pre
        dictions(frame_id)
        preds = model(image)
        # predictions that have low confidence are
           masked from loss calculation.
        loss = criterion(preds, pseudo_keypoints,
        mask_by_threshold = threshold)
        optimizer.zero_grad()
        loss.backward()
        optimizer.step()
```

## Evaluation metrics
### Supervised metrics for pose estimation
**RMSE.** Root Mean Squared Error is a metric to measure the distance between prediction and ground truth annotations in pixel space[7,9]. However, for pose estimation, does not take the scale of the image and individuals into consideration and the distance is thus non-normalized. As our data is highly variable, we also sometimes use normalized errors. We use RMSE for the DLC-Openfield benchmarking, as this was the original authors' main reported metric. Note that during evaluating RMSE, we do not remove predictions that have low confidence due to occlusion. Therefore, all predictions including outliers are penalized by RMSE.

**Normalized error.** For Horse-10 experiments, we compute RMSE between ground-truth annotations and predictions with confidence cutoff 0 (to include all predictions to ensure low confidence predictions are also penalized). The resulting RMSE is then normalized by the eye-to-nose GT distance provided by ref. 16.

**mAP.** Mean average precision (mAP) is the averaged precision of object keypoint similarity (OKS)[85]:

$$OKS = \frac{\sum_{i=1}^{n}\left[\exp\left(-d_i^2/2s^2k_i^2\right)\delta(v_i>0)\right]}{\sum_{i=1}^{n}\left[\delta(v_i>0)\right]} \quad (4)$$

where $d_i$ is the Euclidean distance between each corresponding ground truth and detected keypoint and $v_i$ is the visibility flags of the ground truth, $s$ is the object scale and $k_i$ is a per keypoint constant that controls falloff (see full implementation details at ref. 40). For lab mice, we used 0.1 for all keypoints following[10]. For quadruped, we used the sigmas (per keypoint constant) of the 17 keypoints shared with AP-10K[31] and used 0.067 for the rest of animal keypoints (see below). $s$ is the square root of the bounding box area (product of width $X$ height of the bounding box).

The body parts along with their corresponding $k$ in pixels are: nose (0.026), upper_jaw (0.067), lower_jaw (0.067), mouth_end_right (0.067), mouth_end_left (0.067), right_eye (0.025), right_earbase (0.067), right_earend (0.067), right_antler_base (0.067), right_antler_end (0.067), left_eye (0.025), left_earbase (0.067), left_earend (0.067), left_antler_base (0.067), left_antler_end (0.067), neck_base (0.035), neck_end (0.067), throat_base (0.067), throat_end (0.067), back_base (0.067), back_end (0.067), back_middle (0.035), tail_base

(0.067), tail_end (0.079), front_left_thai (0.072), front_left_knee (0.062), front_left_paw (0.079), front_right_thigh (0.072), front_right_knee (0.062), front_right_paw (0.089), back_left_paw (0.107), back_left_thigh (0.107), back_right_thai (0.087), back_left_knee (0.087), back_right_knee (0.089), back_right_paw (0.067), belly_bottom (0.067), body_middle_right (0.067), body_middle_left (0.067).

## Unsupervised metrics for video prediction smoothness

**Convex hull body area measurement.** To evaluate the smoothness of SuperAnimal model predictions in video, we utilize a simple unsupervised heuristic. It computes the area of a polygon encompassing all keypoints, the idea being that the smoother the detections, the lower the variance of this polygon's area. This is formally noted by $A_{body}$, to estimate the animal body area. $A_{body}$ is calculated using the convex hull containing all keypoints over time. Let $\mathcal{K}$ represent the set of all keypoints for the animal at each time step, and $H(\mathcal{K})$ denote the convex hull containing all keypoints. The animal body area, $A_{body}$, is then given by the area of the convex hull:

$$A_{body} = Area(H(\mathcal{K})) \quad (5)$$

where $Area(H(\mathcal{K}))$ is the function that calculates the area of the convex hull $H(\mathcal{K})$ containing all keypoints over time.

**Jittering metric.** We define jittering, denoted by $J$, as the average of the absolute values of centered, non-signed speeds across all examples and all keypoints. For a given keypoint $k$ and example $e$, the jittering value is computed as follows:

$$J_{k,e} = \frac{1}{N_{k,e}} \sum_{i=1}^{N_{k,e}} |v_{k,e,i}| \quad (6)$$

where: $J_{k,e}$ is the jittering value for keypoint $k$ in example $e$; $N_{k,e}$ is the total number of centered, non-signed speed measurements for keypoint $k$ in example $e$; $v_{k,e,i}$ is the $i$-th centered, non-signed speed measurement for keypoint $k$ in example $e$.

**Keypoint dropping metric.** Keypoint drop is a count of the number of keypoints with predicted likelihood below a set threshold for every predicted frame (the cutoff was set to 0.1 for bottom-up models, and 0.05 for top-down models). In practice, low-confidence predictions are dropped to remove predictions that are not reliable or occluded.

In this work, keypoint dropping is used to complement metrics such as RMSE to evaluate the jittery of predictions or catastrophic forgetting. Note keypoint dropping is only valid for top-view, openfield (almost no occlusion) videos where every keypoint is supposed to be predicted with relatively high confidence. For side-view poses, keypoint dropping is not suitable as many bodyparts are occluded.

Let $K_{total}$ be the total number of keypoints in the video sequence, and $K_{dropped}$ be the count of keypoints that are below a defined threshold $T_{threshold}$ and considered for dropping in environments with little occlusion and a top view.

$$K_{dropped}(t) = \sum_{i=1}^{K_{total}} \delta_i(t) \quad (7)$$

where $K_{dropped}(t)$ is the count of keypoints dropped at time $t$, and $\delta_i(t)$ is an indicator function that returns 1 if the $i$-th keypoint is below the threshold at time $t$, and 0 otherwise:

$$\delta_i(t) = \begin{cases} 1, & \text{if } score_i(t) < T_{threshold} \\ 0, & \text{otherwise} \end{cases} \quad (8)$$

where $score_i(t)$ is the confidence score or measurement of the $i$-th keypoint at time $t$.

**Adaptation gain (or loss) in mAP.** Denotes the adapted model's change in mAP on the adapted video. A negative number means a performance degradation after adaptation.

Every video in Horse-30 dataset is densely annotated. Thus we can calculate the mAP gain on the video after the model is adapted to it. We use the pre-adapted zero-shot mAP as the reference and calculate the difference between the post-adaptation mAP and pre-adaptation mAP.

**Robustness gain (or loss) in mAP.** Calculates mAP gain on all videos from the same dataset. This helps to identify whether the model overfits one single video it trains on or it performs successful domain adaptation with respect to the whole video dataset. We use this robustness gain to complement adaptation gain. We calculate the mAP for the adapted models on all 30 videos of Horse-30 dataset[16]. A positive gain in robustness also suggests that the method can be used on one video and benefit all other videos in the same dataset.

**Video adaptation compared to baselines using supervised metrics (mAP).** We use HRNet-w32 with the detector we trained to perform video adaptation to inference the videos to obtain pseudo-labels.

For video adaptation algorithm, the prediction confidence threshold is set to 0.5 and we perform video adaptation for 4 epochs for each video it adapts to. The learning rate scheduler and augmentations are identical to HRNet-w32's.

**PPLO.** Progressive Pseudo-label-based Optimization[33] implements iterative pseudo-labeling that follows a curriculum, namely, the pseudo-labeling starts with high confidence prediction, and then trains with small confidence predictions, following an easy-to-hard curriculum. We initialize three confidence intervals as [0.9, 0.7, 0.5] and sequentially apply pseudo-labeling to the model for four epochs with each confidence level, making a total of 12 epochs training with PPLO.

The full algorithm of PPLO also requires training on both labeled source data and labeled target data, which the video adaptation does not do. For fairness reasons, we only performed the iterative pseudo-labeling step.

**Kalman filtering.** We apply a constant-velocity Kalman filter (implemented in `filterpy` v1.4.5) as post-processing to our pre-adaptation zero-shot pose predictions. As Kalman filtering does not modify the model weights, we do not report the general robustness gain on it.

**Statistical analysis.** Linear mixed-effects models were fitted in R[86] using the `lme4` package (v1.1.31;)[87]. Training data ratio (or, equivalently, the number of images) and fine-tuning methods were defined as fixed effects, whereas the various datasets and shuffles were treated as random effects; random intercepts and slopes were also added at the dataset level. The best models were selected based on the Akaike Information Criterion (AIC); adding complexity did not result in lower AIC, and even led to singular fits, indicative of overfitting. The weight of evidence for an effect was computed using likelihood ratio tests, as well as with $p$ values provided by `lmerTest` (v.3.1.3). Two-sided pairwise contrasts and Cohen's $d$ standardized effect sizes were computed with the `emmeans` package (v.1.8.9), and degrees of freedom estimated with the Kenward–Roger method. Distributions of prediction errors with and without spatial-pyramid search were compared with the two-sample, one-sided (alternative hypothesis: "less") Kolmogorov–Smirnov test. The significance threshold was set at 0.05.

## Behavioral action segmentation, OFT

As our benchmark dataset, we used the openfield test (OFT) task from Sturman et al.[15]. We calculated the same skeleton-based features by concatenating 10 distances between keypoints, six angles, four body areas, and two additional boolean variables coding whether the nose

and head center was inside the arena, resulting in a 22D vector at each time step. For the action classifier, we used an MLP neural network as the action decoder that acted as a sliding window across 31 time steps to perform action segmentation and used F1 score on supported and unsupported rears as evaluation metrics. As in the original paper, we performed leave-one-out cross-validation on 20 videos and across three annotators.

Note that the original model for OFT task from Sturman et al. includes the center and four corners of the mouse cage, which is critical for their handcrafted features to determine the relative distance between the mouse and the walls. As our SuperAnimal models focus on animal bodyparts only, we take the corner coordinates from their released data for the sake of comparison. In practice, those static environmental keypoints can be provided by taking users' inputs via interactive GUI for videos.

For CEBRA[48], we used the model architecture "offset10-model". The output dimension was set to 32, as found via a simple grid search over the following values: [4, 8, 16, 32]. We trained it for 5000 iterations with batch size 4096, the Adam optimizer, and learning rate 1e−4.

### Behavioral action segmentation, MABe
MABe has two rounds and since only round 2 released videos, we use videos from round 2 as the inputs for our pretrained SA-TVM model. Since our paper focuses on pretrained pose model, we use recommended baselines[49,50] from round 1 that build representation based on pose trajectories instead of RGB-based representation learning baselines (as RGB-based representation learning is known to be better than pose trajectory-based representation[88]. Videos from MABe round 2 have three mice in videos, therefore we used our top-down version SA-TVM. The procedure is as follows: we inferenced our pretrained top-down SA-TVM on all 1830 videos from round 2, converted the pose results into MABe keypoint file format, and ran PointNet code to obtain embeddings. Finally, we use the official evaluation code to compare the performance between using the official MABe poses obtained from fully supervised learning and poses that are obtained via our models' zero-shot predictions.

### Behavioral action segmentation, Horse Gait Analysis
Our SA-Q model was run on the videos from Horse-30[16]. The start (2 s) and end (2 s) of each of the 30 videos were removed from the analysis, to ignore instants when the horse is only partially seen. Front and back hoof contacts and lifts were identified using respectively peak and valley detection from the 2D kinematic traces of the front and back hooves. Beforehand, these trajectories were smoothed using a 2nd-order, low-pass, zero-lag Butterworth filter (cutoff=3 Hz) and centered on a keypoint located on the animal's back; this effectively expresses keypoint coordinates in a reference frame stationary relative to the moving horse, facilitating event detection. We extracted fore and hind limb strides between consecutive ground contacts, and stance phases between a contact of one hoof until it is lifted off the ground. Stride lengths (in pixels), stances, and the number of identified hoof contacts were then computed, and qualitatively compared to those obtained using the densely annotated (ground truth) keypoints (Fig. 4g, h, i).

### Code API
High-level inference API (with optional spatial-pyramid search) for using SuperAnimal models in DeepLabCut:

```
video_path = 'demo-video.mp4'
superanimal_name = 'superanimal_topviewmouse'
scale_list = range(200, 600, 50) # image height pixel size
range and increment
```

```
deeplabcut.video_inference_superanimal(
    [video_path],
    superanimal_name,
    scale_list=scale_list,
    video_adapt=True)
```

### Web App
Many labs use DeepLabCut to define, annotate, and refine animal bodyparts, resulting in high quality, diverse keypoint annotations for animals in different contexts[10,59]. In order to enable a positive feedback loop to turn the collection of animal pose data and models into a community effort we developed a Web App.

The app is available at https://contrib.deeplabcut.org/. This app allows anyone, within their browser, to a) upload their own image and label, b) annotate community images, c) run inference of available community models on their own data, and d) share models to be hosted. The website is written using JavaScript with the Svelte framework, and the models are run on cloud servers.

**Data collection.** The website has an upload portal for groups to upload their models and labeled data in DeepLabCut format to help grow the pre-training datasets and allow researchers to build on top of varied models and data.

**Annotation.** Additionally, the website hosts a labeling web app that allows users to annotate curated images. The datasets currently available for annotation are from iNaturalist[89] and the OpenImage Dataset[90]. After selecting which dataset to label, images are displayed successively with the target animal prominently shown in front of an opaque masked background (which can be toggled off). The keypoint set is selected taking into account the species morphology and keypoint value in subsequent analysis. Once the annotation is complete, the data is saved to the database and made available for use in further research.

**Online inference.** To allow testing DeepLabCut models in the browser, the user selects a few images, which model to run, and receives predictions along with confidence scores for each keypoint. Users are then able to adjust or delete keypoints, as well as download the model weights from HuggingFace. This allows for a quick and hassle-free evaluation of DeepLabCut's capabilities and suitability for specific tasks, making it available to a wider range of users.

### Reporting summary
Further information on research design is available in the Nature Portfolio Reporting Summary linked to this article.

## Data availability
The Two SuperAnimal datasets generated in this study have been deposited in the Zenodo database under DOIs[61,68]. These packaged datasets are detailed in the "Datasheets" in Supplementary Information. The SuperAnimal model weights are banked at HuggingFace: https://huggingface.co/mwmathis/DeepLabCutModelZoo-SuperAnimal-Quadruped and https://huggingface.co/mwmathis/DeepLabCutModelZoo-SuperAnimal-TopViewMouse, and see the detailed "Model Cards" in Supplementary Information.

## Code availability
Code to use the DeepLabCut Model Zoo: https://github.com/DeepLabCut/DeepLabCut; it is available since version 2.3.1 and at the time of final acceptance the code is at version 2.3.9. Code and data to reproduce the figures: https://github.com/AdaptiveMotorControlLab/modelzoo-figures. All other requests should be made to the corresponding author.

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

## Acknowledgements

The authors thank Niels Poulsen, Mu Zhou, Lucas Stoffl, and Shilong Zhang for code assistance, or discussions and feedback. We also thank Ole Kiehn, Jared Cregg, Carmelo Bellardita, Johannes Bohacek,

Matt Smear, Sam Golden, and Nastacia Goodwin for generously sharing data on OpenBehavior or zenodo.org. Funding was provided by the EPFL, CZI grant DAF2020-207363 from the Chan Zuckerberg Initiative DAF, an advised fund of Silicon Valley Community Foundation, The Vallee Foundation, NIH BRAIN 1UF1NS126566-01, a Novartis Foundation for Medical-Biological Research Young Investigator Grant. M.W.M. is the Bertarelli Foundation Chair of Integrative Neuroscience.

## Author contributions

Conceptualization: M.W.M., A.M., S.Y.; Methodology: S.Y., A.F., J.L., St.S., M.W.M., A.M.; Software: S.Y., A.F., J.L., M.V., St.S., A.M., M.W.M.; Investigation: S.Y., A.F., J.L., A.M., M.W.M.; Dataset Curation: M.W.M., A.M., S.Y., T.Q.; MausHaus data collection and labeling: M.W.M.; iNaturalist data curation and labeling: T.Q.; WebApp: St.S., M.V.; Writing-Original Draft: M.W.M., S.Y.; Writing-Editing: M.W.M., A.M., S.Y., M.V., St.S., J.L., A.F.

## Competing interests

M.W.M. and St.S. are co-founders at Kinematik AI. J.L. is a co-founder at Fxy AI. The other authors declare no conflicts of interest. The funders had no role in the conceptualization, design, data collection, analysis, decision to publish, or preparation of the manuscript.
