## [Peer Review File · Nature Communications]

SuperAnimal pretrained pose estimation models for behavioral analysisEditorial Note: Parts of this Peer Review File have been redacted as indicated to remove third-party material where no permission to publish could be obtained.

REVIEWER COMMENTS

Reviewer #1 (Remarks to the Author):

Ye and colleagues introduce SuperAnimal, a series of innovations, which ultimately provide two pretrained pose models that allow users to generate pose estimation tracking without the need to label frames. They demonstrate that the models outperform the current state-of-the-art algorithms on out-of-distribution videos. Even if manual labeling is required for fine-tuning, SuperAnimal boosts performance and saves time. The developers cleverly envision a system that can be updated and improved by the users, as it can handle and combine datasets that are not identically labeled across labs. SuperAnimal addresses a very important challenge in the field, as it allows to generate richer pose-estimation datasets that can be compared across labs, while also reducing time required for labelling frames. SuperAnimal is an exciting new step forward and we recommend this manuscript for publication.

During the review process, we have validated that the “Top-View-Mouse” model works well on out-of-frame videos from our lab (particularly if those videos are similar to those in the training set). After testing SuperAnimal, we propose minor comments to be considered:

Minor comments

General:

- The title should be toned down. The term "plug-and-play" has been overused in marketing and advertising, leading to unrealistic expectations and frustration for consumers. Despite the user-friendly interfaces, DLC is not truly plug-and-play (see also comments below)
- The Github repository (<https://github.com/AdaptiveMotorControlLab/modelzoo-figures>), where the code and data should be available to reproduce the figures does not seem to be up to date. For instance, if the reader wants to check how the example images were selected for Figure 1h or 1i (see also comment below), there is no code available. Please update the files.
- The authors should comment on whether SuperAnimal can also run on multiple animals (e.g. 2 or more mice in one arena) or not. I assume it does not (yet), which is not a problem, but should be clarified.
- It would be nice to be able to easily select which points the SuperAnimal networks label. This would allow specific points to be tracked, perhaps even using the labels the initially trained networks used?

This would make it easier to integrate with existing post processing pipelines. It might also be nice to have a clearer diagram of how to label the points for refinement. It seems several points on the body or tail are a kind of spatial division rather than being located on a feature that is easy to define such as the nose, eye, tail-base etc.

- The installation/use of SuperAnimal could still be improved, here are some suggestions:

- a. a complete beginners guide to installing Deeplabcut ready to use modelzoo would be helpful, with all commands and installs. For instance, to get the supplied conda environment for deeplabcut they suggest git cloning it but don't tell the user to first install git. Small things like this can be big hurdles for labs that have little experience installing non-commercial software packages (see plug-and-play comment above).

- b. highlight the fact that you need to use colab to get access to the superanimal_quadruped and superanimal_topviewmouse networks (unless it can be found elsewhere already, if so please clarify, we looked but couldn't find it). Although colab worked nicely, lots of users will want to move it to their local machines and not link their google accounts with colab

- c. Try to explain the scale_list function in more detail, maybe give an example of e.g. if you have a 1024x768 video set the function to x? or tell users where they can find out what their image height pixel size range and increment are.

The figures would benefit from some adjustments:

- Figure 1b-d: It is hard to figure out which parts belong to which subfigure (particularly 1c vs 1d). Could the authors increase the spacing between them to make it clearer?

- Figure 1h: It was very hard to recognize the keypoints on the mouse on some of the IID pictures, especially if someone wants to compare between predicted points and ground truth. It would be nice if the authors find a way to improve visibility (e.g. by choosing different colors for the keypoints, or maybe simply providing a higher-resolution/bigger image in supplementary data)

- Figure 1h-i: The authors should state (e.g. in the figure legend) how these example images were selected. Are these the ones where SuperAnimal performed "best" (based for instance on visual inspection by the authors) or were they selected randomly?

- Figure 2e: Could the authors add information about the time window (or frame number) these plots were created for, either by adding it to the plot itself or the figure legend?

- Figure 2h: The chosen colors make it hard to distinguish between the different models.

- Figure 3a: The meaning of the icons on the right is not very clear, please add short labels/text

Typo:

- On page 2: "...the pre-trained encoded..." should be "encoder..."

Reviewer #2 (Remarks to the Author):

The authors introduce SuperAnimal models, a project aimed at creating setting-invariant pose estimation models for animals, akin to similar models capable of estimating human pose in diverse settings. To create these models, they assemble an annotated dataset of pose keypoints from 45 species of mammals, consisting of over 45,000 images (for comparison, the widely used MPII Human Pose dataset consists of 25,000 human pose annotations, while COCO contains more than 250,000 human pose annotations.) They present model performance in zero-shot pose estimation and few-shot pose estimation, showing improvements over previous DLC models. They then evaluate the zero-shot version of their TopViewMouse model in two contexts: for tracking of animals in videos, and for detection of rearing behavior. While the zero-shot performance of the TopViewMouse model seems reasonable, the performance of the Quadrupeds model seems so low that I doubt it will be useable for any form of behavioral analysis. I also had some concerns with how models are evaluated, with how representative their behavior analysis results are, and with the design of the “video adaptation” method used to reduce keypoint jitter. Overall, these concerns are quite significant, and I feel must be addressed before the paper can be considered for publication.

My major concerns are as follows:

The authors present both a top-view mouse SuperAnimal model and a quadruped model. While the utility of the mouse model for the study of behavior is examined in Figures 2-3, the quadruped model is only briefly explored in Figure 1 and Figure S3. The quadruped pose estimates in Figure 1 are better than chance, but they are often very incomplete (eg wolf, tiger, sheep, raccoon) and I’m skeptical that they will be of any practical use for behavior analysis. This paper only demonstrates its titular claim of “plug-and-play analysis of animal *behavior*” in the case of top-view videos of mice in a laboratory setting (Figures 2-3). To present the SuperAnimal method as facilitating animal behavior analysis beyond the specific case of lab mice is premature and an overstatement of the results that are shown.

Quantitatively, the quadruped model is reported to have a zero-shot performance of ~ 1.1 normalized units on Horse-10, where 1 = the distance from a horse’s nose to its eye. My guess is that this would work out to a bit over a foot; I have trouble visualizing any application where a model with an *average* keypoint error of more than 12 inches is going to be useable. The normalized error on iRodent is around 0.6, where 1 = the square root of the bounding box area. Assuming a roughly square bounding box, this equates to 1 being on the order of the body length of the animal; an average keypoint error of ~ 0.6 body

lengths is again not likely to be useable. Given these very high error rates, the claim that the SuperAnimal-Quadruped model is a zero-shot learner seems to be not supported by the data, and should be removed.

Some concerns regarding how choice of metrics for reporting performance:

a. The “normalized error” plots in Figure 1F are difficult to interpret – error is reported in units of either eye-to-nose distance (for horses) or the square root of the bounding box area (for rodents). This normalization is not used anywhere else in the computer vision, and the bounding box metric in particular is not a good choice, as simple changes like rotation of the animal will change the area of the bounding box substantially. (To see this, picture an elongated object like a running mouse- when oriented horizontally or vertically, its bounding box will be a long rectangle, with length L and width $W < L$, giving it an area of $W * L$. When oriented diagonally, its bounding box will be a square with diagonal length of roughly L (actually slightly longer), giving it an area of $1/2 L * L$.)

b. Performance is also difficult to evaluate as the SuperAnimal model is only compared to itself (in various configurations), and to ImageNet (which is a poor comparison as it was not developed for pose estimation.) Although the mAP of the object-keypoint similarity is reported in Supplemental Figure S3, the authors do not explain in the methods what values of s and k were used in this calculation. Typically, s is the ratio of the bounding box to the image size, however k is a body-part specific value that is estimated by computing the variance of human-generated labels of that part within an image (see Ronchi and Perona 2017, which introduced this metric.) While field-consensus values of k exist for human body parts, there are no such values for animal body parts. To use the OKS metric, the authors would need to either select a k based on values from the human literature, or preferably, obtain an estimate of the variance of manual annotations of each body part, typically by having multiple individuals annotate a representative set of images.

The methods section on the test time spatial-pyramid search is a little difficult to follow, and I’m not 100% sure how the method works. It seems somewhat similar to the hourglass model for pose estimation, and it would help if the authors could explain the relationship between the two. This methods section also seems to confound two problems: 1) difference in the appearance size of the animal across datasets, and 2) difference in the image resolution across datasets. The former is a known challenge in computer vision, whereas the latter (which is emphasized in the methods section, particularly in the last paragraph) seems like it could be trivially resolved by simply rescaling the input image during inference.

The video adaptation method uses model-predicted keypoint locations as a source of training data to the model itself. I can see how this improves model prediction when quantified in terms of keypoint jitter, given that for each new frame T , the model has a labeled frame $T-1$ in its training set and simply has to recapitulate the label placements of that training example. My concern, however, lies in the problem of catastrophic forgetting- nothing is constraining the model-predicted keypoint locations used for training to be accurate, and it seems quite possible that there could be drift in their definitions over time, as

repeated iterations of fine-tuning lead to forgetting of the original human-generated training set. If the authors want to present this method as a resource for improving accuracy of pose estimation, it should be evaluated not in terms of keypoint jitter (which would not detect such catastrophic forgetting of keypoint definitions), but in terms of metrics like OKS or RMSE, as used in Figure 1.

The paper includes multiple model architectures and modifications (CNNs vs transformers, with and without spatial-pyramid search and video adaptation). It would be very helpful for the authors to include a table explaining which model configuration was used for each result reported in the paper.

The difference in performance in classifying supported rears vs unsupported rears is striking- while detection of supported rears seems useable, the performance on unsupported rears is quite low. I suspect that this is because supported rears can only take place at the walls of the arena, therefore the classifier could detect them just by learning to recognize when the mouse is near the walls of the arena, without incorporating any postural information. It would be helpful to see evaluation on a wider range of behaviors; I strongly suspect that the performance seen in the unsupported rearing condition is more likely to be typical for detecting behaviors that are not dependent on positional information.

Other/minor points

It is not clear how the RMSE metric accounts for missed keypoints (points that are incorrectly reported as occluded). This should be added to the Methods. Or, if missed keypoints do not contribute to the RMSE, then the authors might want to use a different metric that incorporates this form of error.

The citation for OKS points to the original Microsoft COCO paper, which predates the creation of the OKS metric by a few years. While OKS is now a part of the COCO API, it was only added later and is not discussed in the original manuscript. Ronchi and Perona 2017 introduces OKS and explains it in detail.

IID usually means “independent and identically distributed”, making its use as an abbreviation for “within-distribution datasets” a little confusing.

Figure 1f should include units for the RMSE on DLC-Openfield (probably pixels?)

“We outperformed DeepLabCut-ResNet-50 ... by over 2X with 10X less data” – this is worded as if SuperAnimal + 10 frames has 2x the performance of DLC + 100 frames, while in fact DLC + 100 is slightly

better than SuperAnimal + 10. You could write “We outperformed DLC by 2X” or “We matched DLC performance with 10X less data”, but you can’t claim the combination of the two. (A car that gets 60 mpg can drive twice as far per gallon of gas as a car with 30 mpg, and it takes half as much gas to travel the same distance- but it can’t drive twice as far AND burn half as much gas in doing so.)

Some figure text is very small and hard to read.

Figure 2h, bar colors are very similar/the same for multiple models, please use a different color for each model tested.

Reviewer #3 (Remarks to the Author):

Review of Ye et al.

In this manuscript, Ye and colleagues develop new neural network models they call “SuperAnimal” that are pre-trained on a variety of relevant behavioral imaging data (for example, videos of mice in different behavioral chambers/settings), and then can be used “out of the box” with no training or “zero shot” on new behavioral data (in different chambers/settings). While the performance of the model already appears to be good, they then develop a framework to improve performance through what they call fine-tuning (labeling a small amount of data in the new chamber/setting and using SuperAnimal pre-trained weights). The concepts here are very exciting - having a model that can be used without the burden of labeling and training on any new behavioral dataset (independent of different lighting conditions, visual appearance of the behavioral chamber and animals, etc.) would be very impactful for both behavioral and neuroscience research, particularly for mice (what is primarily demonstrated here). Despite my excitement for the method itself, the authors do not provide sufficient details for me to evaluate the model and its performance. This is surprising since best practices are well established for such models - I cite several of these below which might serve as examples for the authors. In general, the authors highlight an exciting new method, but without providing the details to support whether the model performs to match their claims - this could be fixed by the authors following some of the suggestions below.

Methods:

Memory replay is a core method employed here to prevent catastrophic forgetting, but very little detail is provided on how this is implemented.

What, precisely, is meant by "When we fine-tune a SuperAnimal model, we replace the model predicted keypoints with the ground-truth annotations, resulting in hybrid learning of old knowledge and new knowledge"?

What is the strategy for using pseudo-labels versus new labels?

Are pseudo-labels only used for the training images which have GT labels?

Video adaptation is another core method employed here where very little detail is provided.

To be clear: the proposed video adaptation method, which the authors claim is a novelty in the abstract, is the process of using score-filtered predictions as pseudo-labels for fine-tuning?

What is the training procedure exactly?

How are inference target samples selected for use as pseudo-labels?

Is every frame in the video used for adaptation?

In what way does this reduce jitter other than what is shown qualitatively in Fig 2e?

What is meant by "This does not take extensive training time, and can be run during video analysis. For example, if a video (of a given size) can be run at 40 FPS, this would take approx. only 12 FPS to fine-tune the model and make new predictions"?

Does this mean that inference is run at 40 FPS?

If so, how does this relate to the 12 FPS figure if this involves training?

How long was the actual training time?

On which dataset and under which conditions?

The authors show that test-time multi-scale search is essential for robust generalization to new datasets in the zero-shot setting, but the main trade-off of reduced inference speed is not reported or discussed.

Evaluation and Comparisons:

OKS is a better metric than RMSE due to how visibility and scale are accounted for, though its variance scaling constant should be selected judiciously and clearly reported. Here the authors only use OKS as part of the summary mAP calculation. An even more useful metric would be the error at the tail of the distribution, reflecting how well models perform in the worst case scenarios.

Is there a reason that the performance of this method on AP-10K (arXiv:2108.12617) is not reported separately from the pooled Quadruped dataset? AP-10K would be useful for benchmarking this work against other competing approaches.

Is there a reason that comparisons to other methods for zero shot animal pose estimation or domain adaptation are not provided? This makes it difficult to assess the novelty and performance of this method. It would be particularly important to at a minimum reference (but preferably compare against)

approaches like Cao et al 2019 (arXiv:1908.05806), Kulkarni et al 2020 (arXiv:2004.00614), Sanakoyeu et al 2020 (arXiv:2003.00080), and Li et al 2023 (arXiv:2303.15023).

Some discussion is also necessary to position this work against fully self-supervised approaches for animal pose estimation, such as Bala et al 2021 (arXiv:2110.00543) and Sun et al 2021 (arXiv:2112.05121).

I found it difficult to compare methods in Fig 2h since the figure legend reuses colors.

Fig 3: a comparison to a standard behavior segmentation benchmark, such as Sun et al 2022 (arXiv:2207.10553), would provide stronger and more readily comparable evidence for good performance on this downstream task.

More Information on the SuperAnimal Model:

Since the model reported in this study is intended to be used directly for scientific applications, it is important that the authors share sufficient detail about the models, so that use applications do not suffer from any biases that might be baked into these models. I suggest providing the following (though the authors may have additional details they would like to provide):

Dataset datasheet, as described in Gebru et al 2021 (arXiv:1803.09010)

Crowdsourced annotation datasheet, as described in Diaz et al 2022 (arXiv:2206.08931)

Model cards, as described in Mitchell et al (arXiv:1810.03993)

The authors do not discuss any potential pitfalls of employing a foundation-like model, especially in a zero-shot setting where model biases can introduce structural errors that may impact downstream scientific results.

The authors state in the Reporting Summary that "In the following we detail references for those datasets", but these are not provided.

I believe a major contribution of this study is a unified SuperAnimal dataset - the authors should provide this as part of the resource - the dataset should be made publicly available.

Another major contribution is a unified vocabulary for animal keypoints, but this is only provided through figure illustrations rather than machine parseable text or code to do the dataset unification.

The model checkpoints strictly require the authors' own software or web app in order to be used, rather than standalone modules for training and fine tuning. At a minimum, can the authors provide reproducible environments (e.g., using Code Ocean, Hugging Face Spaces, or other platforms).

References to the Golden Lab and Smear Lab datasets should point to valid persistent DOIs or RRIDs (currently just "Sam Golden. Open-field Social Investigation Videos, July 2022." and "Matt Smear. Olfactory Search Video, July 2022." which do not provide a way to access or find more information on these data).

Other Comments:

The authors state in the Reporting Summary that "typical prior datasets were 100-800 frames, but here we provide 40K"; to my reading, this seems counter to the authors' own previous work (Lauer et al, 2022) and that of others (Graving et al, 2019; Bala et al, 2020; arXiv:2108.12617). Can they explain what was meant?

Is SuperAnimal trained on all of the described datasets? The authors should make this clearer in the text (to make it easier to understand which datasets are out of distribution).

The authors should quantify what is different about the OOD datasets (in terms of behavioral chamber characteristics, lighting, number of animals, appearance of animals, etc.)

Response to Reviewers

May 14, 2024

ORIGINALLY SUBMITTED: August 2nd, 2023
**Updated May 2024 to remove images with copy-
rights**

Overall response: We greatly thank the reviewers for their overall positive evaluation of our work. Below we aim to clarify and extend the work as requested. Our major revisions are to significantly improve performance on the Quadruped model, its presentation, and demonstrate the model’s utility on an example behavior. Our comments are in blue throughout.

Reviewer 1

Reviewer 1 (Remarks to the Author):

Ye and colleagues introduce SuperAnimal, a series of innovations, which ultimately provide two pretrained pose models that allow users to generate pose estimation tracking without the need to label frames. They demonstrate that the models outperform the current state-of-the-art algorithms on out-of-distribution videos. Even if manual labeling is required for fine-tuning, SuperAnimal boosts performance and saves time. The developers cleverly envision a system that can be updated and improved by the users, as it can handle and combine datasets that are not identically labeled across labs. SuperAnimal addresses a very important challenge in the field, as it allows to generate richer pose-estimation datasets that can be compared across labs, while also reducing time required for labelling frames. SuperAnimal is an exciting new step forward and we recommend this manuscript for publication.

During the review process, we have validated that the “Top-View-Mouse” model works well on out-of-frame videos from our lab (particularly if those videos are similar to those in the training set). After testing SuperAnimal, we propose minor comments to be considered:

Firstly, we greatly thank the reviewer for this overall positive assessment and acknowledgement of the innovations in the work and for testing

the models and code, it is highly appreciated and know this takes extra efforts.

Minor comments, General:

The title should be toned down. The term “plug-and-play” has been overused in marketing and advertising, leading to unrealistic expectations and frustration for consumers. Despite the user-friendly interfaces, DLC is not truly plug-and-play (see also comments below).

We definitely did not mean to say our deep learning package is plug-and-play, we simply meant the model is a one-click download and can be used for zero-shot inference. There is no other animal pose estimation package that provides pretrained animal pose estimation model weights that are usable in this way. But, we completely understand the reviewers point and amended the title accordingly to drop the term that might cause confusion: “SuperAnimal pretrained pose estimation models for behavioral analysis.”

The Github repository (<https://github.com/AdaptiveMotorControlLab/modelzoo-figures>), where the code and data should be available to reproduce the figures does not seem to be up to date. For instance, if the reader wants to check how the example images were selected for Figure 1h or 1i (see also comment below), there is no code available. Please update the files.

We apologize for the oversight that we did not push the latest updates. This has been now resolved.

The authors should comment on whether SuperAnimal can also run on multiple animals (e.g. 2 or more mice in one arena) or not. I assume it does not (yet), which is not a problem, but should be clarified.

The models can be used for multiple animal pose estimation in top-down mode for the quadruped model and in bottom-up mode for the TopView-Mouse model. We did show qualitative performance on multiple mice in Figure 1g and quantified this in Figure S3, which was now moved to Figure 1h to help clarify.

It would be nice to be able to easily select which points the SuperAnimal networks label. This would allow specific points to be tracked, perhaps even using the labels the initially trained networks used? This would make it easier to integrate with existing post processing pipelines. It might also be nice to have a clearer diagram of how to label the points for refinement. It seems several points on the body or tail are a kind of spatial division rather than being located on a feature that is easy to define such as the nose, eye, tail-base etc.

Right now the user gets all the keypoints from the model in a HDF5 (or csv) pandas-readable file. This means it is easy to index only the body parts you want to use in any downstream step. Also, for fine-tuning the user does not need to use “our” model keypoints – they can add new points that do or do not overlap. We have the label guide in the Suppl. Figure 1 (for TopViewMouse and Quadruped), but this labeling scheme is not strictly required. We added this here for ease of reviewing:

Figure 1: **Keypoint diagrams** Cartoon mouse on the right is adapted from scidraw.io: <https://beta.scidraw.io/drawing/183>.

The installation/use of SuperAnimal could still be improved, here are some suggestions:

A. A complete beginners guide to installing Deeplabcut ready to use modelzoo would be helpful, with all commands and installs. For instance, to get the supplied conda environment for deeplabcut they suggest git cloning it but don’t tell the user to first install git. Small things like this can be big hurdles for labs that have little experience installing non-commercial software packages (see plug-and-play comment above).

We thank the reviewer for this point, but respectfully think this paper is not meant to be a user-guide; we have extensive documentation (deplabcut.org), YouTube videos with over 100K views, an active user forum with over 1500 questions where users can get active help (<https://forum.image.sc/tag/deeplabcut>), and a Nature Protocols paper to assist users with a step by step installation guide (Nath et al. 2019). We also have a free DLC online course that introduces the basics of Git, Python, etc. and adding this to a technical paper feels well beyond the scope. The work as presented here has a dedicated website (modelzoo.deeplabcut.org), user-guide, and testing options on HuggingFace and our custom web app. We added images of these interfaces to help guide the reader to its multiple ways of use in Extended Data Figure 3, and for review we added is figure panel for you here:

Figure 2: **[Redacted] Examples of use cases of SuperAnimal model** s in (top left,clockwise): the browser, on HuggingFace, on Google Colab oratory (GPUsupport), and local use via a graphical user interface. Command line codefor installation is `pip install 'deeplabcut[tf,gui,modelzoo]'`.

B. highlight the fact that you need to use colab to get access to the *superanimal_quadruped* and *superanimal_topviewmouse* networks (unless it can be found elsewhere already, if so please clarify, we looked but couldn't find it). Although colab worked nicely, lots of users will want to move it to their local machines and not link their google accounts with colab.

As noted in the original manuscript it is available within DeepLabCut already (Colab installs `'pip install deeplabcut[modelzoo]'`), so by using the conda file as you note above you have the model weights as well and can use them locally. You can also download them directly from HuggingFace (banked since first submission at <https://huggingface.co/mwmathis>), therefore an end user need not only use them within DeepLabCut.

C. Try to explain the *scale_list* function in more detail, maybe give an example of e.g. if you have a 1024x768 video set the function to x? or tell users where they can find out what their image height pixel size range and increment are.

Unfortunately the *scale_list* is not one-to-one related to the image size, but rather the animal size, hence the need to possibly adapt this and test a range. Thus, while we can note in the Methods that one should be mindful of the animal size and attempt to have the animals be roughly 400 by 400 pixels for optimal scale, this is not a hard requirement.

The figures would benefit from some adjustments:

- Figure 1b-d: It is hard to figure out which parts belong to which subfigure (particularly 1c vs 1d). Could the authors increase the spacing between them to make it clearer?

Thank you, we adjusted the spacing and added a new Figure 2 dedicated just to the Quadruped results to aid in clarity.

- Figure 1h: It was very hard to recognize the keypoints on the mouse on some of the IID pictures, especially if someone wants to compare between predicted points and ground truth. It would be nice if the authors find a way to improve visibility (e.g. by choosing different colors for the keypoints, or maybe simply providing a higher-resolution/bigger image in supplementary data)

Thank you, we increased the size and also include the link here to the high resolution images in the code/data repo such that a reader can inspect more closely: <https://github.com/AdaptiveMotorControlLab/modelzoo-figures/tree/main>.

- Figure 1h-i: The authors should state (e.g. in the figure legend) how these example images were selected. Are these the ones where SuperAnimal performed “best” (based for instance on visual inspection by the authors) or were they selected randomly?

They were selected by hand such that we could see the animal and keypoints clearly. They were not automatically ranked and selected on best performance. Note to address some questions from other reviewers we updated the skeleton plotting, but again did not pick “the best” image by performance to show (they are the same raw images as prior). We added this to the figure legends.

- Figure 2e: Could the authors add information about the time window (or frame number) these plots were created for, either by adding it to the plot itself or the figure legend?

Horizontal scale bars, that indicate the frame number per area shown, were added to the plots (note this is now in Figure 3).

- Figure 2h: The chosen colors make it hard to distinguish between the different models.

Apologies, this has been revised (and moved to Figure 1).

- Figure 3a: The meaning of the icons on the right is not very clear, please add short labels/text

We added a clarifying label about the cartoon diagram, thanks for that suggestion.

Typo: • On page 2: "...the pre-trained encoded..." should be "encoder...".

Thank you, fixed!

1 Reviewer 2

The authors introduce SuperAnimal models, a project aimed at creating setting-invariant pose estimation models for animals, akin to similar models capable of estimating human pose in diverse settings. To create these models, they assemble an annotated dataset of pose keypoints from 45 species of mammals, consisting of over 45,000 images (for comparison, the widely used MPII Human Pose dataset consists of 25,000 human pose annotations, while COCO contains more than 250,000 human pose annotations.) They present model performance in zero-shot pose estimation and few-shot pose estimation, showing improvements over previous DLC models. They then evaluate the zero-shot version of their TopViewMouse model in two contexts: for tracking of animals in videos, and for detection of rearing behavior. While the zero-shot performance of the TopViewMouse model seems reasonable, the performance of the Quadrupeds model seems so low that I doubt it will be useable for any form of behavioral analysis. I also had some concerns with how models are evaluated, with how representative their behavior analysis results are, and with the design of the “video adaptation” method used to reduce keypoint jitter. Overall, these concerns are quite significant, and I feel must be addressed before the paper can be considered for publication.

We thank the reviewer for their time and efforts. We also thank the reviewer for noting our performance for TopViewMouse seems reasonable, and agree we could have better presented the Quadruped model. We would like to point out, however, that building larger pose datasets required innovation in how to build and train models. For example, COCO or MPII pose images have identical annotations within each dataset (respectively), whereas animals have different body shapes and non-standardized keypoints across datasets, which poses significant challenges. We also show not only performance over prior DLC models, but over other established models in the field like ViT-MAE and ViT-DEiT within mmPose, another leading package.

My major concerns are as follows:

The authors present both a top-view mouse SuperAnimal model and a quadruped model. While the utility of the mouse model for the study of behavior is examined in Figures 2-3, the quadruped model is only briefly explored in Figure 1 and Figure S3. The quadruped pose estimates in Figure 1 are better than chance, but they are often very incomplete (eg wolf, tiger, sheep, raccoon) and I’m skeptical that they will be of any practical use for behavior analysis. This paper only demonstrates its titular claim of “plug-and-play analysis of animal *behavior*” in the case of top-view videos of mice in a laboratory setting (Figures 2-3). To present the SuperAnimal method as facilitating animal behavior analysis beyond the specific case of lab mice is premature and an overstatement of the results that are shown.

Firstly we apologize for the poor plotting in Figure 1; we did not fully connect the skeleton and the images were too small, we have now rectified this extensively. Secondly, we completely agree we overly focused on the mouse setting, having neuroscientists in mind who likely use the TopView-Mouse model most. We have now done several key things: (1) We added a new Figure on Quadruped performance, and we significantly improved our model. Concretely, we expanded our work on AnimalTokenPose and also developed a new model based on HRNet-W32 with top-down inference vs. bottom-up as we previously only included (top down is better suited for natural scenes where the animal only occupies a fraction of the space in the image [1]). Here is the new relevant figures showing qualitative performance of our new Quadruped model (Fig. 3). Concretely, we now show competitive performance on Horse-10 (2X improved from our first submission), improved performance on iRodent by nearly 10X, plus added benchmarking on AnimalPose and AP-10K. We show our zero-shot AnimalPose performance is now as good as the model that is fully trained with ImageNet weights on AnimalPose. Lastly, we updated the title.

Figure 3: **[Redacted] Updated qualitative performance of Super Animal-Quadruped HRNet, IID and OOD dataests.**

Quantitatively, the quadruped model is reported to have a zero-shot performance of ~ 1.1 normalized units on Horse-10, where 1 = the distance from a horse’s nose to its eye. My guess is that this would work out to a bit over a foot; I have trouble visualizing any application where a model with an *average* keypoint error of more than 12 inches is going to be useable.

We worked to substantially improve the performance. Our updated performance for zero-shot is now $2\times$ as good, namely around 0.6 — this is a $4\times$ improvement over ImageNet-based transfer learning with 14 frames, and one would need to label over 600 images (so ~ 6 -10 hours of work then training for another 6-10 hours) to get the same performance that our model gives out-of-the-box. And, with fine-tuning using our newly developed methods presented in this paper, we achieve a NE of 0.1091, given enough training data. And, a user can label only 73 frames vs. the 734 frames they would have needed to label beforehand (so a $10\times$ data efficiency) for the same performance as supervised training with ImageNet-based weights! We thank the reviewer for pushing us, and hope they now clearly sees the benefit of this pretrained model class. Here are the new results for ease (Fig. 4), which of course is updated in the paper.

The normalized error on iRodent is around 0.6, where 1 = the square

Figure 4: **Horse-10 performance**, updated (dotted blue line is zero-shot performance with HRNetw32) Seagreen arrows note the change in labeled data required to hit the same performance with ImageNet-based transfer learning, and pink denotes the change in performance between zero-shot vs. needing some labels with fine-tuning. Image adapted from “Pre-training boosts out-of-domain robustness for pose estimation” WACV (Jan 2021) <https://www.mackenziemathislab.org/horse10> and released under a CC-BY-NC license: <https://creativecommons.org/licenses/by-nc/4.0/>.

root of the bounding box area. Assuming a roughly square bounding box, this equates to 1 being on the order of the body length of the animal; an average keypoint error of 0.6 body lengths is again not likely to be useable. Given these very high error rates, the claim that the SuperAnimal-Quadruped model is a zero-shot learner seems to be not supported by the data, and should be removed.

We improved the performance of the model to a zero-shot performance on iRodent of approx. 0.07 NE (so around 10× improved). We also changed the metric to mAP based one your request below and used this in the updated manuscript. We plot both metrics for reference here (Fig. 5). Thus, we hope that we clarified that this model can be used zero-shot or as a few-shot learner. Despite the 10× performance improvement we can show here, iRodent remains a hard challenge, and we want to include it to show remaining limitations — there is a gap to close that new innovations in future work might be able to tackle.

Some concerns regarding how choice of metrics for reporting performance:

- a. The “normalized error” plots in Figure 1F are difficult to interpret – error is reported in units of either eye-to-nose distance (for horses) or the square root of the bounding box area (for rodents). This normalization is

Figure 5: **iRodent performance, updated**. Note we show mAP (higher the better) based on R3 request, but show NE (lower the better) as in the original manuscript for R2, both our updated model and what we original submitted, for comparison.

not used anywhere else in the computer vision, and the bounding box metric in particular is not a good choice, as simple changes like rotation of the animal will change the area of the bounding box substantially. (To see this, picture an elongated object like a running mouse- when oriented horizontally or vertically, its bounding box will be a long rectangle, with length L and width $W \ll L$, giving it an area of $W \cdot L$. When oriented diagonally, its bounding box will be a square with diagonal length of roughly L (actually slightly longer), giving it an area of $1/2 L \cdot L$.)

We used normalized error specifically as it is used in the Horse-10 Benchmark from WACV 2021. To aid interpretation we added a diagram to the Figure where we use this metric for the first time. For iRodent, we changed the metric to mAP, as requested (see new Figure 2).

b. Performance is also difficult to evaluate as the SuperAnimal model is only compared to itself (in various configurations), and to ImageNet (which is a poor comparison as it was not developed for pose estimation.) Although the mAP of the object-keypoint similarity is reported in Supplemental Figure S3, the authors do not explain in the methods what values of s and k

were used in this calculation. Typically, s is the ratio of the bounding box to the image size, however k is a body-part specific value that is estimated by computing the variance of human-generated labels of that part within an image (see Ronchi and Perona 2017, which introduced this metric.) While field-consensus values of k exist for human body parts, there are no such values for animal body parts. To use the OKS metric, the authors would need to either select a k based on values from the human literature, or preferably, obtain an estimate of the variance of manual annotations of each body part, typically by having multiple individuals annotate a representative set of images.

We respectfully disagree that ImageNet-based transfer learning is not a good baseline. This is the standard in the field for SOTA animal pose estimation, see [2, 3, 4, 5, 6, 7, 1]. Only recently, datasets of sufficient size became available to make animal pose-aware pretraining relevant (i.e., allow to build models suitable for transfer learning without an ImageNet backbone). Our work here, to the best of our knowledge, is the first to show that using animal pose-aware base models outperform ImageNet-pretrained models: this is one major claim of this work.

In terms of metrics, we explain above why we used normalized error to match prior benchmarks or we used RMSE in pixels on datasets like DLC-OpenField to specifically allow readers to compare to prior art on each of these datasets.

In regards to mAP calculation, thank you for pointing out our methods section can be improved. We now added what s and k is to the Methods (see Methods Table 1), where we follow the convention in previous works [6, 8, 5] to use k from the COCO human keypoint benchmark. For those keypoints that have no correspondence to COCO human keypoints, we use the averaged k in COCO human keypoints. This setting eases the comparison to previous works. We also now clarify the implementation reference vs. the OKS ref in the Methods.

Method clarifications:

The methods section on the test time spatial-pyramid search is a little difficult to follow, and I'm not 100 sure how the method works. It seems somewhat similar to the hourglass model for pose estimation, and it would help if the authors could explain the relationship between the two. This methods section also seems to confound two problems: 1) difference in the appearance size of the animal across datasets, and 2) difference in the image resolution across datasets. The former is a known challenge in computer vision, whereas the latter (which is emphasized in the methods section, particularly in the last paragraph) seems like it could be trivially resolved by simply rescaling the input image during inference.

Relationship to hourglass model: models like HRNet or Hourglass fuse features from heatmaps of different resolutions, hoping to help pose models to learn smaller objects in the **train dataset** by modifying the architecture. However, our spatial pyramid method is a **test-time augmentation** that aims to improve **domain generalization**, meaning that we want our model to handle both extra large and extra small objects in unseen test datasets that have very different size distributions.

Appearance size and resolution: The unpredictable animal size in images is a hard challenge, especially for bottom-up models (which we used in our first submission for both TopViewMouse and Quadruped). Unlike top-down approaches, bottom-up models do not standardize animal appearance size, and models learn the size of bodyparts in a way that is only best for the size of bodyparts in the training dataset. This is why simply using model like an hourglass is not sufficient. On the other hand, simply re-scaling the resolution of the image once is also not sufficient. Imagine we have two images: one where animal is close to the camera, and another image where the animal is very far, rescaling the image doesn't change the animal ratio. As we clarify in the pseudo-code below, **spatial pyramid** uses multiple criterion's to search for the best scaling factor for every single image. We also added pseudo-code to the Methods and here for ease of review:

```

1 def spatial_pyramid_search(images, model, scale_list,
2   confidence_threshold, cosine_threshold):
3   # generate rescaled version of original images with multiple
4   # scaling factor
5   rescaled_images = rescale_images(images, scale_list)
6   preds_per_scale = []
7   # gather predictions of the model, assuming the final
8   # pred_keypoints are projected to the original image space by
9   # the forward function
10  for rescaled_image in rescaled_images:
11    pred_keypoints = model(rescaled_image)
12    preds_per_scale.append(pred_keypoints)
13
14  # using median to get a good estimate of expected keypoint
15  # positions
16  median_keypoint = get_median_keypoint(preds_per_scale)
17  # If the rescaled image is not suitable for the model,
18  # we expect the model have a confidence less than a given
19  # threshold
20  pred_keypoints = filter_by_confidence(pred_keypoints,
21   confidence_threshold )
22  # A median filter alone does not remove outliers. After
23  # confidence filtering, we compare the remained predictions
24  # to the median keypoint and drop the low quality predictions
25  pred_keypoints = filter_by_cosine_similarity(
26  pred_keypoints, median_keypoint, cosine_threshold)
27
28  return get_median_keypoints(pred_keypoints)

```

The video adaptation method uses model-predicted keypoint locations as a source of training data to the model itself. I can see how this improves model prediction when quantified in terms of keypoint jitter, given that for each new frame T, the model has a labeled frame T-1 in its training set and simply has to recapitulate the label placements of that training example. My concern, however, lies in the problem of catastrophic forgetting- nothing is constraining the model-predicted keypoint locations used for training to be accurate, and it seems quite possible that there could be drift in their definitions over time, as repeated iterations of fine-tuning lead to forgetting of the original human-generated training set. If the authors want to present this method as a resource for improving accuracy of pose estimation, it should be evaluated not in terms of keypoint jitter (which would not detect such catastrophic forgetting of keypoint definitions), but in terms of metrics like OKS or RMSE, as used in Figure 1.

First, we want to note that jitter isn't being used as a metric for measuring forgetting or for key point accuracy, it is showing that with our method the model is actually useful, namely, low jitter in video for the users, and this is achieved through video adaptation. Secondly, we agree and are aware of "drifting in the definition"/"mode collapse", in the typical practice of pseudo-labeling. Therefore, we used the zero-shot prediction and fixed it during the whole process of memory replay so there should not be drifting if the model converges. We hope that clarifies the method more clearly, and we added this to the Methods.

The paper includes multiple model architectures and modifications (CNNs vs transformers, with and without spatial-pyramid search and video adaptation). It would be very helpful for the authors to include a table explaining which model configuration was used for each result reported in the paper.

We now added two Tables in the text that shows the model variants and results. Concretely, unless noted all experiments were done with a SuperAnimal-TopViewMouse and/or SuperAnimal-Quadruped model that held-out all benchmark datasets. Here are the Tables for ease:

The difference in performance in classifying supported rears vs unsupported rears is striking- while detection of supported rears seems useable, the performance on unsupported rears is quite low. I suspect that this is because supported rears can only take place at the walls of the arena, therefore the classifier could detect them just by learning to recognize when the mouse is near the walls of the arena, without incorporating any postural information. It would be helpful to see evaluation on a wider range of behaviors; I strongly suspect that the performance seen in the unsupported rearing condition is more likely to be typical for detecting behaviors that

Table 1: **Main Results on Mouse Benchmarks.** The mAP comparison between CNN baseline (HRNet) and Transformer based models (TokenPose models and ViTs) on SuperAnimal-TopViewMouse. The performance of HRNet is comparable to performance shown in Extended Data Fig. 3c), and transformers most-always outperformed HRNet. Here we used mmPose [8]) as well. Suggesting that users also may consider other architectures when building SuperAnimal models, especially for zero-shot inference.

Model	TriMouse	DLC-Openfield
ViT-MAE	32.4	63.6
ViT-DeiT	34.2	62.4
AnimalTokenPose (HRNet)	33.4	91.5
AnimalTokenPose (CNN)	43.5	82.7
HRNet	28.1	72.3

are not dependent on positional information.

A couple things to unpack here. Firstly, we evaluated our model on a published dataset where its performance is usable to downstream users and those authors. Our goal is to show that SuperAnimal models can be used in this highly common behavioral setting by showing we get the same performance as previous methods. Secondly, one cannot detect the rears at the wall by only proximity of the animal at the wall, they need to actually rear (mice spend most of their time at the wall, this is a classical test for anxiety). Therefore we have left in this analysis as we feel it achieves the desired goal of showing the zero-shot performance use of SuperAnimal-TopViewMouse.

Other/minor points:

It is not clear how the RMSE metric accounts for missed keypoints (points that are incorrectly reported as occluded). This should be added to the Methods. Or, if missed keypoints do not contribute to the RMSE, then the authors might want to use a different metric that incorporates this form of error.

Keypoints can never be missed in the predictions in our models. They can have a low confidence, but they are always predicted on every frame, thus RMSE accounts for “missed” keypoints. We did not drop low confidence points during evaluation.

The citation for OKS points to the original Microsoft COCO paper, which predates the creation of the OKS metric by a few years. While OKS is now a part of the COCO API, it was only added later and is not discussed in the original manuscript. Ronchi and Perona 2017 introduces OKS and

Table 2: **Main Results on Quadruped Benchmarks.** Here, the base SuperAnimal-Quadruped model had none of the heldout datasets, with the exception of AnimalPose (AP). For the AnimalPose experiments we make a new variant that dropped AnimalPose data (SA-AP), thereby making it OOD for this setting as well. Full results can be found in Figure 2 for fine-tuning with different amounts of data, but the best-case fine-tuning performance is shown, which matches the top-performance of the SuperAnimal (SA) variant as shown in Figure 2. *NOTE: Cao et al.[9] do not report a unified single mAP, rather per animal, therefore we trained a model using their dataset to estimate top-line performance if only trained on AP. **Number as reported in [10] using the data from [6].

Benchmark:	AnimalTokenPose (HRNet)	HRNet2 (w32)
mAP (\uparrow , higher the better)		
iRodent: SA zero-shot	36.2	39.3
iRodent: fine-tuning ImageNet w/1%	-	02.8
iRodent: fine-tuning SA w/1%	-	40.4
iRodent: fine-tuning ImageNet w/100%	-	57.0
iRodent: fine-tuning SA w/100%	-	68.0
AP-10K: SA zero-shot	45.3	50.8
AP-10K: AnimalPose weights, zero-shot	-	27.0
AP-10K: ** [10]		
fine-tuning w/ImageNet	-	72.2
AP-10K: fine-tuning w/SA	-	78.3
AnimalPose: SA-AP zero-shot	86.0	85.4
AnimalPose: SA+AP-10K zero-shot	-	69.4
AnimalPose: Fine-tuning ImageNet*	-	88.2
AnimalPose: SA-AP Fine-tuning	-	90.4
Normalized Error (\downarrow , lower the better)		
Horse10: IID SA zero-shot	0.647	0.673
Horse10: OOD SA zero-shot	0.613	0.640
Horse10: IID SA fine-tuning ImageNet	-	0.049
Horse10: OOD SA fine-tuning ImageNet	-	0.179
Horse10: IID SA fine-tuning	-	0.047
Horse10: OOD SA fine-tuning	-	0.109

explains it in detail.

We apologize, we cited what we used for the implementation, but we now include the citation to Ronchi and Perona 2017 to the paper.

IID usually means “independent and identically distributed”, making its use as an abbreviation for “within-distribution datasets” a little confusing.

We apologize for this, we now point to both terms for this (as both are used in animal pose literature).

Figure 1f should include units for the RMSE on DLC-Openfield (probably pixels?)

We now clarify this is in pixels.

“We outperformed DeepLabCut-ResNet-50 ... by over 2X with 10X less data” – this is worded as if SuperAnimal + 10 frames has 2x the performance of DLC + 100 frames, while in fact DLC + 100 is slightly better than SuperAnimal + 10. You could write “We outperformed DLC by 2X” or “We matched DLC performance with 10X less data”, but you can’t claim the combination of the two. (A car that gets 60 mpg can drive twice as far per gallon of gas as a car with 30 mpg, and it takes half as much gas to travel the same distance- but it can’t drive twice as far AND burn half as much gas in doing so.)

Great catch – we were a bit too fast on the gas pedal. Now updated: “Therefore, we outperformed DeepLabCut-ResNet-50 (i.e., our ImageNet pre-training baseline) by over 2X with only 10 frames of labeling, and we can achieve the same performance with 10X less data.”

Some figure text is very small and hard to read.

We apologize, we revised and made every effort to clarify figures.

Figure 2h, bar colors are very similar/the same for multiple models, please use a different color for each model tested.

Apologies for this, we updated (and moved to Figure 1).

Reviewer 3

Review of Ye et al.

In this manuscript, Ye and colleagues develop new neural network models they call “SuperAnimal” that are pre-trained on a variety of relevant behavioral imaging data (for example, videos of mice in different behavioral chambers/settings), and then can be used “out of the box” with no training or “zero shot” on new behavioral data (in different chambers/settings). While the performance of the model already appears to be good, they then develop a framework to improve performance through what they call fine-tuning (labeling a small amount of data in the new chamber/setting and using SuperAnimal pre-trained weights). The concepts here are very exciting - having a model that can be used without the burden of labeling and training on any new behavioral dataset (independent of different lighting conditions, visual appearance of the behavioral chamber and animals, etc.) would be very impactful for both behavioral and neuroscience research, particularly for mice (what is primarily demonstrated here). Despite my excitement for the method itself, the authors do not provide sufficient details for me to evaluate the model and its performance. This is surprising since best practices are well established for such models - I cite several of these below which might serve as examples for the authors. In general, the authors highlight an exciting new method, but without providing the details to support whether the model performs to match their claims - this could be fixed by the authors following some of the suggestions below.

We thank the reviewer for their time and expertise and for noting the potentially exciting concepts. We apologize for a lack of clarity on the Methods and we have worked hard to clarify and provide pseudo-code in the methods where appropriate.

Methods:

Memory-replay is a core method employed here to prevent catastrophic forgetting, but very little detail is provided on how this is implemented.

We clarify in the Methods and provide pseudo-code (also here for ease), thanks for this excellent suggestion:

```
1 def is_defined(keypoints):
2     # check whether the original dataset defines each keypoint.
3     # We use a flag '-1' to denote that a given keypoint is not
4     # defined in the original dataset. Note this is different
5     # from not annotated, which use flag '0'
6     return True if keypoints[2] >= 0 else False
7
8 def load_pseudo_keypoints(image_ids):
```

```

6 # get the pseudo keypoints by image IDs.
7 # note, pseudo keypoints are loaded from disk and fixed
  throughout the process, so not drifting as is expected in
  typical online pseudo labeling
8 return pseudo_keypoints
9
10 def get_confidence(keypoints):
11 # get the model confidence of the predicted keypoints. Unlike
  ground truth data that have 3 discrete flags, predicted
  keypoints have confidence that can be used as likelihood
  readout for post-inference analysis
12 return keypoints[2]
13
14 def memory_replay(model, superset_gt_data_loader, optimizer,
  threshold):
15
16 # gt data is preprocessed such that annotations are now in
  superset keypoint space.
17 # every gt keypoint has 3 flags (-1: not defined, 0: not
  labeled, 1: annotated)
18
19 For batch_data in superset_gt_data_loader:
20
21 gt_keypoints = batch_data['keypoints']
22 image_ids = batch_data['image_ids']
23 images = batch_data['images']
24 # model() is a pytorch style forward function
25 preds = model(images)
26 pseudo_keypoints = load_pseudo_keypoints(image_ids)
27 # 3 here is (x, y, flag)
28 batch_size, num_kpts, 3 = gt_keypoints
29 # iterate through batch
30 For b_id in batch_size:
31 # iterate through keypoints
32 For kpt_id in range(num_kpts):
33 # since this specific body part is not defined in the new
  dataset, we use saved pseudo labels (zero-shot prediction)
  as gt. This prevents catastrophic forgetting and drifting.
  We can also use confidence to filter the pseudo keypoints
34 If not is_defined(gt_keypoints[b_id, kpt_id]) and
  get_confidence(pseudo_keypoints[b_id][kpt_id]) > threshold:
35 # we assume a single animal scenario for simplicity. For
  multiple animals, matching between gt and pseudo keypoints
  need to be completed.
36 gt_keypoints[b_id][kpt_id] = pseudo_keypoints[b_id][
  kpt_id]
37
38 loss = criterion(preds, gt_keypoints)
39 optimizer.zero_grad()
40 loss.backward()
41 optimizer.step()

```

What, precisely, is meant by "When we fine-tune a SuperAnimal model, we replace the model predicted keypoints with the ground-truth annota-

tions, resulting in hybrid learning of old knowledge and new knowledge”?

We use the model predictions on new frames AND we use the original ground truth annotations and images as new inputs for training. When we perform video adaptation we do not have, nor need, access to GT. We hope that clarifies the sentence.

What is the strategy for using pseudo-labels versus new labels?

We provide pseudo algorithm code above to clarify the algorithm.

Are pseudo-labels only used for the training images which have GT labels?

GT labels are not needed for video adaptation, as we demonstrated in Figure 2e where we tested video adaptation on unlabeled OOD datasets.

Video adaptation is another core method employed here where very little detail is provided. To be clear: the proposed video adaptation method, which the authors claim is a novelty in the abstract, is the process of using score-filtered predictions as pseudo-labels for fine-tuning?

While score-filtered predictions is part of our video adaptation and established in the field, our novelty lies on following: 1) application scenario: while pseudo labeling is commonly used in image-based domain adaptation or image classification, to our knowledge it is novel that we found it very useful to improve video frame prediction smoothness. 2) our pseudo labeling is combined with spatial pyramid, which augments pseudo labeling.

What is the training procedure exactly? How are inference target samples selected for use as pseudo-labels? Is every frame in the video used for adaptation?

We take the predictions for all frames in the target video as the training dataset. In practice, we only train for around 1000 iterations of stochastic batches. We select keypoints based on confidence but we do not select the target prediction (i.e. which frame the predictions correspond to) and rely on the random sampler instead. We clarified this in the Methods, and add here:

```
1 def get_pseudo_predictions(frame_id):
2     # return pseudo prediction by frame id
3
4 def video_adaptation(model, video_data_loader, optimizer,
5                       threshold):
6     for data in video_data_loader:
7         frame_id = data['frame_id']
```

```

7   Image = data['image']
8   pseudo_keypoints = get_pseudo_predictions(frame_id)
9   preds = model(image)
10  loss = criterion(preds, pseudo_keypoints, mask_by_threshold
11          = threshold)
11  optimizer.zero_grad()
12  loss.backward()
13  optimizer.step()

```

In what way does this reduce jitter other than what is shown qualitatively in Fig 2e?

We added quantitative measures in Extended Data Figure 4, and shown here for ease:

Figure 6: **Quantification of video statistics.** Without (grey) or without video adaptation (pink), showing some changes in jitter and dropped keypoints.

What is meant by "This does not take extensive training time, and can be run during video analysis. For example, if a video (of a given size) can be run at 40 FPS, this would take approx. only 12 FPS to fine-tune the model and make new predictions"? Does this mean that inference is run at 40 FPS? If so, how does this relate to the 12 FPS figure if this involves training?

This means that if video inference was running at 40 FPS without any adaptation, to now do video adaptation only slows this down to 12 FPS, there is no other training time of the model. A decrease from 40 FPS to 12 FPS means that in practice, it is still useful for downstream users.

How long was the actual training time? On which dataset and under which conditions?

For video adaptation training, the training time is truly the run-time of the model on the video, hence its an example as its always relative to the length of the video input and the original pixel-size of the frames, etc. As an aside, to train the whole SuperAnimal-Quadruped model we used for experiments in this paper it took 17 hours on 2 GPUs (NVIDA titan GPUs with 24GB memory each), but of course users do not need to do this.

The authors show that test-time multi-scale search is essential for robust generalization to new datasets in the zero-shot setting, but the main trade-off of reduced inference speed is not reported or discussed.

We added a note in the results, “Note that in practice this does slow down inference depending on the search parameter space.”.

Evaluation and Comparisons:

OKS is a better metric than RMSE due to how visibility and scale are accounted for, though its variance scaling constant should be selected judiciously and clearly reported. Here the authors only use OKS as part of the summary mAP calculation. An even more useful metric would be the error at the tail of the distribution, reflecting how well models perform in the worst case scenarios.

The main reason we use RMSE and Normalized Error for DLC-Openfield and Horse-10, respectively, is to compare with the original papers and ongoing benchmarks. In the DLC-Openfield dataset the animals do not change size and images are the same size, thus RMSE we believe is sufficient. As Sigma is not reported in the original papers, calculating mAP is difficult (but see response to R2 above), and we provide the mAP values on DLC-Openfield and the related TriMouse dataset in Figure 1, and we now report mAP on iRodent, AP-10K, and AnimalPose.

Is there a reason that the performance of this method on AP-10K (arXiv:2108.12617) is not reported separately from the pooled Quadruped dataset? AP-10K would be useful for benchmarking this work against other competing approaches.

Thanks for the suggestion. We now made a new SuperAnimal-Quadruped model split that has AP-10K held-out as another OOD test. Here we performed several new experiments (Fig. 7). We tested the performance of SuperAnimal on AP-10K zero-shot and for fine-tuning, with our SuperAnimal weights or ImageNet starting weights. We also used AnimalPose zero-shot to establish another baseline zero-shot performance. Here we find SuperAnimal outperforms AnimalPose zero-shot, and our SuperAnimal fine-tuning outperforms ImageNet fine-tuning, fully supporting our earlier results on a new benchmark. To note, our SuperAnimal fine-tuning with our HRNet-w32 backbone shows a performance of 78.3 mAP, while in the AP-10K paper, the authors used HRNet-w32 to obtain 73.8 mAP with ImageNet pretraining. We added this to the new Figure 2 and results table.

Figure 7: **AP-10K Benchmark.** SuperAnimal-Quadruped HRNet-w32 and AnimalTokenPose on AP-10K, zero-shot, and fine-tuning results.

Is there a reason that comparisons to other methods for zero-shot animal pose estimation or domain adaptation are not provided? This makes it difficult to assess the novelty and performance of this method. It would be particularly important to at a minimum reference (but preferably compare against) approaches like Cao et al 2019 (arXiv:1908.05806), Kulkarni et al 2020 (arXiv:2004.00614), Sanakoyeu et al 2020 (arXiv:2003.00080), and Li et al 2023 (arXiv:2303.15023).

Cao et al arXiv:1908.05806: we added benchmarking on this task to our work as well, per your request. We find that our **zero-shot performance** is as good as their fully supervised method. We now added this to new

Figure 2. Kulkarni et al 2020 is tackling surface mapping, not merging pose estimation datasets and showing innovation in this space, and the same for Sanakoyeu et al 2020, which is DensePose estimation. Li 2023 came out on arXiv after we submitted our work to Nature Communications, and per their policy: “When evaluating your revised manuscript, we will not consider any similar papers published in the meantime to compromise the novelty of your study. See <https://www.nature.com/articles/s41467-020-17817-x> for more information.”, therefore we do not benchmark this (also note its goal is different: “a pseudo label-based approach to generate artificial labels for the unlabeled images”). We feel that by adding both AnimalPose and AP-10K we have demonstrated our models performance such that readers can better judge its utility. Thanks for those suggestions!

Figure 8: **AnimalPose Benchmark.** SuperAnimal-Quadruped HRNet-w32 AnimalPose zero-shot matches performance of fully-supervised AnimalPose model, and outperforms AP-10K zero-shot.

Some discussion is also necessary to position this work against fully self-supervised approaches for animal pose estimation, such as Bala et al 2021 (arXiv:2110.00543) and Sun et al 2021 (arXiv:2112.05121).

The Bala preprint just appeared in IJCV June 2023 and we now cite it and discuss it; we note that Bala et al. is not fully self-supervised, but requires annotated “primary landmarks”.

In any case, we state in the discussion: “Alternatively, unsupervised keypoint discovery can be used [11, 12]. While the unsupervised approach requires no pose annotations, the learned keypoints might lack interpretability and it is not clear whether it allows zero-shot inference on OOD data. Therefore, both our approach that creates predictions based on the super-

set of annotated keypoints from different studies and unsupervised keypoint discovery are promising, complementary directions.”

I found it difficult to compare methods in Fig 2h since the figure legend reuses colors.

We apologize, we fixed this.

Fig 3: a comparison to a standard behavior segmentation benchmark, such as Sun et al 2022 (arXiv:2207.10553), would provide stronger and more readily comparable evidence for good performance on this downstream task.

Sturman et al is an established human expert annotated benchmark in the field of mouse behavior. Sun et al. has programmatically defined behaviors, not ground truth annotations. The main goal of our paper was to show the innovations in building foundational-like models for pose estimation. Beyond Sturman et al. we now include another downstream behavioral analysis for stride analysis (Figure 4).

Additionally, we note that SuperAnimal weights were used in another pre-print, showing that SuperAnimal-TopViewMouse predictions with keypoint-MoSeq [?] outperform other behavioral quantification methods on a separate benchmark dataset. For ease of the reviewers, we added that relevant figure panels here (Fig. 9). We cite this work as well. Additionally, we used the SuperAnimal-TopViewMouse model within AmadeusGPT [13] for several other mouse topview datasets, and for example, could show on the elevated plus maze it can match human-annotation performance there as well.

SuperAnimal-TopViewMouse model used in keypoint-MoSeq:

Figure 9: **Extended examples of SA-TVM in use in animal benchmarks adapted from [14].**

More Information on the SuperAnimal Model:

Since the model reported in this study is intended to be used directly for scientific applications, it is important that the authors share sufficient detail about the models, so that use applications do not suffer from any biases that might be baked into these models. I suggest providing the following (though

the authors may have additional details they would like to provide): Dataset datasheet, as described in Gebru et al 2021 (arXiv:1803.09010) Crowdsourced annotation datasheet, as described in Diaz et al 2022 (arXiv:2206.08931) Model cards, as described in Mitchell et al (arXiv:1810.03993)

We updated the model cards at HuggingFace. TopViewMouse: <https://huggingface.co/mwmathis/DeepLabCutModelZoo-SuperAnimal-TopViewMouse>. SuperAnimal-Quadruped DLCRNet: <https://huggingface.co/mwmathis/DeepLabCutModelZoo-SuperAnimal-Quadruped>

The authors do not discuss any potential pitfalls of employing a foundation-like model, especially in a zero-shot setting where model biases can introduce structural errors that may impact downstream scientific results.

We are aware of the systematic bias and potential impact, and we actually did mention this in the annotator bias section in the Suppl. To monitor the model biases, we propose a keypoint domain distance diagram to help monitor the distance between the expected annotations and models' predictions. In practice, fine-tuning can reduce such a distance and only keypoints estimated by fine-tuned and/or validated models should be kept for precision sensitive scientific applications.

The authors state in the Reporting Summary that "In the following we detail references for those datasets", but these are not provided. I believe a major contribution of this study is a unified SuperAnimal dataset - the authors should provide this as part of the resource - the dataset should be made publicly available. Another major contribution is a unified vocabulary for animal keypoints, but this is only provided through figure illustrations rather than machine parseable text or code to do the dataset unification.

The datasets are referenced fully in the Methods and in Suppl. Figure 1. Please note, not all of the mouse data can be made available due to limitations on institutional guidelines on sharing recorded data. Therefore, while we used primarily open source data, as we already referenced, those that are given to us courtesy of other scientists must stay private. The best we can do, as we do here, is to provide model weights. This is also an important point for scientists who might be hesitant to release their data. We also now added the unified keypoint main diagram to the figure repository so users understand the mapping, and a new panel in Extended Data 1 that has the mappings.

The model checkpoints strictly require the authors' own software or web app in order to be used, rather than standalone modules for training and fine tuning. At a minimum, can the authors provide reproducible environ-

ments (e.g., using Code Ocean, Hugging Face Spaces, or other platforms).

This is actually not correct; there is no requirement to use the model only in DeepLabCut software; we banked the TensorFlow model weights at HuggingFace already, and there is a HuggingFace Gradio space: (<https://huggingface.co/spaces/DeepLabCut/DeepLabCutModelZoo-SuperAnimals>). We also showed in the original submission that we can use the model weights in mmpose, another pose estimation ecosystem.

References to the Golden Lab and Smear Lab datasets should point to valid persistent DOIs or RRIDs (currently just "Sam Golden. Open-field Social Investigation Videos, July 2022." and "Matt Smear. Olfactory Search Video, July 2022." which do not provide a way to access or find more information on these data).

Apologies, the urls were dropped in the PDF display of the references and are now updated.

Other Comments:

The authors state in the Reporting Summary that "typical prior datasets were 100-800 frames, but here we provide 40K"; to my reading, this seems counter to the authors' own previous work (Lauer et al, 2022) and that of others (Graving et al, 2019; Bala et al, 2020; arXiv:2108.12617). Can they explain what was meant?

Apologies, this was a typographical error. It should have said: "**typical** prior datasets were 100-800 frames," – it is updated. To be clear, for example Bala et al. is only macaques in front of green screens (thus while very large, would massively imbalance the dataset if included), Lauer et al. contains 8K images of only marmosets, Graving et al 2019 is 800 images of Locusts, and 900 images of zebras from an aerial viewpoint. Most reports, as far as we know, don't have such large datasets, therefore in this reporting summary block **on how we chose the sample size** for our paper, we simply note we are beyond the standard range.

Is SuperAnimal trained on all of the described datasets? The authors should make this clearer in the text (to make it easier to understand which datasets are out of distribution).

No, to show the utility of SuperAnimal training, we built one "base model" that did not have any of the benchmarks datasets in it. Thus, every benchmark is truly OOD. We clarified this in the beginning of the Results section as well. The final production models that are released to the public are trained on all datasets.

The authors should quantify what is different about the OOD datasets (in terms of behavioral chamber characteristics, lighting, number of animals, appearance of animals, etc.)

A full analysis of every dataset is well beyond the scope of this work. We cite each dataset that is available, and we hope this is sufficient for the reviewer and users. Thank you again for all your constructive feedback, and we hope you now support publication of our work.

References

- [1] Mu Zhou, Lucas Stoffl, Mackenzie W. Mathis, and Alexander Mathis. Rethinking pose estimation in crowds: overcoming the detection information-bottleneck and ambiguity. *IEEE/CVF International Conference on Computer Vision*, 2023.
- [2] Alexander Mathis, Pranav Mamidanna, Kevin M Cury, Taiga Abe, Venkatesh N Murthy, Mackenzie Weygandt Mathis, and Matthias Bethge. Deeplabcut: markerless pose estimation of user-defined body parts with deep learning. *Nature neuroscience*, 21:1281–1289, 2018.
- [3] Talmo D Pereira, Nathaniel Tabris, Arie Matsliah, David M Turner, Junyu Li, Shruthi Ravindranath, Eleni S Papadoyannis, Edna Normand, David S Deutsch, Z Yan Wang, et al. Sleap: A deep learning system for multi-animal pose tracking. *Nature methods*, 19(4):486–495, 2022.
- [4] Alexander Mathis, Thomas Biasi, Steffen Schneider, Mert Yuksekgonul, Byron Rogers, Matthias Bethge, and Mackenzie W Mathis. Pretraining boosts out-of-domain robustness for pose estimation. In *Proceedings of the IEEE/CVF Winter Conference on Applications of Computer Vision*, pages 1859–1868, 2021.
- [5] Jessy Lauer, Mu Zhou, Shaokai Ye, William Menegas, Steffen Schneider, Tanmay Nath, Mohammed Mostafizur Rahman, Valentina Di Santo, Daniel Soberanes, Guoping Feng, Venkatesh N. Murthy, George Lauder, Catherine Dulac, Mackenzie W. Mathis, and Alexander Mathis. Multi-animal pose estimation, identification and tracking with deeplabcut. *Nature Methods*, 19:496 – 504, 2022.
- [6] Hang Yu, Yufei Xu, Jing Zhang, Wei Zhao, Ziyu Guan, and Dacheng Tao. Ap-10k: A benchmark for animal pose estimation in the wild. In *Thirty-fifth Conference on Neural Information Processing Systems Datasets and Benchmarks Track (Round 2)*, 2021.

- [7] Eldar Insafutdinov, Leonid Pishchulin, Bjoern Andres, Mykhaylo Andriluka, and Bernt Schiele. DeeperCut: A deeper, stronger, and faster multi-person pose estimation model. In *European Conference on Computer Vision*, pages 34–50. Springer, 2016.
- [8] MMPose Contributors. Openmmlab pose estimation toolbox and benchmark. <https://github.com/open-mmlab/mmpose>, 2020.
- [9] Jinkun Cao, Hongyang Tang, Hao-Shu Fang, Xiaoyong Shen, Cewu Lu, and Yu-Wing Tai. Cross-domain adaptation for animal pose estimation. In *Proceedings of the IEEE/CVF International Conference on Computer Vision*, pages 9498–9507, 2019.
- [10] Yufei Xu, Jing Zhang, Qiming Zhang, and Dacheng Tao. Vitpose+: Vision transformer foundation model for generic body pose estimation. *ArXiv*, abs/2212.04246, 2022.
- [11] Jennifer J Sun, Serim Ryou, Roni H Goldshmid, Brandon Weissbourd, John O Dabiri, David J Anderson, Ann Kennedy, Yisong Yue, and Pietro Perona. Self-supervised keypoint discovery in behavioral videos. In *Proceedings of the IEEE/CVF Conference on Computer Vision and Pattern Recognition*, pages 2171–2180, 2022.
- [12] Praneet Bala, Jan Zimmermann, Hyun Soo Park, and Benjamin Y Hayden. Self-supervised secondary landmark detection via 3d representation learning. *International Journal of Computer Vision*, pages 1–15, 2023.
- [13] Shaokai Ye, Jessy Lauer, Mu Zhou, Alexander Mathis, and Mackenzie Weygandt Mathis. Amadeusgpt: a natural language interface for interactive animal behavioral analysis. *Thirty-seventh Conference on Neural Information Processing Systems*, 2023.
- [14] Caleb Weinreb, Mohammed Abdal Monium Osman, Libby Zhang, Sherry Lin, Jonah E Pearl, Sidharth Annapragada, Elizabeth Conlin, Winthrop F. Gillis, Maya Jay, Shaokai Ye, Alexander Mathis, Mackenzie W. Mathis, Talmo D. Pereira, Scott W. Linderman, and Sandeep Robert Datta. Keypoint-moseq: parsing behavior by linking point tracking to pose dynamics. *bioRxiv*, 2023.

REVIEWER COMMENTS

Reviewer #1 (Remarks to the Author):

The authors have carefully addressed all my concerns, and their responses to the other reviewers and the additional improvements on the model seem thoughtful and thorough (although I cannot assess some of the technical details). My lab did not have the time to test model performance during the re-review process, as there simply wasn't enough time. However, the corresponding author has an excellent track-record in producing software solutions (incl. documentation, updates and outreach activities) that lead to widespread uptake by the community, and some of their work has revolutionized the field already. It is this process that will also decide about the power and usefulness of SuperAnimal. I am strongly in favor of publishing this manuscript in Nature Communications and in letting the scientific community determine the impact of this work over the months and years to come.

Reviewer #2 (Remarks to the Author):

I thank the authors for their detailed response to the first round of comments; the revised manuscript makes it much easier to understand what they have done. However, I still have concerns that some claims in the manuscript are not well supported by the results presented. I also feel the paper is lacking comparisons between the methods used in the SuperAnimal model and previous methods that have addressed some of the same problems.

Major points:

New model performance on iRodent: I thank the authors for taking my critique of the Quadruped model into consideration. I'm confused about Figure 5 in their rebuttal letter, however: they report a 10x performance improvement on iRodent, but strangely this 10x improvement is present not just for the SuperAnimal models but also for ImageNet—the previous normalized error of ImageNet at 3 fine-tuning images was ~ 2.4 , whereas it's 0.2 in the new version. Shouldn't the ImageNet performance between the two model versions be unchanged, given that it's the "control"? What happened?

Usability of the Quadruped model: more broadly, I remain skeptical of the utility of the Quadruped model as a zero-shot tool. The zero-shot performance of the model is still pretty poor, for example ~ 0.6 normalized error on Horse-10 is twice the threshold for a keypoint match in the original benchmark (which was 0.3). And a mAP of 0.4 on iRodent is also very low. Training the Quadruped model on labeled

frames does improve performance with fewer examples than is needed by ImageNet, however there is a significant difference between a model that is data-efficient and a zero-shot pose estimation model, and SuperAnimal-Quadruped is still being presented in the text as “an efficient way to achieve strong zero-shot performance.” This claim is not supported by the results shown.

Informativeness of metrics used: The authors use two unsupervised methods to deal with domain shifts: spatial-pyramid search and video adaptation. The performance of the spatial-pyramid search is evaluated in terms of the RMSE, which is fine (though figure 3C is quite hard to read.) But quantitative evidence that video adaptation improves model performance is still limited. The convex hull metric shows a decrease in the number of “off-body” errors where keypoints are placed far from the animal but doesn’t provide enough information to know whether a pose estimate is reasonable. The keypoint dropping and jitter metrics in figure S4e show that video adaptation makes pose estimates modestly smoother (though significance testing is needed), but smoother keypoint trajectories are not necessarily more accurate. The only way to really know whether this method is producing an improved accuracy of pose estimates would be to evaluate RMSE or mAP on a held-out set of human-annotated frames.

Relationship to prior work: First continuing on the subject of video adaptation, it is not clear from the manuscript to what extent the authors’ methods are an improvement on other approaches that have been previously proposed. The authors should also at the very least test whether their video adaptation approach is any better than more classical time series denoising methods such as Kalman filtering. A better evaluation would also contrast performance with other algorithms that have used temporal smoothness constraints to improve pose estimation quality, such as DeepGraphPose (Wu et al., NeurIPS 2020). Finally, given that the authors are not the first group to try using pseudolabels to boost model performance, it would be good to see a comparison to other methods such as Progressive Pseudo-label-based Optimization (PPLO, Cao, Tang et al., ICCV 2019) which is particularly relevant here given that this method was also introduced in the context of quadruped pose estimation.

Consistency between quantitative results and what is claimed: the plots of model performance in Figures 1-2 suggest that different models/methods perform better on different datasets, however the authors don’t always take this into account when reporting their conclusions. For example, SA + memory replay performs best on the DLC-Openfield dataset, leading the authors to conclude in the text that memory replay boosts model performance- however in the Horse-10 and iRodent datasets, memory replay does worse than naïve fine-tuning. In Horse-10, SA + memory replay in fact does even worse than the SA zero-shot model at 14 fine-tuning images, suggesting that training the model on labels from the Horse-10 dataset somehow worsens its performance. The actual takeaway from all of this may simply be that different model settings work best for different datasets- but this is a hard conclusion for the reader to draw when results on different datasets are spread across multiple figures and reported using different metrics.

Plotting of tracking data: In the behavioral analysis of Horse-30 data, the authors write in the method that trajectories were smoothed using a 2nd-order, low-pass, zero-lag Butterworth filter for subsequent stride detection. This is fine, however I think given that the filtering is presumably required for the performance reported in Figure 4h-i, it would be helpful to see both the raw and the filtered tracking data in Figure 4g, to give a sense of how much the filtering cleans up the raw trajectories. (I also wasn't completely sure which version of the data is being shown in 4g.)

Clarity of the method description: I had trouble figuring out from the text and from Figure 1c which parts of the network are trained during "task-aware fine-tuning", and also which parts of the network cartoon in 1c were being called the encoder and the decoder. My guess in 1c is that the orange blocks labeled "pre-training dataset" are what is called the "encoder" in the text, and the purple blocks labeled "downstream dataset" are the "decoder", is this right? I was debating between this and interpreting the first two (contracting) layers are the encoder and the last two (expanding) layers as the decoder. Further confusing is that the text describes "combined encoder-decoder fine-tuning", which I would take to mean all layers of the network are trained, which is not what the color-coding in the figure is suggesting.

Metrics and comparison models: I still hold that it's unnecessarily confusing that the way models are evaluated, and the models they are compared to, changes across figures/datasets: Fig 1e and 3c (DLC-Openfield, Smear Lab, Golden Lab, and MausHaus) report RMSE, 1h and 2d-f (DLC-openfield, trimouse, iRodent, AP-10K, and AnimalPose) report mAP, and Fig 2c (Horse-10) reports normalized error. Of these, RMSE and normalized error are metrics where smaller numbers mean better performance, while for mAP higher numbers mean better performance. The number of fine-tuning images also differs across figures. The paper would be a lot easier to draw conclusions from with addition of a unified "results" table that reports performance for a common set of models and training set sizes, using one or more common performance metrics, on each dataset. Similarly, AP-10K is presented as a point of comparison in some figures but not others, and some figures report performance of "ImageNet + Randomly Initialized Decoder" while others show "ImageNet Fine-tuning". (I assume the latter two are different given that Figure 2C shows very different performances for SA + R.I.D. vs SA + Fine-tuning.) The thing being measured in Figure 1e, 1h, 2c-f, and 3c is always the same, namely pose estimation accuracy, therefore the metrics and control models being used should also be consistent.

Minor notes

Double-check wording in some parts of the Methods- for example: "A batch size of 8 was used and the SuperAnimal-TopViewMouse and were trained for a total of 750k iterations, respectively."

Table 1 and Figure S2 (possibly elsewhere?) change "thai"->"thigh" (eg "front_left_thai").

Methods section HRNet-w32: it's fine that the ED figures and main text use different train/test splits but given that they're two splits of the same dataset they are not "fully independent replication[s]" as claimed here. This point isn't critical to your claims though so you should be fine just cutting it.

Plotting results: the line graphs in Figure 1 and 2 use what looks like a log scale on the x axis, but the spacing of x-axis tick marks isn't quite consistent with that. For example, 17 is not at the midpoint of 3 and 35 on a log scale. I suggest making x-coordinates of plotted data precise to make these graphs easier to interpret.

Reviewer #3 (Remarks to the Author):

The authors have made significant improvements to the manuscript, with extensive additional details on the methodology, clarifications on the datasets and models used, and a few new evaluation results.

That said, issues remain related to the nature of this work as a foundation model for scientific application. Namely, we feel the authors must meet minimum standards for reporting and data access. Furthermore, without providing the datasets in the form used for evaluations here, the authors may be misconstrued as intentionally making it more difficult for others to build competing approaches and make fair and direct comparisons – something which they benefit from doing here by using other publicly available datasets.

Other concerns about the presentation and claims about the significance and utility of the approach are outlined below.

Issues:

>> "If the models need fine-tuning, we show SuperAnimal models are 10× more data efficient" (abstract)

This is a very misleading statement. The SuperAnimal models achieve the same performance at 10 fine-tuning frames as an ImageNet-pretrained model at ~100 fine-tuning frames. This difference disappears at ~500 fine-tuning frames, which is also the best performance overall. A more accurate portrayal of the work presented here would state that "If the models need fine-tuning, we show SuperAnimal models are

more data efficient at small training set sizes (10-100 frames) and achieve comparable performance at larger training set sizes (>500 frames)."

Importantly, no mention is made of the results showing that all improvements yielded by this work disappear at 500 fine-tuning frames as per Fig. 1e.

> Fig 3: a comparison to a standard behavior segmentation benchmark, such as Sun et al 2022 (arXiv:2207.10553), would provide stronger and more readily comparable evidence for good performance on this downstream task.

>> Sturman et al is an established human expert annotated benchmark in the field of mouse behavior. Sun et al. has programmatically defined behaviors, not ground truth annotations. The main goal of our paper was to show the innovations in building foundational-like models for pose estimation. Beyond Sturman et al. we now include another downstream behavioral analysis for stride analysis (Figure 4).

The MABe22 dataset in Sun et al. (arXiv:2207.10553) is a much larger and much more community-tested benchmark dataset for behavioral segmentation, which includes a broader and more representative set of behavioral classes. These are NOT programmatically defined, but rather expert annotated and have served as an excellent and widely adopted reference for testing precisely what the authors purport to show here.

An alternative source of evaluation of downstream performance would include use with the openly available SimBA videos and pretrained classifiers (<https://osf.io/tmu6y/>), which come with DLC tracked data to begin with and should be straightforward to evaluate on.

Downstream evaluations (such as in supervised behavior segmentation), while not the innovations the authors intend to showcase here, are a crucial and important measure as they provide evidence of the real world performance of the approach. Importantly, they afford a fair and orthogonal evaluation of the performance of the tracking models by measuring the functional significance of the claimed improvements. Given their importance, it is not unreasonable to request a more thorough evaluation of downstream performance on more representative behavior segmentation datasets like the two mentioned above.

>> Additionally, we note that SuperAnimal weights were used in another pre-print, showing that SuperAnimal-TopViewMouse predictions with keypointMoSeq [13] outperform other behavioral quantification methods on a separate benchmark dataset. For ease of the reviewers, we added that relevant figure panels here (Fig. 9). We cite this work as well. Additionally, we used the SuperAnimal-

TopViewMouse model within AmadeusGPT [14] for several other mouse topview datasets, and for example, could show on the elevated plus maze it can match human-annotation performance there as well.

The Keypoint-MoSeq paper does not demonstrate that pose tracking with SuperAnimal-TopViewMouse outperforms pose tracking with any alternative approach – it evaluates the performance of different unsupervised behavior segmentation methods given the same tracking. Considering the amount of outlier filtering and other forms of robustness to keypoint noise specific to that method, I do not see this as a valid evaluation of the work in question here – maybe beside the point since none of that data is presented in THIS manuscript.

There are a number of issues with the work presented in the AmadeusGPT preprint, all of which are outside the scope of this paper's review. If the authors wish to use it as evidence for the performance or validity of the current work, I encourage them to do so by including data on downstream behavior classification performance.

> Since the model reported in this study is intended to be used directly for scientific applications, it is important that the authors share sufficient detail about the models, so that use applications do not suffer from any biases that might be baked into these models. I suggest providing the following (though the authors may have additional details they would like to provide): Dataset datasheet, as described in Gebru et al 2021 (arXiv:1803.09010) Crowdsourced annotation datasheet, as described in Diaz et al 2022 (arXiv:2206.08931) Model cards, as described in Mitchell et al (arXiv:1810.03993)

>> We updated the model cards at HuggingFace. TopViewMouse: <https://huggingface.co/mwmathis/DeepLabCutModelZoo-SuperAnimal-TopViewMouse>. SuperAnimal-Quadruped DLCRNet: <https://huggingface.co/mwmathis/DeepLabCutModelZoo-SuperAnimal-Quadruped>

The efforts made to improve the documentation of the model is appreciated, but insufficient given the potential for widespread use of the work presented here.

Minimum requirements include:

1. A dataset datasheet should be provided as described in Gebru et al. (arXiv:1803.09010) including sections: Motivation, Composition, Collection Process, Preprocessing/Cleaning/Labeling, Uses, Distribution, and Maintenance, as well as associated sub-sections. Appendix A in (arXiv:1803.09010) has a clear example.

2. A model card should be provided as described in Mitchell et al. (arXiv:1810.03993) including sections: Model Details, Intended Use, Factors, Metrics, Training Data, Evaluation Data, Ethical Considerations, Caveats and Recommendations. Figs 2 and 3 in (arXiv:1810.03993) have clear examples.

These are the standard for foundation models which the authors state they consider this work to be. For a recent example, see Segment Anything (arXiv:2304.02643) which takes the exact steps described above (Appendix F) to ensure responsible and ethical use of their foundation model. As the intended use case for this work is the scientific domain, it would be appropriate to apply even more stringent reporting requirements, so this is not a particularly high bar to meet.

> The authors state in the Reporting Summary that "In the following we detail references for those datasets", but these are not provided. I believe a major contribution of this study is a unified SuperAnimal dataset - the authors should provide this as part of the resource - the dataset should be made publicly available. Another major contribution is a unified vocabulary for animal keypoints, but this is only provided through figure illustrations rather than machine parseable text or code to do the dataset unification.

>> The datasets are referenced fully in the Methods and in Suppl. Figure 1. Please note, not all of the mouse data can be made available due to limitations on institutional guidelines on sharing recorded data. Therefore, while we used primarily open source data, as we already referenced, those that are given to us courtesy of other scientists must stay private. The best we can do, as we do here, is to provide model weights. This is also an important point for scientists who might be hesitant to release their data. We also now added the unified keypoint main diagram to the figure repository so users understand the mapping, and a new panel in Extended Data 1 that has the mappings.

While references and ED1 are helpful, there is significant additional processing that the authors have done, many steps of which cannot presently be reproduced even with the descriptions provided.

The authors can also:

1. Provide the code for standardizing the pose annotations for the specific datasets used here.
2. Provide the publicly available datasets in their standardized format (or at a minimum, the pose annotations).

The model weights encode the biases and structural sources of error present in the source annotations, but these are intractable to audit without access to the source data and annotations.

These datasets that cannot be submitted to scrutiny, but which constitute the core of the contributions reported here, can simply be excluded from the training set to create an auditable subset. Both versions of the weights could be provided, with ample disclaimers that one model was trained on private and potentially problematic source data.

>> This app allows anyone, within their browser, to a) upload their own image and label, b) annotate community images, c) run inference of available community models on their own data, d) share models to be hosted.

The app does not appear to allow for uploading user images for annotation, and other than a Google Form, nothing appears to be present in the web app other than the curated images for public datasets. It is not clear how any of this, other than the inference functionality (also available via HuggingFace Spaces) pertains to the work presented here. If we understand correctly, the labels on the curated datasets are not used in this work.

REVIEWER COMMENTS

Reviewer #1 (Remarks to the Author):

The authors have carefully addressed all my concerns, and their responses to the other reviewers and the additional improvements on the model seem thoughtful and thorough (although I cannot assess some of the technical details). My lab did not have the time to test model performance during the re-review process, as there simply wasn't enough time. However, the corresponding author has an excellent track-record in producing software solutions (incl. documentation, updates and outreach activities) that lead to widespread uptake by the community, and some of their work has revolutionized the field already. It is this process that will also decide about the power and usefulness of SuperAnimal. I am strongly in favor of publishing this manuscript in Nature Communications and in letting the scientific community determine the impact of this work over the months and years to come.

We thank Reviewer 1 for supporting our work.

Reviewer #2 (Remarks to the Author):

I thank the authors for their detailed response to the first round of comments; the revised manuscript makes it much easier to understand what they have done. However, I still have concerns that some claims in the manuscript are not well supported by the results presented. I also feel the paper is lacking comparisons between the methods used in the SuperAnimal model and previous methods that have addressed some of the same problems.

Major points:

New model performance on iRodent: I thank the authors for taking my critique of the Quadruped model into consideration. I'm confused about Figure 5 in their rebuttal letter, however: they report a 10x performance improvement on iRodent, but strangely this 10x improvement is present not just for the SuperAnimal models but also for ImageNet—the previous normalized error of ImageNet at 3 fine-tuning images was ~2.4, whereas it's 0.2 in the new version. Shouldn't the ImageNet performance between the two model versions be unchanged, given that it's the "control"? What happened?

Thanks for the feedback. We'd like to take the opportunity to step back. When we originally submitted, our goal was to highlight a method to build "super-sets" of animal models (so called SuperAnimal models) that could be used for building better base (foundation) models that could replace ImageNet pre-trained weights. Over the last 9 months of us intensely working to build this, we have improved the models, and the method. Therefore, what we present in this second revision is now much better models, showing our zero-shot SuperAnimal-Quadruped and SuperAnimal-TopViewMouse models are **on par with fully supervised models**. Thus, now our paper is two things: (1) the method to do this, (2) the foundation models to use.

Therefore, we want to highlight our new results, and how we achieved them.

- We would like to clarify that in our paper we used ImageNet weights or Quadruped40K or MouseTopView5K, respectively, and THEN we added our SuperAnimal method (SA) training method on top. Therefore, when we improve the SA (i.e., better augmentation, etc, the “baseline” ImageNet changes to. Our goal was to show that SuperAnimal weights are better than ImageNet weights, especially for real-world sized data. This nicely follows our prior work on transfer learning (WACV 2021) and Kaming He’s work on transfer learning (2019).
- “Shouldn’t the ImageNet performance between the two model versions be unchanged, given that it’s the “control”? What happened?” We documented in our first revision and shown in Figure 3, we had switched from bottom-up model to using a top-down model, which boosted performance by ~ 30-50 mAP, depending on the benchmark (see Figure 3b-e).

Moreover, to simplify comparisons, now in v3 we show, again in a unified Table as requested: **Zero-shot** performance of our SA-Quadruped on **4 Benchmarks**:

- **Note, the best model achieves a 72.971 mAP on iRodent.**

Table S2. Main Results on Quadruped Benchmarks. Here, the base SuperAnimal-Quadruped model had none of the held-out datasets. Full results can be found in Figure 2 for fine-tuning with different amounts of data, but the best fine-tuning performance is shown, which matches the top-performance of the SuperAnimal (SA) variant as shown in Figure 2. *NOTE: Cao et al.(33) do not report a unified single mAP, rather per animal, therefore we trained a model using their dataset to estimate top-line performance if only trained on AP. **Number as reported in (41) using the data from (31).

Method	Pre-trained Weights	Data Ratio	mAP	RMSE	Dataset	NE_IID	NE_OOD	Architecture
zero-shot	SuperAnimal	-	68.038	12.971	AP-10K	-	-	HRNetw32
zero-shot	SuperAnimal	-	66.110	12.849	AP-10K	-	-	AnimalTokenPose
transfer learning	ImageNet	1.00	70.548	11.228	AP-10K	-	-	HRNetw32
memory replay	SuperAnimal	1.00	80.113	11.296	AP-10K	-	-	HRNetw32
zero-shot	AP-10K	-	79.447	5.774	AnimalPose	-	-	HRNetw32
zero-shot	SuperAnimal	-	84.639	4.884	AnimalPose	-	-	HRNetw32
zero-shot	SuperAnimal	-	83.043	5.154	AnimalPose	-	-	AnimalTokenPose
transfer learning	ImageNet	1.00	86.864	5.757	AnimalPose	-	-	HRNetw32
fine-tuning	AP-10K	1.00	86.794	4.860	AnimalPose	-	-	HRNetw32
memory replay	SuperAnimal	1.00	87.034	4.636	AnimalPose	-	-	HRNetw32
zero-shot	AP-10K	-	65.729	4.929	Horse-10	0.296	0.287	HRNetw32
zero-shot	SuperAnimal	-	71.205	3.958	Horse-10	0.227	0.228	HRNetw32
zero-shot	SuperAnimal	-	68.977	4.081	Horse-10	0.239	0.233	AnimalTokenPose
transfer learning	ImageNet	0.01	0.934	46.255	Horse-10	2.369	2.36	HRNetw32
transfer learning	ImageNet	1.00	90.516	1.837	Horse-10	0.036	0.135	HRNetw32
fine-tuning	AP-10K	0.01	66.284	5.029	Horse-10	0.286	0.285	HRNetw32
fine-tuning	AP-10K	1.00	93.973	1.220	Horse-10	0.036	0.083	HRNetw32
memory replay	SuperAnimal	0.01	73.366	3.719	Horse-10	0.209	0.202	HRNetw32
memory replay	SuperAnimal	1.00	95.165	1.153	Horse-10	0.040	0.073	HRNetw32
zero-shot	AP-10K	-	40.389	37.417	iRodent	-	-	HRNetw32
zero-shot	SuperAnimal	-	58.557	33.496	iRodent	-	-	HRNetw32
zero-shot	SuperAnimal	-	55.415	34.666	iRodent	-	-	AnimalTokenPose
transfer learning	AP-10K	0.01	12.910	92.649	iRodent	-	-	HRNetw32
transfer learning	ImageNet	0.01	0.785	152.225	iRodent	-	-	HRNetw32
transfer learning	ImageNet	1.00	58.857	35.651	iRodent	-	-	HRNetw32
fine-tuning	AP-10K	0.01	43.144	37.704	iRodent	-	-	HRNetw32
fine-tuning	AP-10K	1.00	61.635	26.758	iRodent	-	-	HRNetw32
memory replay	SuperAnimal	0.01	60.853	31.801	iRodent	-	-	HRNetw32
memory replay	SuperAnimal	1.00	72.971	24.884	iRodent	-	-	HRNetw32

Usability of the Quadruped model: more broadly, I remain skeptical of the utility of the Quadruped model as a zero-shot tool. The zero-shot performance of the model is still pretty poor, for example ~0.6

normalized error on Horse-10 is twice the threshold for a keypoint match in the original benchmark (which was 0.3). And a mAP of 0.4 on iRodent is also very low. Training the Quadruped model on labeled frames does improve performance with fewer examples than is needed by ImageNet, however there is a significant difference between a model that is data-efficient and a zero-shot pose estimation model, and SuperAnimal-Quadruped is still being presented in the text as “an efficient way to achieve strong zero-shot performance.” This claim is not supported by the results shown.

We extensively re-did every experiment in the paper now with better models (see above), and we are confident this is a usable model. We also now achieve SOTA on Horse-10.

We are happy to clarify our text, but also feel we now clearly demonstrate the usability of SA-Quadruped; revised: “*an efficient way to achieve strong zero-shot **and few-shot** performance*”. And also note now our model is as good zero-shot as fully supervised.

Moreover, we added 3 examples with SA-Quadruped in the last round of revisions for zero-shot: the dog, the elk, and the horse, and for the horse quantified that with zero-shot and minimal smoothing of data we are on-par with human-level counts of actions. We did this to show usability of the models.

We would like to add, we agree we did not solve pose estimation in this one paper, but we strongly feel we are adding a lot of value to the community by providing a new framework to train better base-models that can be broadly used. Whether that is zero-shot (amazing), or few-shot (we also think that's pretty great). We also added a comparison with other related works (even ones that came out this fall) on the AP-10K benchmark, which shows we use much less pre-training data and much less parameters to achieve competitive results.

AP-10K Benchmark

To summarize what improvement we made, which can be found in the Methods and Results:

A. Scaling the data size

1) Previously, we had 3 OOD datasets (Horse-10, iRodent, AP-10K) that were never included in the base Quadruped datasets; we now follow a leave-one-out strategy for all OOD datasets. For example, the experiments for iRodent uses a model that is pretrained on all quadruped datasets besides iRodent, thus expanding the images in the pretrained datasets.

2) Previously, we used 80% images of the pretrain dataset for training, leaving 20% of the images for validation. Since our goal is to pretrain the best foundation model we can for you, we decided to use 100% images of the pretrain dataset to pretrain the model and test on OOD as previously.

3) More data: we now add AP-10K and APT-36K into our base model pretrain datasets (unless held-out for testing, of course), expanding our Quadruped40K to Quadruped80K. Note APT-36K is also used by both VitPose++ and UniPose as the pre-training data.

B. Better train and test augmentation

We add random horizontal flip augmentation and flip test into the train test pipeline as they are common practice in previous papers (AP-10K (Yu et al, 2021) and VitPose (Xu et al. 2022)).

New Results:

Performance on Horse-10:

First, we want to note that 0.3 normalized error (NE) from the original Horse-10 paper is from a fully supervised trained model while our 0.6 came from our model's zero-shot predictions. However, we do believe our zero-shot performance should be able to outperform the 0.3 OOD NE baseline from the original paper.

Therefore, we carefully examined Horse-10 data, and we identified two issues:

- 1) We found a difference in the calculation of the NE implementation between ours and the original paper. Matching theirs results in a change from 0.6 to 0.4: again, this is zero-shot.
- 2) We found two keypoints mismatched in our manual conversion table when we map Quadruped data keypoint space to original Horse-10's keypoints. This mismatch impacts both zero-shot and fine-tuning performance. Fixing both issues gives us zero-shot performance of 0.25 OOD NE with Quadruped40K and **0.23 OOD NE with Quadruped80K**, which is better than the fully supervised baseline in the original paper, and our fine-tuning gives 0.07 NE.

Performance on iRodent:

iRodent is a hard dataset. To better argue our point, and show you why we made this benchmark, here is the distribution of animal sizes in the dataset compared to others; notably, while the COCO human benchmark was the dominant benchmark used to evaluate different pose estimation methods, we notice it has much less variance in the object (person) sizes within the dataset. Therefore, it's likely that this is commonly overlooked when there is a distribution shift for the size of the object, and adds to the challenges of iRodent. We added a new Extended Data Figure on the dataset, added here for ease:

Therefore, we believe that the best way to judge the model performance is by comparing it to the ImageNet+SA method (~58.9 mAP) baseline instead of looking at the absolute number.

After the improvements we detailed above, the new **zero-shot performance** in iRodent is 58.6 mAP, which is comparable to the fully supervised trained model with HRNet-w32 using ImageNet weights. Additionally, fine-tuning our SuperAnimal weights can give us 73.0 mAP, bringing an additional 14.1 mAP gain compared to ImageNet baseline. We hope our newest SA-Quadruped models meet your expectations for good utility.

Besides the absolute numbers, we also want to note the limitation of using mAP to evaluate a zero-shot pose model. As we noted in Supplementary Materials previously, there is always annotator bias between the model’s keypoints and the target domain keypoints, especially on body keypoints that are difficult for different annotators to agree on, and we argue it’s causing an annotator bias between our pretrain datasets and the target domain dataset. Therefore, while we now have more competitive zero-shot performance, we still expect this is an under-estimation of the true effectiveness of the model.

Performance on AnimalPose:

Our zero-shot mAP is 84.6, 5 mAP higher than AP-10K’s 79.4 zero-shot. Ours is very close to the fully supervised ImageNet baseline that has 86.8 mAP.

Performance on AP-10K:

We now have 80.1 mAP on the AP-10K, which is now ranked third place in the formal leaderboard. Note the Top1 and Top2 have 82.4 mAP and 80.4 mAP. However, their ViTPose+H (632 M parameters) and ViTPose+L (307 M parameters) are 20 times and 10 times bigger than our HRNetw32 (29M parameters) model, thus imposing a hard hardware requirement to end users. In the meantime, their ViTPose+B model (86 M parameters), which is 2 times bigger than ours, has 74.5 mAP, which is 5.6 mAP lower than ours. We bring a 7 mAP gain in HRNetw32 that uses ImageNet weights.

Informativeness of metrics used: The authors use two unsupervised methods to deal with domain shifts: spatial-pyramid search and video adaptation. The performance of the spatial-pyramid search is evaluated in terms of the RMSE, which is fine (though figure 3C is quite hard to read.) But quantitative evidence that video adaptation improves model performance is still limited. The convex hull metric shows a decrease in the number of “off-body” errors where keypoints are placed far from the animal but doesn’t provide enough information to know whether a pose estimate is reasonable. The keypoint dropping and jitter

metrics in figure S4e show that video adaptation makes pose estimates modestly smoother (**though significance testing is needed**), but smoother keypoint trajectories are not necessarily more accurate. **The only way to really know whether this method is producing an improved accuracy of pose estimates would be to evaluate RMSE or mAP on a held-out set of human-annotated frames.**

We would push back a little to say that keypoint dropping does give a sense of performance, and we've added statistics to show that this is statistically significant, with small to large effect sizes. See Supplemental Tables for extensive testing (also linked in Results).

But nonetheless, we added a new analysis: since Horse-30 (the full dataset) has dense annotations for 30 horse videos, we decided to use this dataset to test how video adaptation adds to the zero-shot performance in terms of mAP, as you ask. See below, where we add a requested baseline as well. In short, it does improve mAP.

Relationship to prior work: First continuing on the subject of video adaptation, it is not clear from the manuscript to what extent the authors' methods are an improvement on other approaches that have been previously proposed. The authors should also at the very least test whether their video adaptation approach is any better than more classical time series denoising methods such as Kalman filtering. A better evaluation would also contrast performance with other algorithms that have used temporal smoothness constraints to improve pose estimation quality, such as DeepGraphPose (Wu et al., NeurIPS 2020). Finally, given that the authors are not the first group to try using pseudolabels to boost model performance, it would be good to see a comparison to other methods such as Progressive Pseudo-label-based Optimization (PPLO, Cao, Tang et al., ICCV 2019) which is particularly relevant here given that this method was also introduced in the context of quadruped pose estimation.

As requested, we compare video adaptation to two baselines; PPLO, as via progressive pseudo labeling, and a Kalman Filter.

We didn't claim video adaptation being technically novel, instead we find its a highly practical approach to adapt a pose model to videos (which, to our best knowledge, has not been done before). Unlike many domain adaptation algorithms (Cao et al. 2019), **our video adaptation does not require access to source domain data or annotated target domain data.** It's not practical to expect our end-users to download our ever-growing source datasets or expect the users to have annotations when they simply want to run inference on a video without any annotations.

For the same reason, we choose not to compare to DeepGraphPose, because it adds a graphical model as regularizer on top of the pose predictor, thus requiring access to annotated data in both source and target domain. It thus is not a suitable comparison to our unsupervised-based video adaptation which does NOT require access to the source dataset at all.

We also didn't compare to the full algorithm of progressive pseudo-labeling because the full version requires access to annotations of both source domain and target domain. We thus only keep the "pseudo labeling with a curriculum" (or they call it, self-pacing pseudo labeling) part of PPLO to compare to video adaptation.

As far as we know, most pseudo-labeling algorithms — including PPLO — were tested in static image datasets where images are diverse. In contrast, to the best knowledge, we are the first to use it to adapt a pre-trained pose model to videos where frames are mostly similar. To that end, we observed that it's critical to fix the running stats of batch norms (BNs) during the adaptation training, as the running

statistics collected from frames of one single video can be inaccurate and prone to overfitting. This is not described in previous pseudo-labeling methods. And we found pseudo-labeling without fixing BN's running stats is harmful and prone to overfitting.

We use two new evaluation metrics (adding to jittering) called adaptation gain (delta mAP) and robustness gain, that help quantify video-based adaptation. This is added to Figure 3.

- Adaptation gain: denotes the adapted model's change in mAP on the adapted video. A negative number means a performance degradation after adaptation.
- Robustness gain: Different from adaptation gain that calculates the mAP on a single video, we calculate mAP gain on all videos of Horse-30. This helps to identify whether the model overfits one single video it trains on or it performs successful domain adaptation with respect to the whole video dataset (30 videos). This is now added to Figure 3.

The metrics below are reported as median [first and third quartiles]

Adaptation gain

Kalman: -4.03 [-25.53, -2.25]

self-pacing: 0.13 [-0.73, 0.78]

video adaptation: 1.16 [0.28, 2.40]

Robustness gain across videos (thus no Kalman, as it is a per-video method):

self-pacing: 0.46 [-0.30, 1.11]

video adaptation: 4.16 [3.58, 4.89]

We calculated the adaptation gain for both PPLO and video adaptation. We show that video adaptation gives an increase of 1.16 mAP in adaptation gain and 4.16 mAP gain in robustness gain. In contrast, PPLO gives close to 0.13 mAP increase in adaptation again and 0.46 mAP increase in robustness. The Kalman filter gives -4 mAP in adaptation. This is aligned with R2's intuition that smoothness does not necessarily bring a performance gain. Unlike video adaptation that changes models' weights, the Kalman filter is a per-video post-processing method and robustness gain is not applicable.

We believe video adaptation is better than PPLO due to two reasons:

- 1) The iterative pseudo labeling of PPLO causes the model to overfit to the video it adapts to.
- 2) PPLO does not take into account that the running stats of BN from video frames give inaccurate running stats while video adaptation uses the pre-trained model's running stats in BN.

Consistency between quantitative results and what is claimed: the plots of model performance in Figures 1-2 suggest that different models/methods perform better on different datasets, however the authors don't always take this into account when reporting their conclusions. For example, SA + memory replay performs best on the DLC-Openfield dataset, leading the authors to conclude in the text that memory replay boosts model performance- however in the Horse-10 and iRodent datasets, memory replay does worse than naïve fine-tuning. In Horse-10, SA + memory replay in fact does even worse than the SA zero-shot model at 14 fine-tuning images, suggesting that training the model on labels from the Horse-10 dataset somehow worsens its performance. The actual takeaway from all of this may simply be that different model settings work best for different datasets- but this is a hard conclusion for the reader to draw when results on different datasets are spread across multiple figures and reported using different metrics.

As you asked in Revision 1, we already made a unified Summary Table so all results are easily accessible to the reader. We now also provide a more extensive table with our new updates.

Regarding which part of the SA method is best to use when, we note that the motivation of memory-replay is to overcome the catastrophic forgetting for a pre-trained pose model (without accessing the source domain data). So it's expected that there is a trade-off between keeping the previous knowledge and the new knowledge. We found increasing the threshold of pseudo-labeling (from the previous 0.3 to 0.7 for Quadruped80K) improves memory-replay's performance. In the updated results, now we show that memory-replay is always better or on par with naive fine-tuning. Additionally, we show in Extended Data Figure 4 that memory-replay mitigates catastrophic forgetting that is evaluated by keypoint dropping.

We can tone down the perceived claim that memory-replay is absolutely better in all situations, that was not what we meant to imply (and would have just not shown naive fine-tuning!).

Plotting of tracking data: In the behavioral analysis of Horse-30 data, the authors write in the method that trajectories were smoothed using a 2nd-order, low-pass, zero-lag Butterworth filter for subsequent stride detection. This is fine, however I think given that the filtering is presumably required for the performance reported in Figure 4h-i, it would be helpful to see both the raw and the filtered tracking data in Figure 4g, to give a sense of how much the filtering cleans up the raw trajectories. (I also wasn't completely sure which version of the data is being shown in 4g.)

We now replaced the images in Figure 4 with the raw data, and added the filtered variant to the Extended Data Figure 7. Also here for ease: (note with the new model it also improved the gait analysis R1 from 0.89 to 0.91.):

Raw GT (left) and predictions (right)

Smoothed GT (left) and predictions (right)

Clarity of the method description: I had trouble figuring out from the text and from Figure 1c which parts of the network are trained during “task-aware fine-tuning”, and also which parts of the network cartoon in 1c were being called the encoder and the decoder. My guess in 1c is that the orange blocks labeled “pre-training dataset” are what is called the “encoder” in the text, and the purple blocks labeled “downstream dataset” are the “decoder”, is this right? I was debating between this and interpreting the first two

(contracting) layers are the encoder and the last two (expanding) layers as the decoder. Further confusing is that the text describes “combined encoder-decoder fine-tuning”, which I would take to mean all layers of the network are trained, which is not what the color-coding in the figure is suggesting.

We updated Figure 1 and added a clear label which is encoder and which is decoder now, as this links to our RID or Fine-tuning the head aspect of the work.

Metrics and comparison models: I still hold that it’s unnecessarily confusing that the way models are evaluated, and the models they are compared to, changes across figures/datasets: Fig 1e and 3c (DLC-Openfield, Smear Lab, Golden Lab, and MausHaus) report RMSE, 1h and 2d-f (DLC-openfield, trimouse, iRodent, AP-10K, and AnimalPose) report mAP, and Fig 2c (Horse-10) reports normalized error.

Of these, RMSE and normalized error are metrics where smaller numbers mean better performance, while for mAP higher numbers mean better performance.

The number of fine-tuning images also differs across figures. The paper would be a lot easier to draw conclusions from with addition of a unified “results” table that reports performance for a common set of models and training set sizes, using one or more common performance metrics, on each dataset.

Similarly, AP-10K is presented as a point of comparison in some figures but not others,

and some figures report performance of "ImageNet + Randomly Initialized Decoder" while others show "ImageNet Fine-tuning". (I assume the latter two are different given that Figure 2C shows very different performances for SA + R.I.D. vs SA + Fine-tuning.)

The thing being measured in Figure 1e, 1h, 2c-f, and 3c is always the same, namely pose estimation accuracy, therefore the metrics and control models being used should also be consistent.

We still feel strongly it’s best to use the metrics that others have used before to be able to share comparable results - and we made the summary table already as you requested in Revision 1 that has all metrics such as RMSE, mAP and normalized error for all our results.

That being said, we agree we can simplify our presentation and tidy up our language, which we did and can be summarized in the following:

- In the fine-tuning experiments we ran, the # of images differ, but it’s a constant % of the data, as was done in prior works where we adopted benchmarks for fine-tuning comparisons. We changed the figures to read % of frames used and added the actual # in the figure caption.
- We changed Figure 1 to be only RMSE for TopViewMouse (which then is consistent with Figure 3 mouse plots), and all Quadruped data is reported as mAP and RMSE, and Horse-10 NE for benchmark reporting.
- We fixed the inconsistency in the wording of ImageNet randomly initialized decoder (RID) and Imagenet Fine-tuning (good catch they are the same, i.e., Figure 3e should say ImageNet-RID), But SA+RID and SA+Naive Fine Tuning ARE different; Fine-tuning means we keep the trained decoder head and fine-tune it, whereas RID means we initialize an untrained decoder and learn it.

Minor notes

Double-check wording in some parts of the Methods- for example: “A batch size of 8 was used and the SuperAnimal-TopViewMouse and were trained for a total of 750k iterations, respectively.”

- Fixed, thanks.

Table 1 and Figure S2 (possibly elsewhere?) change “thai”->“thigh” (eg “front_left_thai”).

- We agree this is annoying, but this is because the original source dataset used thai not thigh, so that is what is in the original dataset if someone does use it (we do not change the source data).

Methods section HRNet-w32: it’s fine that the ED figures and main text use different train/test splits but given that they’re two splits of the same dataset they are not “fully independent replication[s]” as claimed here. This point isn’t critical to your claims though so you should be fine just cutting it.

- Agree, we cut it.

Plotting results: the line graphs in Figure 1 and 2 use what looks like a log scale on the x axis, but the spacing of x-axis tick marks isn’t quite consistent with that. For example, 17 is not at the midpoint of 3 and 35 on a log scale. I suggest making x-coordinates of plotted data precise to make these graphs easier to interpret.

We tried a log scale plot previously and it didn’t render well as it tended to squeeze very small training data regime. We now report the training ratios, and hope this is an acceptable compromise.

Reviewer #3 (Remarks to the Author):

The authors have made significant improvements to the manuscript, with extensive additional details on the methodology, clarifications on the datasets and models used, and a few new evaluation results.

That said, issues remain related to the nature of this work as a foundation model for scientific application. Namely, we feel the authors must meet minimum standards for reporting and data access. Furthermore, without providing the datasets in the form used for evaluations here, the authors may be misconstrued as intentionally making it more difficult for others to build competing approaches and make fair and direct comparisons – something which they benefit from doing here by using other publicly available datasets.

I (the lead senior author) strongly push back on the assertion that we are not transparent. Our paper is not even published as we have made an absolutely good faith effort to (1) release already the models that support our results on HuggingFace, (2) release part of our new private data (on zenodo), (3) additionally use only open-source datasets which we did not alter in any way, (4) provide the public with multiple ways to use the models (GUI, HuggingFace, and stand-alone web interface).

To reiterate, the goal of our work was to introduce a method for building unified pre-trained pose-aware models. We show that even now –given the data limitations– we can achieve SOTA zero-shot performance against fully supervised models on hard animal pose benchmarks (Horse-10), and match

the performance of larger models trained with more data (AP-10K). I think this is a major advance in the field.

To remove any doubt, (1) we already released the models; (2) we fully intend to release all data. We include now Model Cards and Datasheets in the revision. Upon acceptance, we will release the final new datasets.

Other concerns about the presentation and claims about the significance and utility of the approach are outlined below.

Issues:

>> "If the models need fine-tuning, we show SuperAnimal models are 10x more data efficient" (abstract)

This is a very misleading statement. The SuperAnimal models achieve the same performance at 10 fine-tuning frames as an ImageNet-pretrained model at ~100 fine-tuning frames. This difference disappears at ~500 fine-tuning frames, which is also the best performance overall. A more accurate portrayal of the work presented here would state that "If the models need fine-tuning, we show SuperAnimal models are more data efficient at small training set sizes (10-100 frames) and achieve comparable performance at larger training set sizes (>500 frames)."

Importantly, no mention is made of the results showing that all improvements yielded by this work disappear at 500 fine-tuning frames as per Fig. 1e.

As the seminal work that studies transfer learning (Kaiming He et al 2019) suggested and is now common knowledge, pre-trained weights are less relevant if the target domain has sufficient data. The number of sufficient data depends on the difficulty of the dataset and we agree that for the dlc-openfield dataset, 500 frames is sufficient to make SuperAnimal pretrained weights less relevant, but users don't want to label 500 frames, or even 100 if they can label 10. Being able to have ~2 pixel error gain without more labeling even makes nature Methods papers, and a 1 mAP gain can make a CVPR paper. Thus, we are not being misleading at all: we state what is true for both SuperAnimal models.

Here, we also want to warmly remind you that you didn't consider our SA-Quadruped results where the tasks are clearly much harder. The newest results show impressive gains both zero-shot and after fine-tuning. In particular, we outperform fully trained models with ImageNet weights by 14.1 mAP in iRodent and 9.6 mAP in AP-10K. We are 5.6 mAP higher than the SOTA paper when we compare to models (ViT-Pose+ B) that are 2 times bigger than ours. As you likely know, this is considered a big jump in computer vision.

Those results show that our pretrained weights are very competitive and useful even in harder datasets, and in fact, with our new results, we are not just 10X more efficient, we can be up to 100X more.

> Fig 3: a comparison to a standard behavior segmentation benchmark, such as Sun et al 2022 (arXiv:2207.10553), would provide stronger and more readily comparable evidence for good performance on this downstream task.

>> *Sturman et al is an established human expert annotated benchmark in the field of mouse behavior. Sun et al. has programmatically defined behaviors, not ground truth annotations. The main goal of our*

paper was to show the innovations in building foundational-like models for pose estimation. Beyond Sturman et al. we now include another downstream behavioral analysis for stride analysis (Figure 4).

The MABe22 dataset in Sun et al. (arXiv:2207.10553) is a much larger and much more community-tested benchmark dataset for behavioral segmentation, which includes a broader and more representative set of behavioral classes. These are NOT programmatically defined, but rather expert annotated and have served as an excellent and widely adopted reference for testing precisely what the authors purport to show here.

Firstly, I (the lead senior author) want to make a critical point: MABe is not a standard behavior segmentation benchmark. This is a paper with < 20 citations, that has changing data. The evidence is in our favor that when we submitted our paper, there was no human annotated behavior data for Sun et al. 2022, and the methods section says it was programmatically annotated: I quote from their paper (<https://arxiv.org/abs/2207.10553v1>):

*"The MABe22 Mouse Triplets dataset was collected and analyzed in the laboratory of Vivek Kumar at Jackson Labs (JAX), and was assembled by Ann Kennedy at Northwestern University. Mice were bred and videos of interacting mice were collected by Tom Sproule at JAX. The video dataset was tracked, **and behavior annotations algorithmically generated**, by Brian Geuther and Keith Sheppard at JAX, with pose estimation performed using a modified version of HRnet described in [64]."*

Sun et al was updated to include mention of new human annotations in late June 2023 for their new version that was published elsewhere (Fri, 30 Jun 2023 22:45:47 UTC (28,628 KB):<https://arxiv.org/abs/2207.10553v2>), but notably **while our paper was under revision review!** Here again is proof of the actual data: <https://data.caltech.edu/records/rdsa8-rde65> where the record clearly shows June 29th, 2023 as the uploaded and modified date (see bottom of the page). How many papers have used this "excellent and widely adopted reference for testing precisely what the authors purport to show here" since June 30th?

And what do we purport to show here that others have done? We purport to show a method to build better unified pose estimation models (which is novel), and as a simple proof-of-usability (which I don't see other pose estimation papers doing), we show performance on five OOD videos (with metrics), and two benchmark datasets where pose performance would be critical (Sturman et al. and Horse-30, which both have human annotated videos).

Thus, it's completely unreasonable to move the goalposts on us. *"When evaluating your revised manuscript, we will not consider any similar papers published independently in the meantime to compromise the novelty of your study. See here for more information."* Surely this must extend to arxiv updates and requests for new experiments where data wasn't even available during revision 1?

Nonetheless, while we find your statements blatantly untrue and misleading, we ran the MABe benchmark videos with SuperAnimal-TopViewMice. See below.

An alternative source of evaluation of downstream performance would include use with the openly available SimBA videos and pretrained classifiers (<https://osf.io/tmu6y/>), which come with DLC tracked data to begin with and should be straightforward to evaluate on.

SimBA is an unpublished method (and we don't need DLC data, we need raw videos that are out of domain such that we can run video inference).

Downstream evaluations (such as in supervised behavior segmentation), while not the innovations the authors intend to showcase here, are a crucial and important measure as they provide evidence of the real world performance of the approach. Importantly, they afford a fair and orthogonal evaluation of the performance of the tracking models by measuring the functional significance of the claimed improvements. Given their importance, it is not unreasonable to request a more thorough evaluation of downstream performance on more representative behavior segmentation datasets like the two mentioned above.

We don't disagree with the sentiment (as we did just that with two other datasets), but we want to point out that **no other published pose estimation paper we are aware of shows supervised behavioral segmentation as a readout of the pose estimation performance**. Every benchmark's (ie., <https://paperswithcode.com/task/pose-estimation>) goal is to show single frame pose performance. We find this request a dangerous precedent to set for the field. The cost and CO2 to just appease this one reviewer should be considered. So, although we fundamentally disagree with this reviewer and find it unethical to request more work that could not have been done before submission, **we benchmark on MABe and show our zero-shot keypoints are as good as their fully supervised models** (in-so-far-as the performance one can get with using either is the same).

The goal:

The aim of these experiments are to show that our pose estimation outputs are as usable as the officially released pose estimation outputs. Our goal is not to compete on behavioral classification benchmarks, as our paper has nothing to do with advances in behavioral classification.

The data available:

MABe has 2 rounds. Round 1 provided only pose estimation output data for users to build better unsupervised behavioral classifiers. Round 2 released raw videos (https://www.aicrowd.com/challenges/multi-agent-behavior-challenge-2022/problems/mabe-2022-mouse-triplets-video-data/dataset_files); it has raw RGB video data to build unsupervised representation learning models. Pose therefore was not a requirement to use.

Therefore, we use videos from round 2 as the inputs running inference with our for our SuperAnimal-TopViewMouse model. Since our paper is about building better pretrained pose models, we use recommended baselines (Sun et al. 2022) from round 1 that build representation based on pose trajectories instead of round 2 RGB-based representation learning baselines (as RGB-based representation learning is known to be better than pose trajectory-based representation (PointNet)).

Because videos from MABe round 2 have 3 mice in the videos, we used a top-down version SuperAnimal-TopviewMouse. See our Methods for information on the detector.

The procedure we took is as follows:

We inference our top-down SuperAnimal-TopviewMice model on all 1830 videos from round 2 (with 1800 frames per video, ie 3.3 Million frames of data total), converted the pose results into the MABe keypoint file format, and ran one of the strong pose-based representation baselines called PointNet. Finally, we use the official evaluation code to compare the performance between using the official MABe poses (obtained from fully supervised learning) and poses that are obtained via our models' zero-shot predictions. We added these results to Suppl. Table 28 and an example image in Extended Data Figure

7, and Suppl. Video 6. Image on the left is adapted from the MABE challenge:
https://www.aicrowd.com/challenges/multi-agent-behavior-challenge-2022/problems/mabe-2022-mouse-triplets_under_a_Creative_Commons_Attribution_4.0_International_License:
<https://creativecommons.org/licenses/by/4.0/>.

Task No.	Official MABe pose	SuperAnimal zero-shot
T0	0.095018	0.095018
T1	0.096345	0.096350
T2	0.657165	0.657245
T3	0.020959	0.020963
T4	0.34015	0.34020
T5	0.718520	0.718519
T6	0.565967	0.565954
T7	0.261730	0.261697
T8	0.005427	0.005427
T9	0.025384	0.025381
T10	0.021717	0.021703
T11	0.107985	0.107988
T12	0.610986	0.610956

We show that with SuperAnimal keypoints, we get almost the same performance in downstream action segmentation as the official pose does in all 13 considered tasks, even though our model is never trained on MABe videos. This demonstrates the effectiveness of our models in downstream action segmentation tasks.

>> Additionally, we note that SuperAnimal weights were used in another pre-print, showing that SuperAnimal-TopViewMouse predictions with keypointMoSeq [13] outperform other behavioral quantification methods on a separate benchmark dataset. For ease of the reviewers, we added that relevant figure panels here (Fig. 9). We cite this work as well. Additionally, we used the SuperAnimal-TopViewMouse model within AmadeusGPT [14] for several other mouse topview datasets, and for example, could show on the elevated plus maze it can match human-annotation performance there as well.

The Keypoint-MoSeq paper does not demonstrate that pose tracking with SuperAnimal-TopViewMouse outperforms pose tracking with any alternative approach – it evaluates the performance of different unsupervised behavior segmentation methods given the same tracking. Considering the amount of outlier filtering and other forms of robustness to keypoint noise specific to that method, I do not see this as a valid evaluation of the work in question here – maybe beside the point since none of that data is presented in THIS manuscript.

"The Keypoint-MoSeq paper does not demonstrate that pose tracking with SuperAnimal-TopViewMouse outperforms pose tracking with any alternative approach" -- this is not what we say at all. We simply note they, ZERO-SHOT, are good enough to be used for "behavioral quantification" – if they were terrible predictions how could they even be competitive?

There are a number of issues with the work presented in the AmadeusGPT preprint, all of which are outside the scope of this paper's review. If the authors wish to use it as evidence for the performance or validity of the current work, I encourage them to do so by including data on downstream behavior classification performance.

AmadeusGPT (now published at NeurIPS 2023) is an example to show the practical value of SuperAnimal models. We also find it wildly inappropriate to come after our other work in your review.

Since the model reported in this study is intended to be used directly for scientific applications, it is important that the authors share sufficient detail about the models, so that use applications do not suffer from any biases that might be baked into these models. I suggest providing the following (though the authors may have additional details they would like to provide): Dataset datasheet, as described in Gebru et al 2021 (arXiv:1803.09010) Crowdsourced annotation datasheet, as described in Diaz et al 2022 (arXiv:2206.08931) Model cards, as described in Mitchell et al (arXiv:1810.03993)

>> We updated the model cards at HuggingFace. TopViewMouse: <https://huggingface.co/mwmathis/DeepLabCutModelZoo-SuperAnimal-TopViewMouse>. SuperAnimal-Quadruped DLCRNNet: <https://huggingface.co/mwmathis/DeepLabCutModelZoo-SuperAnimal-Quadruped>

The efforts made to improve the documentation of the model is appreciated, but insufficient given the potential for widespread use of the work presented here.

The goal of this paper is to introduce a new method for building models with unified pose priors across disjoint data. We show in 6 benchmarks that is a really decent way forward for the field. Then, as examples of how such models can be used we show now 3 benchmarks/analysis for behavior as a downstream task. We release the weights in the realm of being open source and reproducible.

Aside from two new datasets, all of the data we used is already public and it is on those authors to document their datasets. **Moreover, your “minimum requirements” again are not a standard in the field.** Can the reviewer kindly point us to one other animal pose estimation paper that has such a model card? Or can they point us to one other pose model that is even as detailed as the ones we build for you in Revision 1?

Minimum requirements include:

1. A dataset datasheet should be provided as described in Gebru et al. (arXiv:1803.09010) including sections: Motivation, Composition, Collection Process, Preprocessing/Cleaning/Labeling, Uses, Distribution, and Maintenance, as well as associated sub-sections. Appendix A in (arXiv:1803.09010) has a clear example.
2. A model card should be provided as described in Mitchell et al. (arXiv:1810.03993) including sections: Model Details, Intended Use, Factors, Metrics, Training Data, Evaluation Data, Ethical Considerations, Caveats and Recommendations. Figs 2 and 3 in (arXiv:1810.03993) have clear examples.

These are the standard for foundation models which the authors state they consider this work to be. For a recent example, see Segment Anything (arXiv:2304.02643) which takes the exact steps described above (Appendix F) to ensure responsible and ethical use of their foundation model. As the intended use case for this work is the scientific domain, it would be appropriate to apply even more stringent reporting requirements, so this is not a particularly high bar to meet.

We are not responsible for documenting already publicly available data nor could we even do this. We did no preprocessing or edits to these datasets, as you allude to below. Nonetheless, we added Datasheets into the Supplemental information following your request.

> The authors state in the Reporting Summary that "In the following we detail references for those datasets", but these are not provided. I believe a major contribution of this study is a unified SuperAnimal dataset - the authors should provide this as part of the resource - the dataset should be made publicly available. Another major contribution is a unified vocabulary for animal keypoints, but this is only provided through figure illustrations rather than machine parseable text or code to do the dataset unification.

We literally reference every dataset we use, and we made iRodent, our new dataset, publicly available BEFORE publication to meet your demands. We will make the last of the data available (MausHaus) and the merged datasheet (again, no edits ...) available upon acceptance. We also have a machine parsable file in the paper repo and it was there for you to look at during revision 1.

>> The datasets are referenced fully in the Methods and in Suppl. Figure 1. Please note, not all of the mouse data can be made available due to limitations on institutional guidelines on sharing recorded data. Therefore, while we used primarily open source data, as we already referenced, those that are given to us courtesy of other scientists must stay private. The best we can do, as we do here, is to provide model weights. This is also an important point for scientists who might be hesitant to release their data. We also now added the unified keypoint main diagram to the figure repository so users understand the mapping, and a new panel in Extended Data 1 that has the mappings.

While references and ED1 are helpful, there is significant additional processing that the authors have done, many steps of which cannot presently be reproduced even with the descriptions provided.

We did absolutely zero preprocessing of the images or annotations. The only set that was held out due to ethics approval concerns is now cleared (300 images of ~85K) and will be released upon acceptance. Example merging code will also be released with the final paper.

The authors can also:

1. Provide the code for standardizing the pose annotations for the specific datasets used here.
2. Provide the publicly available datasets in their standardized format (or at a minimum, the pose annotations).

1 was already provided.

2 every dataset is provided. Upon acceptance will release the final set, and this is noted in the manuscript.

The model weights encode the biases and structural sources of error present in the source annotations, but these are intractable to audit without access to the source data and annotations.

All datasets and annotations are publicly available to you aside from ~300 images out of 85K, which we will package and release upon acceptance.

These datasets that cannot be submitted to scrutiny, but which constitute the core of the contributions reported here, can simply be excluded from the training set to create an auditable subset. Both versions

of the weights could be provided, with ample disclaimers that one model was trained on private and potentially problematic source data.

Frankly it's insulting that you think the core of our contribution of years of work is to collect public datasets into a common folder.

>> This app allows anyone, within their browser, to a) upload their own image and label, b) annotate community images, c) run inference of available community models on their own data, d) share models to be hosted.

The app does not appear to allow for uploading user images for annotation, and other than a Google Form, nothing appears to be present in the web app other than the curated images for public datasets. It is not clear how any of this, other than the inference functionality (also available via HuggingFace Spaces) pertains to the work presented here. If we understand correctly, the labels on the curated datasets are not used in this work.

That is incorrect. You can upload an image and easily test, and even contribute by fixing mistakes (see below). How is this related? You wanted ample evidence these models work, we benchmark behavior and provide users a non-install way to test them. Here is an image <https://stock.adobe.com/fr/search?k=horse>, not from our training or test set you can test on, for example:

[Redacted]

REVIEWERS' COMMENTS

Reviewer #2 (Remarks to the Author):

The revised benchmarking of model performance is much clearer, and I found this series of results sections (Zero-shot SA-TVM, Fine-tuning SA-TVM, Zero-shot SA-Q, and Fine-tuning SA-Q) helpful for their organization of the many model architectures, training methods, evaluation metrics, and datasets used in this paper. This is very nice work! The improvements to model performance are also quite substantial compared to the original manuscript, and I commend the authors for the work they've put in.

One minor note, the description of the robustness gain metric is clear in the rebuttal, but in the Methods of the revised text less detail is provided, and it might be confusing to readers as it stands. I suggest tweaking the Methods text to explicitly state that for robustness gain you're adapting on one video then evaluating on the remainder, vs in adaptation gain where you evaluate on the same video on which adaptation was performed.

And one small nitpick on the results section titled "unsupervised behavioral analysis" – what's being shown in this section isn't that SA does unsupervised behavioral analysis, it's that SA pose estimates are compatible with unsupervised behavioral analysis pipelines like Keypoint-Moseq (as well as supervised analyses like Sturman et al's rearing detection task.) "SA compatibility with unsupervised behavioral analysis" would be more appropriate.

These are quite minor points that the authors should easily be able to address. Once done, given the overall improvement in both the manuscript and the models themselves, I am happy to recommend this article for publication in Nature Communications.

Reviewer #2 (Remarks to the Author):

The revised benchmarking of model performance is much clearer, and I found this series of results sections (Zero-shot SA-TVM, Fine-tuning SA-TVM, Zero-shot SA-Q, and Fine-tuning SA-Q) helpful for their organization of the many model architectures, training methods, evaluation metrics, and datasets used in this paper. This is very nice work! The improvements to model performance are also quite substantial compared to the original manuscript, and I commend the authors for the work they've put in.

Thank you very much for the feedback!

One minor note, the description of the robustness gain metric is clear in the rebuttal, but in the Methods of the revised text less detail is provided, and it might be confusing to readers as it stands. I suggest tweaking the Methods text to explicitly state that for robustness gain you're adapting on one video then evaluating on the remainder, vs in adaptation gain where you evaluate on the same video on which adaptation was performed.

Thank you for noting this, we merged this into the main manuscript file.

And one small nitpick on the results section titled "unsupervised behavioral analysis" – what's being shown in this section isn't that SA does unsupervised behavioral analysis, it's that SA pose estimates are compatible with unsupervised behavioral analysis pipelines like Keypoint-Moseq (as well as supervised analyses like Sturman et al's rearing detection task.) "SA compatibility with unsupervised behavioral analysis" would be more appropriate.

We updated the header to read: "SuperAnimal models can be used with unsupervised behavioral analysis"

These are quite minor points that the authors should easily be able to address. Once done, given the overall improvement in both the manuscript and the models themselves, I am happy to recommend this article for publication in Nature Communications.